# Shared sensitivity to data distribution during learning in humans and transformer networks

Jacques Pesnot Lerousseau ◎ [1,2,3] ✉ & Christopher Summerfield ◎ [1] ✉

Do humans learn like transformers? We trained both humans ($n = 530$) and transformer networks on a rule learning task where they had to respond to a query in a sequence. At test, we measured 'in-context' learning (generalize the rule to novel queries) and 'in-weights' learning (recall past experiences from memory). Manipulating the diversity and redundancy of examples in the training distribution, we found that humans and transformer networks respond in very similar ways. In both types of learner, redundancy and diversity trade off in driving in-weights and in-context learning, respectively, whereas a composite distribution with a balanced mix of redundancy and diversity allows the two strategies to be used in tandem. However, we also found that while humans benefit from dynamic training schedules that emphasize diverse examples early, transformers do not. So, while the same data-distributional properties promote learning in humans and transformer networks, only people benefit from curricula.

The relationship between memory and reasoning is among the oldest problems in the cognitive sciences. Humans can make strong inductive inferences, allowing them to reason about novel data—for example, using the laws of calculus to compute integrals on a maths exam, or applying grammar rules to understand a sentence never heard before. However, the ability to encode and retain specific instances of past experience in memory is also a critical hallmark of healthy cognitive function. This duality was first articulated in the 1940s by Cattell, who distinguished 'crystallized' from 'fluid' intelligence—the former indexing the integrity of core skills and knowledge and the latter our ability to reason beyond extant data[1]. This dichotomy prefigured seminal dual-process frameworks in psychology and neuroscience, which separated heuristics from rational computation[2], information integration from rule-based categorization[3], associative from symbolic processes[4] and model-free from model-based reinforcement learning[5]. However, the nature of the computations that allow humans (and perhaps other animals) to use both memory and inductive inference to solve complex problems remains an open question in psychology, neuroscience and artificial intelligence research.

Throughout the twentieth century, symbolic systems that strictly separated memory and inference remained popular[6–8], but connectionist models have since reemerged as theories of biological cognition[9,10]. Neural networks can be trained either to store and retrieve information from memory or to learn generalizable patterns in data[11], doing so by modifying their weights, which serves both to store information and support generalization ('in-weights' learning). Nevertheless, one surprising finding is that modern deep networks can be pretrained to generalize over patterns in sequential data after just a few examples, a capacity (dubbed 'in-context' learning) that is reminiscent of human inductive inference[12,13]. Rather than relying on weight updates, in-context learning arises from the networks' internal processing: it is best understood as an emergent result of meta-learning, where training leads the network to 'learn how to learn' from the structure of its input, enabling it to perform few-shot learning without updating its weights[14,15]. In-context learning has come to prominence with the arrival of a new neural network architecture known as the transformer. Transformer networks use self-attention to compute how much each token in a sequence should influence the representation of every other

[1]Department of Experimental Psychology, University of Oxford, Oxford, UK. [2]Institut de Neurosciences des Systèmes, Aix-Marseille Université, Inserm, Marseille, France. [3]Institute for Language, Communication, and the Brain, Aix-Marseille Université, Marseille, France. ✉e-mail: jacques.pesnot-lerousseau@univ-amu.fr; christopher.summerfield@psy.ox.ac.uk

token. This allows the model to integrate information across positions and build context-aware representations at each layer[16]. Large transformer networks trained on giant text corpora are able to generate fluent sentences, equations or code on the fly[17–19], and it has been claimed that these networks can make inferences beyond their training data in ways that resemble human fluid intelligence[20,21]. Conversely, the idea that human cognition might emerge from a relatively undifferentiated neural network architecture has once again become fashionable in the neurosciences[22,23]. While the distinction between in-context and in-weights learning is reminiscent of dual-process frameworks, it is important to note that classical dual-system models do not make specific predictions about how learning strategies should vary with the statistical structure of the training data. This is the central focus of our work.

In a recent line of work, machine learning researchers have studied how the distributional properties of training data variously promote in-weights (memory-based) and in-context (inference-based) learning in transformer networks[24–33]. Using cleverly designed probes that can distinguish the two types of learning, researchers have shown that training distributions that involve lots of repetitions (redundancy) promote in-weights learning, whereas distributions that involve lots of diverse examples (diversity) promote in-context learning, with hints that a sweet spot may exist in between. Here we asked whether the results reported in these papers also hold true for human participants performing a comparable task. We found that human learners and transformers respond to the training data distribution in remarkably similar ways, and that a near-identical manipulation allows both humans and transformers to learn in-weights and in-context solutions in tandem. However, we also observed an important dissociation: humans, but not transformers, benefit from curricula that prioritize diverse examples early on in training.

## Results

### Transformers trade off in-context and in-weights learning depending on the training data distribution

We adapted a paradigm previously used to distinguish in-context and in-weights learning in transformers[32]. On each trial, the learner is prompted with a sequence of {item: *label*} pairs, and then a single item is queried for its label {**item:?**}. A real-world analogy might be learning vocabulary items in a foreign language. For example, during training the learner sees pairs such as the following:

oiseau: *bird*; chien: *dog*; <u>chat</u>: <u>*cat*</u>; poisson: *fish*; **chat:?** (training trial)

At test, we can evaluate both in-context and in-weights learning by varying the novelty and familiarity of the sequences. These evaluations occur without any feedback (or gradient updates). In-context learning is indexed by zero-shot performance on previously unseen sequences with comparable structure, such as:

<u>katze</u>: <u>*cat*</u>; hund: *dog*; vogel: *bird*; fisch: *fish*; **katze:?** (in-context test trial)

By contrast, in-weights learning is quantified as a tendency to repeat answers to queries previously experienced during training, ignoring any contextual information:

pferd: *horse*; hund: *dog*; vogel: *bird*; fisch: *fish*; **chat:?** (in-weights test trial)

We used this approach to study how the training data distribution influences the learning strategy used by transformer networks and humans (Fig. 1a–c). Like previous studies involving transformers only, we varied the diversity and redundancy of training examples. To illustrate, consider two extremes: a fully redundant distribution in which every training trial contains the same item–label pair and a fully diverse distribution in which every trial contains entirely novel item–label pairs. We can interpolate between these extremes by sampling

trials from a rank-frequency (or Zipfian) distribution parameterized by the exponent $\alpha$, where $\alpha$ controls the skewness (Fig. 1a). At $\alpha = 0$, the distribution is uniform (fully diverse); at $\alpha > 0$, the distribution is skewed, and in the limit of $\alpha \to +\infty$, the distribution is concentrated around one single example (fully redundant).

Using this task, we first attempted to replicate previously reported findings using a simple transformer architecture comprising two attention-only layers (one attention head each) followed by a classifier (Fig. 1d and Extended Data Fig. 1). Inputs were coded as vectors sampled from multidimensional Gaussian distributions (Methods). We first confirmed that transformers were able to learn the task (Fig. 2a). Indeed, on training trials, transformers learned well irrespective of the statistics of the training distribution (all accuracies near 100%, except when $\alpha = 1$). However, at test, we found that performance varied sharply with the distributional properties of the training data. Transformers trained on a uniform distribution ($\alpha = 0$) scored nearly perfectly on in-context test trials (accuracy of 100%), whereas those trained on a skewed distribution scored close to chance on these trials (~10% for transformers trained on $\alpha > 1$). By contrast, transformers trained on a uniform distribution ($\alpha = 0$) performed at chance on in-weights test trials, whereas transformers trained on a skewed distribution scored very highly (accuracy near 100% for transformers trained on $\alpha = 4$). In both cases, a transition between these two regimes occurred close to $\alpha = 1$, at which point approximately half of transformers learned to solve the task, and half remained at chance. These findings, which are shown in Fig. 2a, replicate previous reports that the relative balance between in-weights and in-context learning depends on the distribution of examples in the training data[30,32].

We also trained other classes of neural networks on the task, including feed-forward architecture (multi-layer perceptron (MLP)) and long short-term memory (LSTM) networks. In general, these architectures had no difficulty learning the task, but none showed effective in-context learning (Fig. 2c and Extended Data Fig. 2). This result replicates previous findings showing that neural architecture matters for in-context learning[32]. It should be noted that in-context learning is not exclusive to transformer networks—under specific conditions, both feed-forward and recurrent architectures such as LSTM networks can learn in-context[34,35]. However, transformers adopt this strategy more robustly and flexibly across a wider range of settings, including those used in our study. This is probably due to the attention mechanism, which explicitly provides an opportunity to integrate information present in the context when processing the query (see the mechanistic interpretability analysis below).

### Humans trade off in-context and in-weights learning in a similar manner to transformers (Experiment 1)

Next, we designed a variant of the task that could be performed by human participants, recruited via an online platform. The context sequence was composed of seven alternating images (items) and numbers (labels) that were presented in a ring. The query item was presented centrally, inside the ring. During training, the query image was always also present in the context (for example, 'cat' in the example above; we call this the 'target image'). As shown in Fig. 1b, the correct (or target) label was always located three steps clockwise from the target image. Participants responded by pressing a digit between 0 and 9 on their keyboard. They were not instructed as to the rule but learned gradually from fully informative feedback that was provided after each trial. Thus, during training, agents could use two strategies to solve the task: they could either memorize the class label for each image from the trialwise feedback (in-weights learning), or they could learn the '+3 steps' rule to infer the correct label from the context of any sequence, including potentially novel sequences (in-context learning).

We used a between-group design, in which four groups ($n = 30$ each, Experiment 1) experienced training distributions characterized by different parameters $\alpha \in \{0, 1, 2, 4\}$. The results are presented in

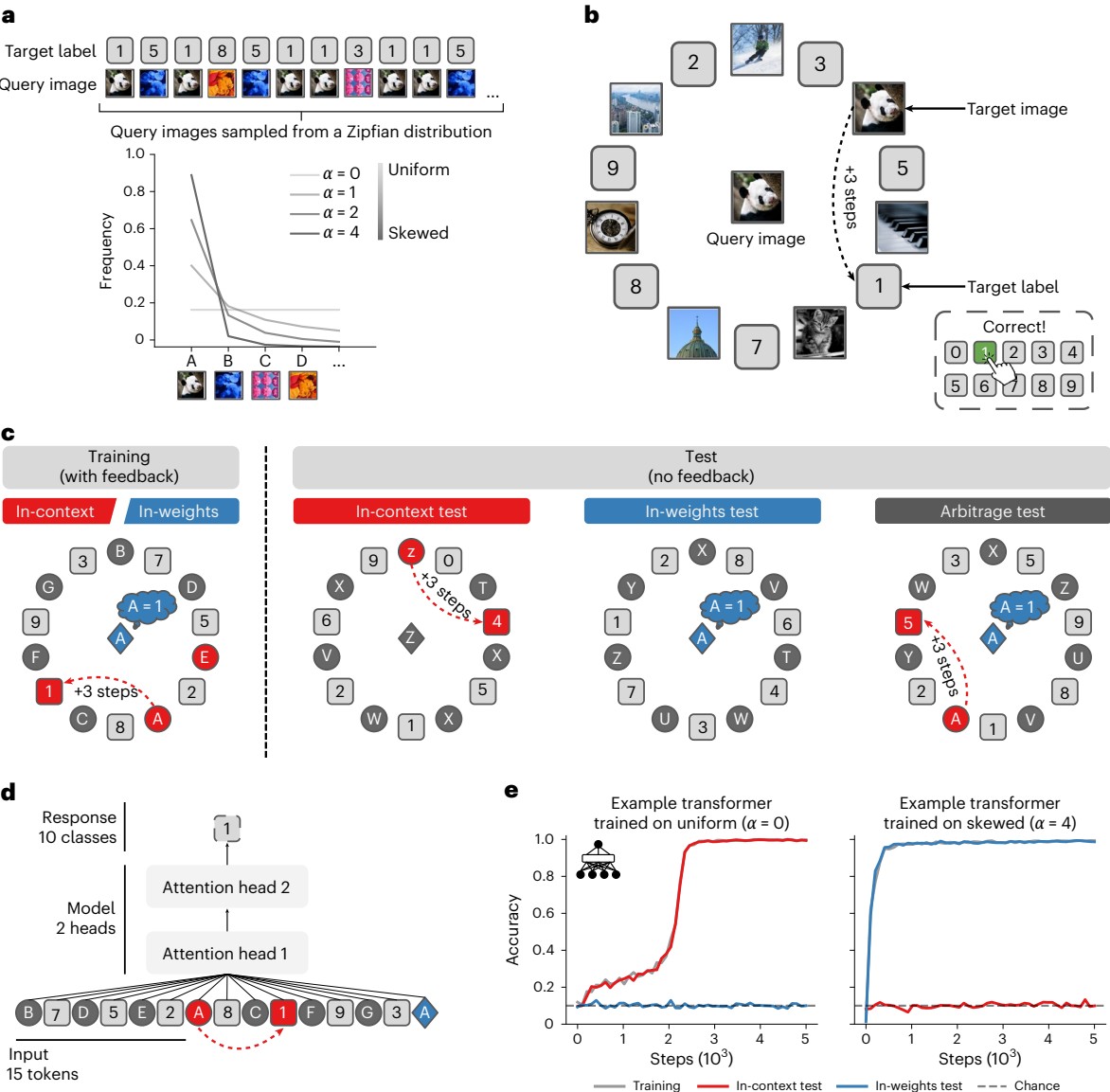

**Fig. 1 | Paradigm. a**, We studied learning in an image–label association task by manipulating the distribution of the training data. Under a uniform distribution ($\alpha = 0$), all images are equally likely to appear. In skewed distributions, some images are more likely than others ($\alpha > 0$). **b**, Example training trial. In a given trial, agents were asked to select the label corresponding to the query image, presented at the centre of the screen. Seven images and seven labels were also presented in a surrounding circle (the context). During training, a copy of the query image (the target image) was always present in the context. The correct label was always located three steps clockwise relative to the target image (the target label). **c**, Paradigm overview. During training, two learning strategies are available. The in-context learning strategy consists in using the context to infer the correct label—that is, using the '+3 steps' rule. The in-weights learning strategy consists in learning each image–label association in memory using the feedback. Test blocks were designed to probe which strategy (or strategies) the agent is using. On in-context test blocks, novel images (depicted in grey)

were presented, such that the only way to be accurate was to use information from the context—that is, the in-context strategy. On in-weights test blocks, a training image (depicted in blue) was presented as the query image, but novel images (depicted in grey) were presented in the context, such that the only way to be accurate was to use information stored in memory—that is, the in-weights strategy. On arbitrage test blocks, a training image was presented as the query image, and the context indicated a different label than the one that was presented during training. This was done to reveal the dominant strategy used by the agent when presented with conflicting evidence for the two strategies. **d**, A minimal transformer model, composed of two attention-only layers of one attention head each, was trained on the task. **e**, Accuracy curves for two example transformers trained on two different training distributions, uniform ($\alpha = 0$, left) and skewed ($\alpha = 4$, right). When $\alpha < 1$, transformers learn in-context but not in-weights. Conversely, when $\alpha > 1$, transformers learn in-weights but not in-context.

Fig. 2b. Like transformer networks, humans in all four groups learned to become proficient at the task. They had mostly reached a stable level of accuracy by the final training block (average accuracy of $85.6 \pm 2.3\%$), and the data distribution did not impact their performance in training (effect of $\alpha$ on accuracy, $\beta = 0.247 \pm 0.179$; $P = 0.168$; Bayes factor (BF), 0.042; 'strong' evidence in favour of an absence of effect). Thus, as for transformers, manipulating the training data distribution did not immediately affect agents' learning or their ability

to associate images with labels, as performance remained consistent regardless of $\alpha$.

However, again like transformer networks, the performance of human participants at test was greatly influenced by the training data distribution. This was the case for both in-context test trials (effect of $\alpha$ on accuracy, $\beta = -1.543 \pm 0.208$, $P = 0.0$, BF > 100, 'decisive' evidence) and in-weights test trials (effect of $\alpha$ on accuracy, $\beta = 1.89 \pm 0.106$, $P = 0.0$, BF > 100, 'decisive' evidence). Similar to transformers,

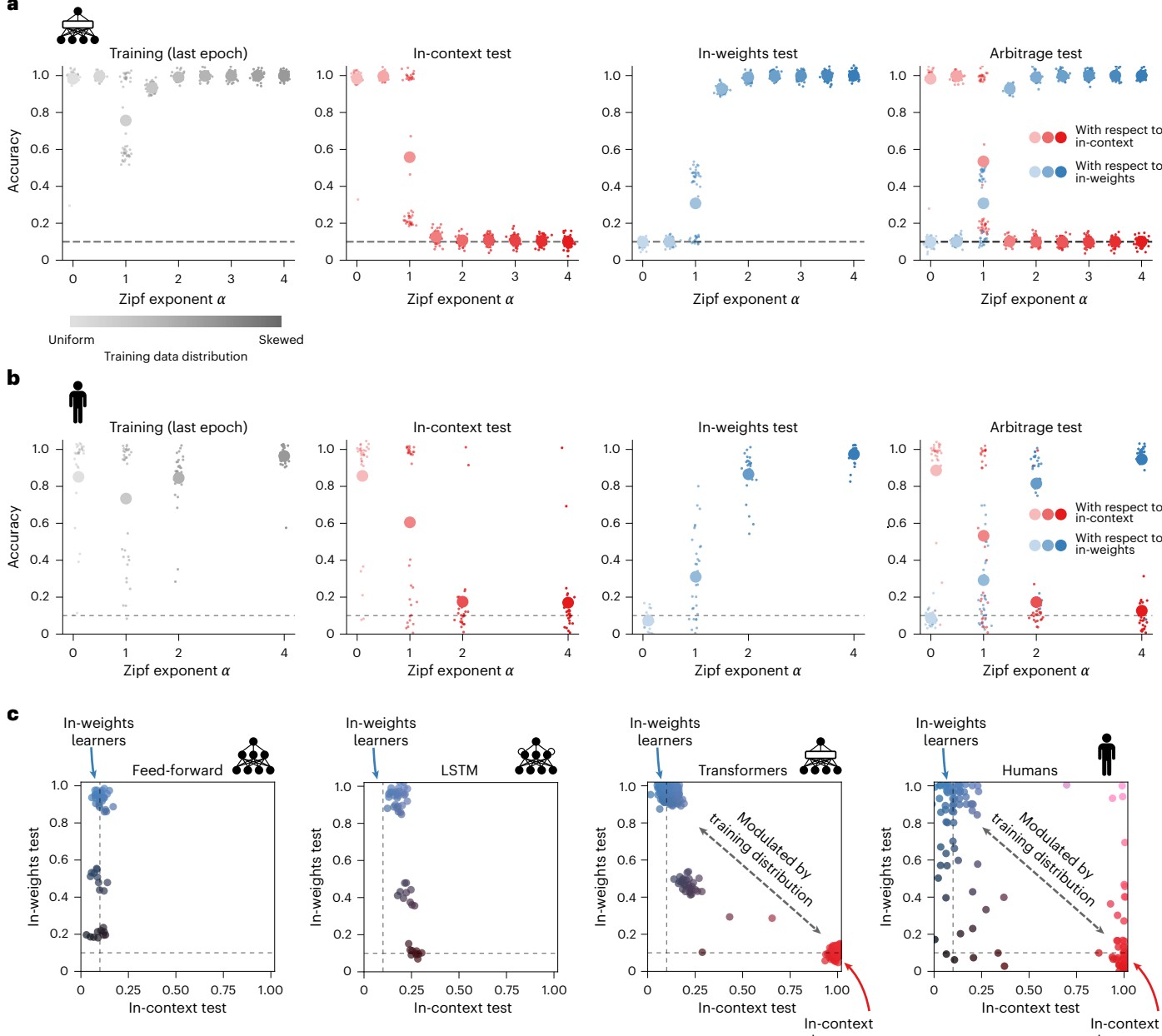

**Fig. 2 | Transformers and humans trade off in-context and in-weights learning depending on the training data distribution (Experiment 1). a**, Training and test performances for transformers ($n$ = 30 per training data distribution). **b**, Same for human participants (Exp. 1, $n$ = 30 per training data distribution). The small dots indicate data from individual transformers/humans; the large dots indicate group averages. **c**, Scatter plots of the in-context versus in-weights test performances for feed-forward networks (left), LSTM networks (middle left), transformers (middle right) and humans (right). Feed-forward and LSTM networks do not learn in-context. Transformers and human participants trade off in-context and in-weights learning. Each dot indicates data from an individual model/human.

participants trained on a uniform distribution were very accurate on in-context test trials (85.7 ± 5.3% for the group trained on $\alpha = 0$), while participants trained on a skewed distribution were near chance level (17.0 ± 3.9% for the group trained on $\alpha = 4$). Conversely, on the in-weights test, participants trained on a uniform distribution were at chance level (7.26 ± 0.8% for the group trained on $\alpha = 0$), while participants trained on a skewed distribution showed near-perfect performance (97.4 ± 0.8% for the group trained on $\alpha = 4$). Once again, a transition between successful strategies occurred around $\alpha = 1$. These findings are reported in Fig. 2b.

To better understand what drives performance in the in-weights test, we analysed accuracy as a function of item frequency during training (Extended Data Figs. 3 and 4). Both transformer networks and

human participants performed better on frequent items, confirming that they learned from repeated exposure.

Finally, we also used a class of test that we call an 'arbitrage' trial, designed to disambiguate in-context and in-weights responding with a single query. Arbitrage test trials resembled in-weights test trials in that the query matched examples in the training data, and so the trial could be solved from memory. However, they also resembled in-context test trials, in that the query item was repeated in the context, so that the +3 rule could be applied. Crucially, the query item was paired with a different label in the context than the one it was paired with during training.

vogel: *bird*; hund: *dog*; <u>chat</u>: <u>*kitty*</u>; fisch: *fish*; **chat:?**
(arbitrage test trial)

Arbitrage trials had no inherently correct answer but allowed us to evaluate whether humans and transformer networks were using an in-context or an in-weights approach to solve the trial. We posed this type of trial to both human participants and transformer networks. Note that this condition is nearly identical to set-ups used in recent machine learning studies: the 'ICL2' trials in ref. 30, the 'Flip' condition in ref. 36 and the 'Swap' condition in ref. 35.

The results followed a similar pattern to those observed on in-context and in-weights test trials. Transformers trained on uniform data ($\alpha = 0$) responded according to in-context learning and not in-weights learning, whereas transformers trained on skewed data ($\alpha > 1$) responded the other way around. Once again, transformers trade off in-weights for in-context learning around $\alpha = 1$. Similarly, human participants responded according to in-context learning when trained on a uniform distribution ($\alpha = 0$) and progressively more according to in-weights learning as the skewness of the distribution increased ($\alpha > 0$). Indeed, we observed a strong negative effect of $\alpha$ on accuracy with respect to in-context learning ($\beta = -1.542 \pm 0.183$, $P = 0.0$, BF > 100, 'decisive' evidence) and a strong positive effect of $\alpha$ on accuracy with respect to in-weights learning ($\beta = 1.752 \pm 0.111$, $P = 0.0$, BF > 100, 'decisive' evidence). Note that these two accuracies do not necessarily sum to 1, as agents can respond according to neither strategy.

To confirm the robustness of our findings, we conducted a preregistered replication of Experiment 1 with a new sample of human participants ($n = 30$ per training distribution; the preregistration is available at AsPredicted no. 231356, https://aspredicted.org/rqgz-rdfk.pdf). All key effects were replicated (Extended Data Fig. 4), including the trade-off between in-context learning and in-weights learning as a function of the training distribution.

### In-context and in-weights learning trade off in both humans and transformer networks

In all three types of test trial, we observed a transition in learning strategies that occurred around $\alpha = 1$. At this point transformers and humans seem to trade off in-context for in-weights learning. This implies that no (or very few) agents learn both strategies simultaneously. We confirmed that this was the case by plotting individual transformers' and individual participants' in-context test performance against their in-weights test performance (Fig. 2c). The majority of transformers were either pure in-context learners (26.7%; cluster of red points in the bottom right in Fig. 2c) or pure in-weights learners (66.7%; cluster of blue points in the top left in Fig. 2c), whereas just 6.7% learned both strategies. Similarly, most human participants were clustered in two groups, corresponding to in-context and in-weights learners (negative correlation between in-context and in-weights across the entire cohort, $\beta = -0.286 \pm 0.097$, $P = 0.004$, BF = 6.66, 'strong' evidence). The majority of transformers and humans thus appear to trade off between in-context and in-weights learning, favouring one strategy depending on the data distribution.

Nevertheless, we noted that a few participants had good performance in both tests (5/127, 4%), meaning that humans can in principle learn both strategies simultaneously. Similarly, a few transformers had better-than-chance—but poor—performance in both tests (6.7%; cluster of grey points in Fig. 2c). These transformers learned some image classes in-weights but also discovered a suboptimal in-context learning strategy consisting in choosing one random label from the context, reducing the chance performance from 1/10 to ~1/7, thus slightly improving performance. All these models were trained with the critical value $\alpha = 1$ (on a side note, they are also the models that did not reach perfect performance at the end of training; Fig. 2a). This suggests that transformers can also in principle learn both strategies independently and at the same time, although a Zipfian distribution might not be optimal. This is what we explored in Experiment 2.

### Transformers and humans learn both strategies in tandem when exposed to a non-Zipfian, composite training distribution (Experiment 2)

Experiment 1 revealed that a training distribution with maximal diversity ($\alpha = 0$) promotes in-context learning, while training with high levels of redundancy ($\alpha > 1$) promotes in-weights learning. Crucially, however, we see that in both humans and transformer networks, a training distribution that advantages one type of learning seems to impair the other, so that no (or very few) learners were able to acquire both an in-weights and an in-context strategy. Inspired by this result, we reasoned that a distribution that contains a mix of redundancy and diversity might favour learning both strategies at the same time. We thus moved beyond standard Zipfian distributions and created a 'composite' distribution where a fraction $P_c$ of the query images are sampled from a uniform distribution ($\alpha = 0$) and the remainder are sampled from a skewed distribution ($\alpha_s > 0$) (Fig. 3a).

First, we trained the same transformer architecture on this composite distribution. The results from a full sweep of parameters are shown in Extended Data Fig. 5, but here we focus on the case where $P_c = 0.5$ and $\alpha_s = 2$. In contrast to what we observed with Zipfian distributions, under this parameterization transformers performed well in both in-context and in-weights test trials simultaneously. Plotting individual transformers' in-context test performance against their in-weights test performance revealed a large cluster of models located in the top-right corner (~31/50, 62%; Fig. 3d, left). These models have high levels of accuracy in both in-context and in-weights. This confirms that transformers are able to learn both strategies independently, if exposed to a distribution containing both redundant and diverse training examples.

Human participants trained on this composite distribution (Experiment 2; Fig. 3b) also had high levels of accuracy for both in-context test trials ($65.6 \pm 5.9\%$) and in-weights trials ($57.4 \pm 5.0\%$). Note that this does not directly imply that participants learned both strategies simultaneously, as what is true at the population level might not be reflected at the individual level—there could simply be two subgroups, one learning in-context and one learning in-weights. We thus introduced a 'double learning index' to quantify the amount of learning of both strategies at the individual level. Formally, it was computed as a product of the individual performance in-context and in-weights trials scaled to account for chance level (Methods). The index varies between 0 (when the individual is at chance in either one of the two tests) and 1 (when the individual has perfect performance in both tests). We confirmed that human participants had a greater double learning index value when trained on a composite distribution ($0.27 \pm 0.05$ a.u.) than when trained on a uniform distribution ($\alpha = 0$, $-0.02 \pm 0.01$ a.u.; difference between groups, $\beta = -0.295 \pm 0.066$, $P = 0.0$, BF > 100, 'decisive' evidence) or a skewed distribution ($\alpha = 2$, $0.08 \pm 0.04$ a.u., $\beta = -0.191 \pm 0.067$, $P = 0.005$, BF > 100, 'decisive' evidence) (Fig. 3c). We further confirmed that human participants truly became 'double learners' by plotting individual participants' in-context test performance against their in-weights test performance (Fig. 3d). We observed a large cluster of double-learners participants (17/50, 34%), located in the top-right corner.

### Humans, but not transformers, benefit from curricula that prioritize diverse samples early on in training (Experiment 3)

We have so far investigated static, unstructured training regimes, where examples are sampled independently and identically across training. Next, we asked whether a dynamic training curriculum would improve learning in transformers and humans. The question was whether the order of presentation of the trials would influence performance—for example, because learning one strategy interacts with the learning of the other strategy. For that, we used the same composite distribution as previously, known to promote the learning of both strategies, but we manipulated the order of the skewed and uniform trials across training.

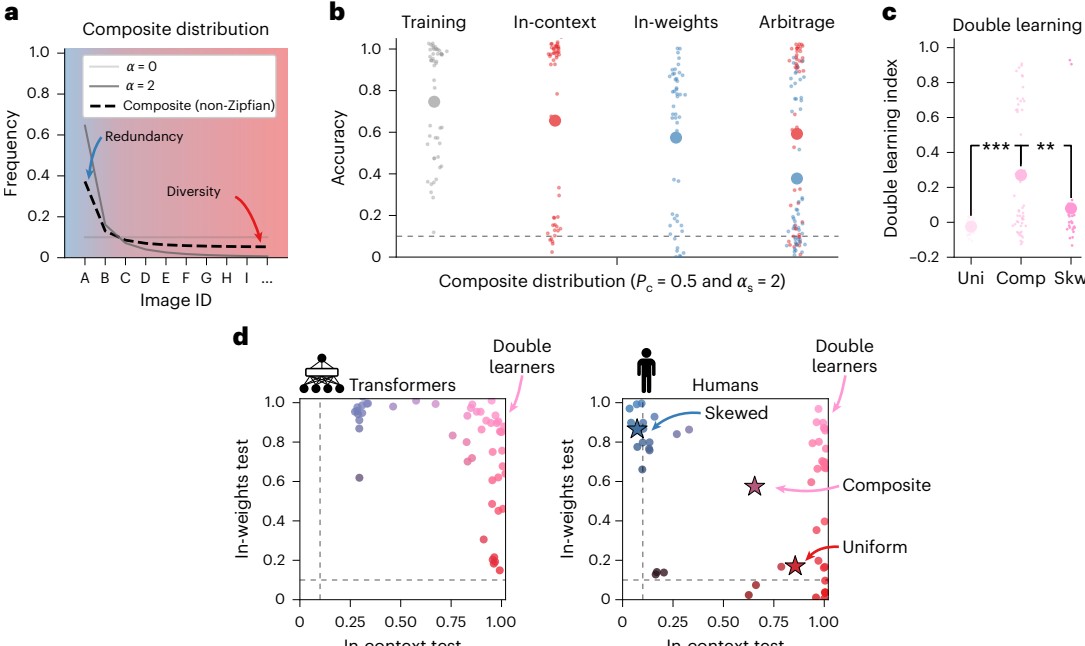

**Fig. 3 | Transformers and humans can learn both in-context and in-weights when trained on a composite, non-Zipfian distribution (Experiment 2).** **a**, Composite distribution, where a fraction $P_c = 0.5$ of the query images are sampled from a uniform distribution ($\alpha = 0$) and the rest from a skewed distribution ($\alpha_s = 2$). This distribution contains redundant images, thus promoting in-weights learning, but also rare, diverse images, thus promoting in-context learning as well. **b**, Training and test performances of humans (Exp. 2, $n = 50$) when training query images were sampled from this composite distribution. On average, human participants became accurate in both in-context and in-weights test blocks. The small dots indicate data from individuals; the large dots indicate group averages. **c**, Double learning index for human

participants trained on uniform (Uni, $\alpha = 0$, Exp. 1, $n = 30$), composite (Comp, Exp. 2, $n = 50$) and skewed distributions (Skw, $\alpha = 2$, Exp. 1, $n = 30$). Human participants had a greater double learning index value when trained on a composite distribution than when trained on a uniform distribution (linear regression with the group as a fixed effect, $\beta = -0.295 \pm 0.066$, $P = 0.0$, BF > 100, 'decisive' evidence) or a skewed distribution ($\beta = -0.191 \pm 0.067$, $P = 0.005$, BF > 100, 'decisive' evidence). **$P < 0.01$; ***$P < 0.001$. **d**, Scatter plots of the in-context versus in-weights test performances for transformers (left) and human participants (right). The dots indicate data from individual transformers/ humans. The stars indicate group averages for uniform (blue, $\alpha = 0$, Exp. 1), composite (pink, Exp. 2) and skewed distributions (red, $\alpha = 2$, Exp. 1).

Specifically, we designed two training curricula for transformers. The first curriculum (C1) involved maximally diverse exemplars in the first half of the training (the 'uniform part' of the composite distribution, $\alpha = 0$) and then more redundant exemplars in the second half of the training (the 'skewed part' of the composite distribution, $\alpha_s > 0$). The second curriculum (C2) reversed this ordering (Fig. 4a). Transformers trained on these curricula failed to become double learners. Indeed, the double learning index was near zero for all transformers, and this was true for a wide range of $\alpha_s$, as shown in Fig. 4d. Even in an extremely skewed regime ($\alpha_s = 4$), transformers do not become good double learners. In fact, there is an important interference from learning in initial trials. For example, when $\alpha = 4$, 92% of the trials are dominated by one item–label pair and >99% by the first five item–label pairs, so in-weights learning should be straightforward. Nevertheless, when initially trained on a uniform distribution (C1), transformer networks failed to learn this task. These data are illustrated in Fig. 4e and Extended Data Fig. 6, which shows the test performance of transformers trained on C1 or C2 as training progresses. During the first part of the C1 training, transformers become pure in-context learners (the red curve goes to the bottom-right corner). In the second part of the C1 training, transformers progressively forget the in-context strategy as they learn in-weights (the red curve goes to the top-left corner). A double-learning transformer would keep high performance for in-context trials while learning in-weights (the red curve would go to the top-right corner). We observed the same pattern in opposite directions for transformers trained on C2 (the blue curve in Fig. 4e). Thus, transformers converge towards one strategy during the first part of the training according to the training distribution, but then forget this strategy, showing a form of catastrophic interference[37,38].

We next used a similar approach to investigate this question in humans (Experiment 3). Training was composed of four blocks: two blocks where query images were sampled from a uniform distribution ($\alpha = 0$) and two blocks from a skewed distribution ($\alpha_s = 2$). We then defined a curriculum as a permutation of the block order, denoted C1 and C2 (Fig. 4a). We used a between-group design, in which two groups of human participants ($n = 50$ per group) each experienced one curriculum. Both groups thus experienced the same trials but not in the same order. We preregistered our predictions prior to data collection (AsPredicted no. 173550, https://aspredicted.org/yhvp-6y3y.pdf, hypothesis H1). On the basis of pilot data, we predicted that C1 would favour in-context learning while not impairing in-weights learning relative to C2. The results are shown in Fig. 4b,c and reveal that, in line with our predictions, participants trained on C1 showed better performance on in-context trials than participants trained on C2 (difference between groups, $\beta = -2.635 \pm 0.784$, $P = 0.019$, BF = 5.3, 'substantial' evidence, $P$ value Bonferroni corrected). This was also the case in arbitrage trials, where participants trained on C1 responded more using the in-context strategy than participants trained on C2 (difference between groups, $\beta = -2.676 \pm 0.711$, $P = 0.004$, BF = 12.1, 'strong' evidence). However, participants in both groups had the same performance on in-weights trials (difference between groups, $\beta = 0.096 \pm 0.428$, $P = 1.0$, BF = 0.019, 'strong' evidence).

These results suggest that, in line with our preregistered predictions, a human curriculum that prioritizes diverse examples early on in training (C1) is beneficial for in-context learning while not impairing in-weights learning in humans. We believe this reveals an asymmetry between in-context and in-weights learning in humans. Participants can still learn image–label associations even when they have discovered the

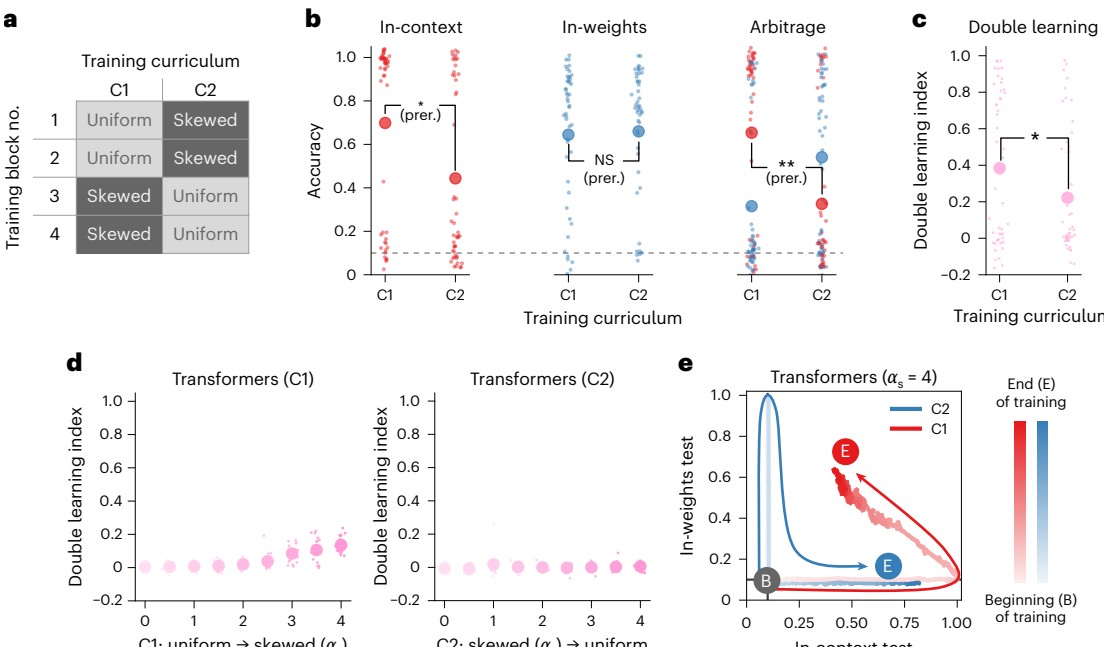

**Fig. 4 | Humans, but not transformers, benefit from a training curriculum promoting in-context learning first (Experiment 3). a**, Training curricula based on the composite distribution ($P_c = 0.5$, $\alpha_s > 0$). The first curriculum (C1) involved maximally diverse exemplars in the first half of the training (the uniform part of the composite distribution, $\alpha = 0$) and then more redundant exemplars in the second half of the training (the skewed part of the composite distribution, $\alpha_s > 0$). The second curriculum (C2) reversed this ordering. **b**, Two groups of human participants (Exp. 3, $n = 50$ per group) were exposed to two training curricula, C1 and C2 (composite distribution, $P_c = 0.5$, $\alpha_s = 2$). Human participants trained on C1 showed better performance on in-context trials than participants trained on C2 (logistic regression with the group as a fixed effect, $\beta = -2.635 \pm 0.784$, $P = 0.019$, BF = 5.3, 'substantial' evidence, $P$ value Bonferroni corrected). They also responded more using the in-context strategy in arbitrage trials ($\beta = -2.676 \pm 0.711$, $P = 0.004$, BF = 12.1, 'strong' evidence, $P$

value Bonferroni corrected). However, both groups had similar performance on in-weights trials ($\beta = 0.096 \pm 0.428$, $P = 1.0$, BF = 0.019, 'strong' evidence, $P$ value Bonferroni corrected). The small dots indicate data from individuals; the large dots indicate group averages. NS, $P > 0.05$; *$P < 0.05$; **$P < 0.01$. NS, not significant; prer., preregistered contrasts. **c**, Double learning index of human participants (linear regression with the group as a fixed effect, $\beta = -0.162 \pm 0.076$, $P = 0.036$, BF = 0.969). *$P < 0.05$. The small dots indicate data from individuals; the large dots indicate group averages. **d**, Double learning index for transformers trained on the C1 curriculum (left) and the C2 curriculum (right). Transformers were trained with different values of $\alpha_s$ for the skewed part of the composite distribution. **e**, Test performances over the course of training of transformers trained on C1 (red) and C2 (blue). The bold lines indicate group averages ($n = 20$ transformers per curriculum). The arrows were manually added to emphasize the direction of the trajectories.

in-context rule (C1) but have trouble discovering the in-context rule if they are first exposed to a training regime that favours in-weights learning (C2). For completeness, we tested all permutations of the block order as well as two 'interleaved' curricula where uniform and skewed distributions alternate during training (C3 and C4). The results are presented in Extended Data Fig. 7 and show that no other group contrasts were statistically significant (all $P > 0.05$, Bonferroni corrected; Extended Data Table 1).

### Transformers and humans use an induction mechanism for in-context learning (Experiment 4)

One limitation of our comparison between transformers and humans is that it offers little insight into the mechanisms by which in-context learning is happening. To better understand the similarities between transformers and humans, we studied the inference process as it unfolds, using a mixture of tools from the emerging field of mechanistic interpretability (in transformers)[39] and a behavioural mouse-tracking study (in humans)[40]. The results suggest that both humans and transformers solve the task using a two-step process composed of a binding operation followed by a searching operation.

For transformers, we first trained a transformer on the $\alpha = 0$ distribution to create a pure in-context learning model. We then investigated the attention patterns of its two attention heads during an in-context learning test trial. Attention patterns can be illustrated as square matrices that plot how the transformer weights information about each item $i$ when predicting each other item $j$. First, in attention

head 1, the transformer associates each item with its corresponding label, which is located three positions ahead: we observed in Fig. 5a (matrix of attention head 1) that the attention weight for each item is concentrated on the token that is three positions ahead. This reflects a binding operation, where the attention head writes information about each item into the embedding of its corresponding label[41–43]. Crucially for the next step, it writes information about the target item into the embedding of the target label. Second, in attention head 2, the transformer searches for a match between the query item and the preceding context tokens. Since attention head 1 has already written information about the target item into the embedding of the target label, the match occurs at the target label's location: in Fig. 5a (matrix of attention head 2, last column), we see that the attention weights for the query item are concentrated on the target label. The model then reads the information stored at this label. This computational architecture has been previously described in detail in refs. 41,42,44 and is referred to as an 'induction head'. The two attention heads are essentially implementing a minimal induction operation of the form [A][B]…[A] → [B]. This copying operation indeed solves our in-context learning task 'item; label; … ; item:?'.

For humans, we trained a new in-person group of participants (Experiment 4, $n = 20$) on a uniform distribution ($\alpha = 0$) to induce in-context learning, alongside a control group ($n = 20$) who encountered a skewed distribution ($\alpha = 2$). For these participants, unlike in the previous experiments, we used a mouse-tracking paradigm to reveal the computational processes underlying human in-context inference

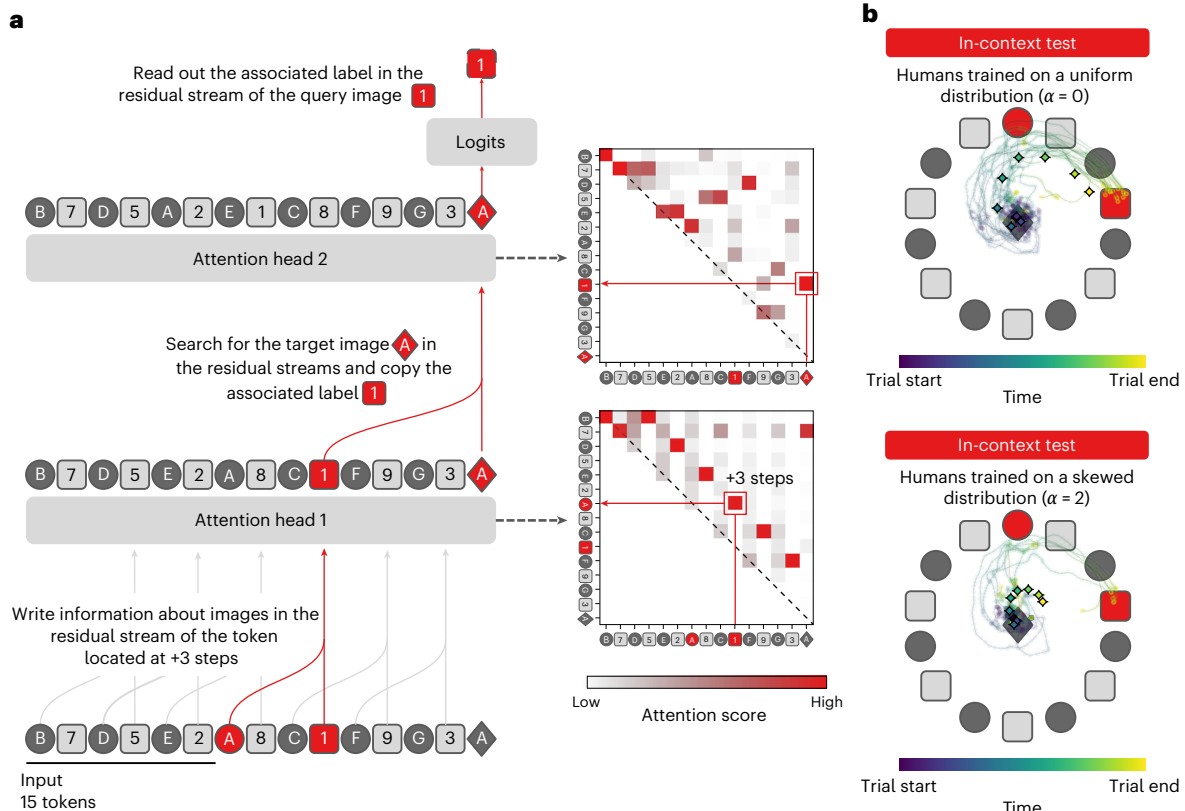

**Fig. 5 | Transformers and humans use an induction mechanism for in-context learning (Experiment 4). a**, Right, schematic representation of the computations realized by a two-layer transformer performing in-context learning. Left, attention matrices of both layers for the example sequence. The transformer binds the representations of the images and the labels in attention head 1 and searches for the target image in the context in attention head 2 (the induction head). **b**, Cursor trajectories of participants revealing their attention patterns. Top, trajectories in the in-context test block for human participants

trained on a uniform ($\alpha = 0$) distribution. Participants search for the target image in the context and then associate it with the target label. Bottom, trajectories in the in-context test block for participants trained on a skewed ($\alpha = 2$) distribution (Exp. 4, $n = 20$ per group). Trajectories were aligned trial-by-trial to a common frame where the target image is located on the top of the context circle. The small lines are individual average trajectories; the diamonds are group average trajectories.

as it unfolds (Methods, Fig. 5b and Extended Data Fig. 8). In test trials, the display was blurred and obscured, so that the locations of the images and labels could be seen but not their content. Participants were allowed to move a sharp aperture with their mouse to reveal part of the screen. Thus, similar to an eye-tracking device, tracking mouse position allowed us to track which information participants were viewing on the screen.

First, we confirmed that the participants trained on $\alpha = 0$ became in-context learners, whereas the participants trained on $\alpha = 2$ did not, replicating once again the results of Experiment 1. Indeed, the training data strongly influenced performance on in-context test trials (effect of $\alpha$ on accuracy, $\beta = -2.437 \pm 0.636$, $P = 0.0$, BF = 44.3, 'strong' evidence), in-weights test trials ($\beta = 2.262 \pm 0.129$, $P = 0.0$, BF > 100, 'decisive' evidence) and arbitrage test trials (effect of $\alpha$ on accuracy with respect to in-context learning, $\beta = 2.002 \pm 0.264$, $P = 0.0$, BF > 100, 'decisive' evidence). As in Experiment 1, we confirmed that the training distribution did not directly influence the performance at the end of training ($\beta = -0.09 \pm 0.408$, $P = 0.824$, BF = 0.027, 'strong' evidence) but only the strategy used by the participants.

Mouse trajectories are depicted in Fig. 5b (top). In step 1, after looking at the query image, participants search for the target image in the context. In step 2, once they have found the target image, they aim for the target label located at +3 steps clockwise and give a response. Note that these two steps correspond exactly to the two attention heads of the transformer: step 1 is implemented by attention head 2 (the searching operation), and step 2 is implemented by attention head 1 (the binding

binding operation). We quantified the occurrence of these two steps in humans by counting the number of times the participant's trajectory hit the target image and the target label on in-context test trials. We confirmed that participants trained on $\alpha = 0$ hit the target image more often ($84.5 \pm 5.4\%$) than those trained on $\alpha = 2$ ($43.0 \pm 9.8\%$) (effect of $\alpha$ on the probability of a hit, $\beta = -2.117 \pm 0.591$, $P = 0.0$, BF = 17.3, 'strong' evidence). Similarly, participants trained on $\alpha = 0$ hit the target label more often ($82.8 \pm 4.1\%$) than those trained on $\alpha = 2$ ($32.6 \pm 8.4\%$) (effect of $\alpha$ on the probability of a hit, $\beta = -2.433 \pm 0.582$, $P = 0.0$, BF > 100, 'decisive' evidence). The mouse-tracking data thus suggested that participants trained on a uniform distribution ($\alpha = 0$) were using a two-step process, perhaps implementing an induction head similar to transformer networks. However, one difference between humans and transformer networks is that transformers bind all items with their corresponding labels in the context, while humans only bind the target image with the target label. This is because transformers are parallel architectures, applying the same operation to all the tokens at the same time.

Finally, to test whether our findings generalize to more abstract forms of reasoning, we trained transformers on a transitive inference task. In this task, the model had to infer $A > C$ from examples such as $A > B$ and $B > C$ presented in the context. As in the main task, performance depended on the training distribution: models trained on a uniform distribution ($\alpha = 0$) solved the task using in-context learning, while models trained on a skewed distribution ($\alpha > 1$) relied on in-weights learning. These results confirm that the link between

training distribution and learning strategy holds even in tasks requiring more abstract generalization (Extended Data Fig. 9).

## Discussion

Transformers are feed-forward neural networks augmented with self-attention that process long sequences of inputs in parallel. By contrast, the brain more closely resembles a recurrent neural network, where inputs are necessarily processed over sequential time steps. A priori, there is little reason to believe that humans and transformer networks would learn in comparable ways. We were thus quite surprised to find that their sensitivity to the distributional properties of the training data was so similar. Both humans and transformer networks show the same sensitivity to increasing skewness of the training distribution, with a transition between in-weights and in-context learning occurring in both cases at $\alpha = 1$. Both humans and transformer networks traded in-weights for in-context learning when the training distribution was Zipfian, but both became double learners when trained on a composite distribution that jointly prioritized both diversity and redundancy in the training samples. Finally, both humans and transformer networks appear to use a binding-plus-searching operation to solve the task, as revealed by mechanistic interpretability analysis (in transformers) and analysis of viewing trajectories (in humans).

Previous studies using a similar methodology have argued that $\alpha = 1$ represents a 'sweet spot' at which both in-weights and in-context learning are possible in transformers. We show here that what seems to be true at the level of the population is not true at the individual model level, as no single network learned both strategies in tandem using Zipfian distributions. At $\alpha = 1$, some models converge to in-context learning and some to in-weights learning, but every model trades off one strategy for the other. We tried different model sizes and confirmed that this was also the case with larger and deeper models, with and without interleaved feed-forward layers between attention layers (up to four attention heads per layer, up to ten layers; Extended Data Figs. 10 and 11). Furthermore, we used mechanistic interpretability to confirm that attention heads were performing either in-context learning or in-weights learning but never both. To test this, we quantified the similarity between idealized attention patterns for in-context and in-weights learning and observed the attention patterns of models trained on different Zipfian distributions. The results are displayed in Extended Data Fig. 12 and show that models trained on $\alpha < 1$ are similar to in-context learning heads, while models trained on $\alpha > 1$ are similar to in-weights learning heads. Conversely, and in line with ref. 30, we show that composite, non-Zipfian distributions promote the learning of both strategies in tandem in transformers. While our results are based on relatively small transformer models trained from scratch, prior work suggests that many such behaviours generalize to larger-scale settings[32,41,45]. We nonetheless caution that scaling and pretraining introduce additional factors that may alter the dynamics of learning strategy selection.

Despite these striking similarities, transformers did not benefit from curricula that prioritized either diversity or redundancy in examples, whereas humans clearly did. This difference probably reflects a well-known limitation of neural networks: catastrophic interference. Once transformers settle on a strategy, they often forget earlier information—especially when training is blocked. In humans, early diversity boosts generalization, even when redundancy comes later. In transformer networks, later training tends to overwrite earlier strategies, making them less flexible to curriculum structure.

However, the broader failure of neural networks to benefit from structured training remains a puzzle in machine learning. For example, the BabyLM challenge (https://babylm.github.io/) is a competition in which machine learning researchers attempt to train language models with fewer than 100 million words. In its first iteration, many of the entrants attempted to use some sort of curriculum, but none were particularly successful[46]. Recent theoretical work suggests that curricula can help neural networks trained with gradient descent by guiding learning dynamics early on, especially by increasing diversity in input directions during the initial phase of training. This early diversity helps steer the model towards useful solutions more efficiently[47]. This implies that overparameterized deep neural networks (which typically already begin with a very high-dimensional initialization in weight space) are unlikely to benefit from curricula. However, this problem remains unsolved, and how to structure training examples to train neural networks more efficiently and effectively remains an open question.

Our findings have two potentially important implications for how people learn. The first is that for humans, as for transformers, a curriculum that promotes both redundancy and diversity allows people to learn strategies that rely on both memory and inference. This speaks to a long-standing debate in education research, which has asked whether schools should emphasize rote learning or critical thinking[48]. The answer implied by our data is that both are important. Presenting diverse examples that teach students how to tackle new problems is crucial, but being able to retrieve information about past experiences requires repetition. Of course, we cannot know whether insights from the simple, stylized setting employed here would translate to the classroom, but at least our work sets up a hypothesis that could be tested in more translational settings.

The second finding provides an interesting caveat to this claim: in humans, it is beneficial to provide diverse training examples early on. Early diversity does not seem to be overwritten by repetition that occurs later in training, whereas people that start learning from repeated examples never quite master the task. It is likely that early redundancy encourages learners to overfit to a specific strategy, making it more difficult to later embrace generalities. This result aligns with recent findings on asymmetries between in-context and in-weights learning. Specifically, Singh et al.[49] showed that in-context learning tends to give way to in-weights learning asymptotically, but not the reverse. Furthermore, Singh et al.[36] showed that once a model adopts an in-weights learning strategy, it struggles to recover in-context learning—while the reverse transition remains possible. We observed a similar pattern in humans: participants trained first on skewed data (favouring in-weights learning) failed to adopt in-context learning later, but those trained first on uniform data (favouring in-context learning) could shift strategies. These findings suggest that early learning conditions constrain later flexibility. We find this observation interesting, but we are unsure about its generality. It would be interesting to test whether this result replicates in other tasks involving a mixture of in-weights and in-context learning.

Our work compares humans and transformer networks. We found that in one interesting respect—the emergence of in-weights learning and in-context learning in response to the training data distribution—they show some striking similarities. However, this should not be taken to imply overlap between humans and transformers at the algorithmic level. Indeed, other classes of neural network, including simple multi-layer perceptrons, may in principle be capable of in-context learning[34,35]. Transformers are feed-forward networks with a highly structured architecture based on self-attention, diverging sharply from the recurrent, feedback-driven and biologically grounded computations of the human brain. Nevertheless, the way that they trade off memory-based strategies and inference-based strategies exhibits surprising commonalities with how this happens in human cognition.

## Methods
### Stimuli and paradigm
**Participants.** In total, we collected data from 530 participants (121 for Experiment 1, 50 for Experiment 2, 199 for Experiment 3, 40 for Experiment 4 and 120 for the replication of Experiment 1). The participants were recruited on the crowdsourcing platform Prolific (https://app.prolific.co/). The inclusion criteria included being between 18 and 40

years old, reporting no neurological condition, being an English speaker, being located in the USA or the UK, not having participated in another version of the task, having a minimal approval rate of 90% on Prolific and having a minimum of five previous submissions on Prolific. Participants received on average £10 per hour for their time and effort, including a bonus on performance (£8.5 per hour for random performances and £10.5 per hour for perfect performances). All experiments were approved by the Medical Sciences Research Ethics Committee of the University of Oxford (approval reference no. R50750/RE005). Before starting the experiment, informed consent was taken through an online form, and the participants indicated that they understood the goals of the study, how to raise any questions, how their data would be handled and that they were free to withdraw from the experiment at any time.

**Stimuli.** We selected 2,000 pictures from the Common Objects in Context dataset[50]. The pictures represented a large variety of items (animals, people, landscapes, food and objects). The images were cropped and scaled to 300 × 300 pixels.

**Procedure.** *JavaScript online experiments*. The experiments were written in JavaScript, using jsPsych (version 7.3.1, https://www.jspsych.org/7.3/)[51], and hosted on a web server. The scripts are available at https://osf.io/xb43k.

*Instructions*. The participants were instructed that the task was deterministic. The exact instructions were "This task is a learning task. You may have poor performances at the beginning but you will improve over the course of the experiment. On each trial, you will see a sequence of images and numbers. Your task is to press on the correct number on your keyboard, from 0 to 9. The rule determining which number you have to choose is 100% deterministic. This means that once you have discovered the rule, you will have 100% of correct responses."

*Main task*. On each trial, the participants were presented with an image at the centre of the screen (the query image) surrounded by seven images and seven labels arranged in a ring (the context). The participants were asked to select the correct label associated with the query image by pressing on their keyboard among ten possible labels: {'0', '1', '2', '3', '4', '5', '6', '7', '8', '9'}. Trials consisted of the following events: (1) a black loading screen for 500 ms, (2) stimulus presentation (query image and context) and response recording until a response was made, and (3) trialwise feedback for 1,000 ms. The stimuli remained visible on screen during feedback. For trials without trialwise feedback, a black screen was presented for 1,000 ms instead of the feedback screen. In training blocks, the participants received blockwise feedback on their performance in the last block on top of trialwise feedback.

Four block types were presented:

- Training blocks. The query images were sampled from a Zipfian distribution with parameter $\alpha$ (see below). A copy of the query image (the target image) was always present in the context. The location of this target image was sampled uniformly from the seven possible locations. The six other images in the context were sampled uniformly from our pool of 2,000 images. The correct label was always located three steps clockwise from the target image (the target label). The other six labels in the context also followed the same rule: each was located three steps clockwise from its corresponding context image. The three-steps-clockwise rule and the use of seven context images were chosen on the basis of pilot data to avoid trivial or symmetry-based rules that led to rapid learning. Fully informative trialwise feedback was provided during training: after each trial, the participants were shown whether their response was correct or incorrect, as well as the correct response. Blockwise feedback was also given after each block during training. The mapping between images and labels was arbitrary and not semantically meaningful.

- In-context test blocks. The query images were novel images sampled uniformly from unseen images during training. A copy of this query image (the target image) was always present in the context. The location of the target image was sampled uniformly from the seven possible locations. The six other images in the context were sampled uniformly from our pool of 2,000 images. The correct label was always located three steps clockwise from the target image (the target label). The other six labels in the context were sampled uniformly between 0 and 9. No feedback was given during in-context test blocks.

- In-weights test blocks. The query images were old images sampled from the same Zipfian distribution as the training. No target image was present in the context. The seven images in the context were sampled uniformly from our pool of 2,000 images. The seven labels in the context were sampled uniformly between 0 and 9. No feedback was given during in-weights test blocks.

- Arbitrage test blocks. The query images were old images sampled from the same Zipfian distribution as the training. A copy of the query image (the target image) was always present in the context. The location of this target image was sampled uniformly from the seven possible locations. The six other images in the context were sampled uniformly from our pool of 2,000 images. The seven labels in the context were sampled uniformly between 0 and 9. No feedback was given during arbitrage test blocks.

*Rank-frequency (Zipfian) distribution*. In training blocks, query images were sampled from a rank-frequency (Zipfian) distribution of parameter $\alpha$. A Zipfian distribution on $N$ elements assigns to the element of rank $k$ (counting from 1) the probability:

$$f(k, N, \alpha) = \frac{1}{H_{N,\alpha}} \frac{1}{k^\alpha}$$

where $H_{N,\alpha}$ is a normalization constant and is equal to the $N$th generalized harmonic number. When $\alpha = 0$, the distribution is the uniform distribution. When $\alpha > 0$, the distribution is skewed, with larger values of $\alpha$ associated with a higher degree of skewness. On 150 trials, the frequency rankings were as follows:

- For $\alpha = 0$, the distribution was uniform, and the frequency of the images was [1, 1, 1, …, 1, 1] (all images are novel and appear once).
- For $\alpha = 1$, the distribution was skewed, and the frequency of the images sorted in decreasing order was [25, 13, 9, 7, 5, 5, 4, 4, 3, 3, 3, 3, 2, 2, 2, 2, 2, 2, 2, 2, 2, 2, 2, 1, 1, …].
- For $\alpha = 2$, the distribution was skewed, and the frequency of the images sorted in decreasing order was [92, 23, 11, 6, 4, 3, 2, 2, 2, 1, 1, …].
- For $\alpha = 4$, the distribution was highly skewed, and the frequency of the images sorted in decreasing order was [139, 9, 2].

*Experiment 1 and Experiment 1 replication*. In Experiment 1, training consisted of five blocks of 30 trials (150 training trials in total). Participants were assigned randomly to one of four groups (between-participant design), corresponding to four distributions of the query images during training: a Zipfian distribution with $\alpha \in \{0, 1, 2, 4\}$. After training, the participants performed the three test blocks: one in-context test block of 30 trials, one in-weights test block (30 trials) and one arbitrage test block (30 trials). The order of the test blocks was randomized across participants.

*Experiment 2*. In Experiment 2, training consisted of four blocks of 30 trials (120 training trials in total). Query images during training were sampled from a composite distribution—that is, 60 trials with query images sampled from a uniform distribution ($\alpha = 0$) and 60 trials with query images sampled from a skewed distribution ($\alpha = 2$). The order of

all trials was shuffled for each participant, meaning both distributions were fully interleaved. After training, the participants performed the three test blocks: one in-context test block of 30 trials, one in-weights test block (30 trials) and one arbitrage test block (30 trials). Query images in the in-weights and arbitrage test blocks were sampled from the skewed distribution ($\alpha = 2$). The order of the test blocks was randomized across participants.

*Experiment 3*. In Experiment 3, training consisted of four blocks of 30 trials (120 training trials in total). Two types of training blocks were presented: training blocks with query images sampled from a uniform distribution ($\alpha = 0$) and training blocks with query images sampled from a skewed distribution ($\alpha = 2$). Participants were assigned randomly to one of four groups (between-participant design), corresponding to four training curricula: C1 (the first block is skewed, the second block is skewed, the third block is uniform and the fourth block is uniform), C2 (uniform, uniform, skewed, skewed), C3 (skewed, uniform, skewed, uniform) and C4 (uniform, skewed, uniform, skewed). After training, the participants performed the three test blocks: one in-context test block of 30 trials, one in-weights test block (30 trials) and one arbitrage test block (30 trials). Query images in the in-weights and arbitrage test blocks were sampled from the skewed distribution ($\alpha = 2$). The order of the test blocks was randomized across participants.

*Experiment 4*. In Experiment 4, training consisted of five blocks of 30 trials (150 training trials in total). Participants were assigned randomly to one of two groups (between-participant design), corresponding to two distributions of the query images during training: a Zipfian distribution with $\alpha \in \{0, 2\}$. After training, the participants performed the three test blocks: one in-context test block of 30 trials, one in-weights test block (30 trials) and one arbitrage test block (30 trials). The order of the test blocks was randomized across participants. During the test blocks, we used MouseView.js[52] to track the attention of the participants on the screen during stimulus presentation. For that, the display was blurred and obscured so that the locations of images and labels could be seen but not their content. The participants were allowed to move a sharp aperture with their mouse to reveal part of the screen. We used the default parameter values of MouseView.js, with an aperture of size 15% (roughly the size of an image on the screen).

*Preregistrations*. The replication of Experiment 1 was preregistered on AsPredicted (no. 231356, https://aspredicted.org/rqgz-rdfk.pdf). Experiment 3 was also preregistered on AsPredicted (no. 173550, https://aspredicted.org/yhvp-6y3y.pdf). All hypotheses and planned analyses are publicly available in the corresponding preregistration documents.

### Neural networks
Our model was largely based on the work of Reddy[30], which investigated the mechanistic basis of in-context learning in transformers.

**Stimuli.** The network was trained to predict the label 'label$_q$' of a query item 'item$_q$' given an alternating sequence of $N$ images and $N$ labels:

item$_1$; label$_1$; item$_2$; label$_2$; … ; item$_N$; label$_N$; item$_q$;?

The images and labels were embedded in $D + P$ dimensions. The first $D$ dimensions encoded content, while the latter $P$ dimensions encoded positional information. Position was encoded by a one-hot $P$-dimensional vector. Images were $D$-dimensional vectors sampled independent and identically from a $D$-dimensional Gaussian distribution with mean 0 and variance 1. Each of the $K$ images was assigned one of the $L$ labels ($L \le K$). Labels were drawn prior to training and were also sampled independent and identically from a $D$-dimensional Gaussian distribution with mean 0 and variance 1.

**Architecture.** The inputs were passed through a two-layer attention-only network of intrinsic dimensionality $D_M$ followed by a classifier. Each attention layer had one attention head with a causal mask. The classifier was composed of two fully connected layers with ReLU activations and $D_M$ hidden units each. The last layer was a fully connected layer that predicted the probabilities of the $L$ labels.

We also tested an interleaved MLP model (Extended Data Fig. 11), where each attention layer was followed by a feed-forward (MLP) block consisting of two dense layers with $D_M$ units (with ReLU activation), a residual connection and a layer normalization step.

Mimicking our human experiment, the dimensions of the problem were set to $L = 10$ and $N = 7$. The dimensions of the inputs were set to $K = 2^{14}$ and $D = 8$. The dimension of the model was set to $D_M = 16$.

**Training.** The network was trained using a cross-entropy loss. For training, we used a batch size of 128 and the Adam optimizer with a learning rate of 0.01. The models were trained on 5,000 steps.

**Alternative models.** We compared the performance of the transformer network with two other architectures, keeping the number of layers and total number of parameters fixed: a two-layer feed-forward fully connected network with ReLU activations, and a two-layer LSTM network. All models were trained on the same data and evaluated using the same procedure as the transformer, including positional encodings in their input representations.

The feed-forward model received the entire input sequence flattened into a single vector. The standard LSTM received inputs one item at a time, with the query presented last, matching the set-up used for transformers and human participants. We also tested a query-first LSTM variant, where the query appeared at the start of the sequence, followed by the context items. This was designed to test whether knowing the target query early would help the model focus on relevant context and learn an in-context strategy. Despite these variations, none of the models showed reliable in-context learning (Extended Data Fig. 2).

**Transitive inference task.** We designed a second modelling task to test whether the effects of training distribution on learning strategy generalize beyond the image–label association setup. In this task, each training environment consisted of six unique images, each with an implicit rank. The model received ten training triplets per trial, each expressing a one-step comparison between images (for example, 'image 4 > image 3'), followed by a query that required a two-step transitive inference (for example, 'image 4 ? image 2').

On each trial, the model received:

- A context of one-step comparisons between image pairs from a single environment (for example, 'image 4 > image 3', 'image 3 > image 2').
- A query requiring a two-step inference (for example, 'image 4 ? image 2'), where the model had to choose the correct relational symbol ('>' or '<').

We manipulated the training distribution by varying the skewness (Zipf exponent $\alpha$) of how often each environment appeared during training, following the same logic as in our main task. At test, three types of blocks were used:

- In-context test: the context came from a novel environment, so the only way to respond correctly was to use in-context learning.
- In-weights test: the query pair had been seen during training, but the context came from a novel environment; accuracy relied on memorized pair–label associations.
- Arbitrage test: environments were reused from training but with reversed item orders (for example, 'image 4 < image 3'), to probe which strategy dominated when in-context and in-weights learning gave conflicting answers.

We used the same architecture, training set-up and evaluation metrics as in the image–label association task. The full results are presented in Extended Data Fig. 9.

## Statistical analysis

**Outliers.** No outliers were removed from the analyses.

**Model selection.** Statistical analyses were done using R version 4.4.2 (ref. 53) and the package lme4 (ref. 54). For all analyses, model complexity was monitored using the Bayesian information criterion (BIC), a standard measure to arbitrate between complexity and accuracy. The reported P values are Satterthwaite approximations. We also report the BF for each effect as approximated using the difference between the BIC of the model with the effect $BIC_1$ and the model without the effect $BIC_0$ and defined as $BF = \exp((BIC_0 - BIC_1)/2)$. The BF quantifies the support of the data in favour of an effect. We followed ref. 55 for the interpretation of its values: BF > 3, BF > 10 and BF > 100 were respectively taken as substantial, strong and decisive evidence in favour of an effect (BF < 0.3, BF < 0.1 and BF < 0.01 as evidence in favour of the absence of an effect).

**Accuracy.** In Experiment 1 and Experiment 4, the probability of being correct (0, incorrect; 1, correct) was modelled as an independent logistic regression for each block type, with $\alpha$ as a fixed effect and one random intercept per participant.

In Experiment 3, the probability of being correct was modelled as an independent logistic regression for each block type and each group contrast, with the group as a fixed effect and one random intercept per participant. We applied a Bonferroni correction to correct for multiple comparisons.

**Power analysis.** The sample size for the replication of Experiment 1 was determined via power simulations based on data from Experiment 1, assuming a 50% smaller effect size than observed in that study. The simulations suggested a minimum of 10–20 participants per group, depending on the block. To ensure robust power across all analyses, we conservatively set the sample size to 30 per group.

**Double learning index.** We defined a double learning index as a value between 0 (no double learning) and 1 (perfect performance in both in-context and in-weights test trials). For each participant, it was defined as:

$$D = \text{scale}(m_{IC}) \times \text{scale}(m_{IW})$$

$$\text{scale}(m) = \frac{m - \text{chance}}{1 - \text{chance}}$$

where $m_{IC}$ is the average performance of the participant in the in-context test trials, $m_{IW}$ is the average performance of the participant in the in-weights test trials and 'scale' is a linear mapping accounting for chance level ('chance', here 10%). Because it is a product, this index is 0 if either of the two performances is at chance (and thus non-zero only if both performances are above chance).

In Experiment 2 and Experiment 3, the double learning index was modelled as a linear regression with the group as a fixed effect.

**Mouse trajectories.** For visualization purposes, trial-by-trial cursor trajectories were first rotated in a common frame where the target image was located on the top of the context circle and then resampled to 100 time points between the start and the end of the trial using linear interpolation. A 'hit' trial was defined as the target image being at a minimum distance of 20% of the screen height at least one time during the trial. In Experiment 4, the probability of a hit (0, no hit; 1, hit) was modelled as a logistic regression for in-context test trials, with $\alpha$ as a fixed effect and one random intercept per participant.

**Reporting summary**

Further information on research design is available in the Nature Portfolio Reporting Summary linked to this article.

## Data availability

The anonymized data, materials and preregistration documents are all available via OSF at https://osf.io/xb43k.

## Code availability

The scripts for stimulus presentation and data analysis are available via OSF at https://osf.io/xb43k.

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

## Acknowledgements
We thank J. Drevet for enhancing the clarity and aesthetics of the figures. This work was supported by the Fondation Pour l'Audition RD-2021-2 (J.P.L.); the Institute for Language, Communication, and the Brain (J.P.L.); European Research Council Consolidator Grant No. 725937—CQR01290.CQ001 (C.S.); and an ATRAE award from the Spanish Ministry of Education to C.S. The funders had no role in study design, data collection and analysis, decision to publish or preparation of the manuscript.

## Author contributions
Conceptualization: J.P.L. and C.S. Data curation: J.P.L. Formal analysis: J.P.L. Funding acquisition: J.P.L. and C.S. Investigation: J.P.L. Methodology: J.P.L. and C.S. Project administration: C.S. Resources: C.S. Supervision: C.S. Visualization: J.P.L. Writing—original draft: J.P.L. and C.S. Writing—review and editing: J.P.L. and C.S.

## Competing interests
The authors declare no competing interests.

## Additional information
**Extended data** is available for this paper at https://doi.org/10.1038/s41562-025-02359-3.

**Correspondence and requests for materials** should be addressed to Jacques Pesnot Lerousseau or Christopher Summerfield.

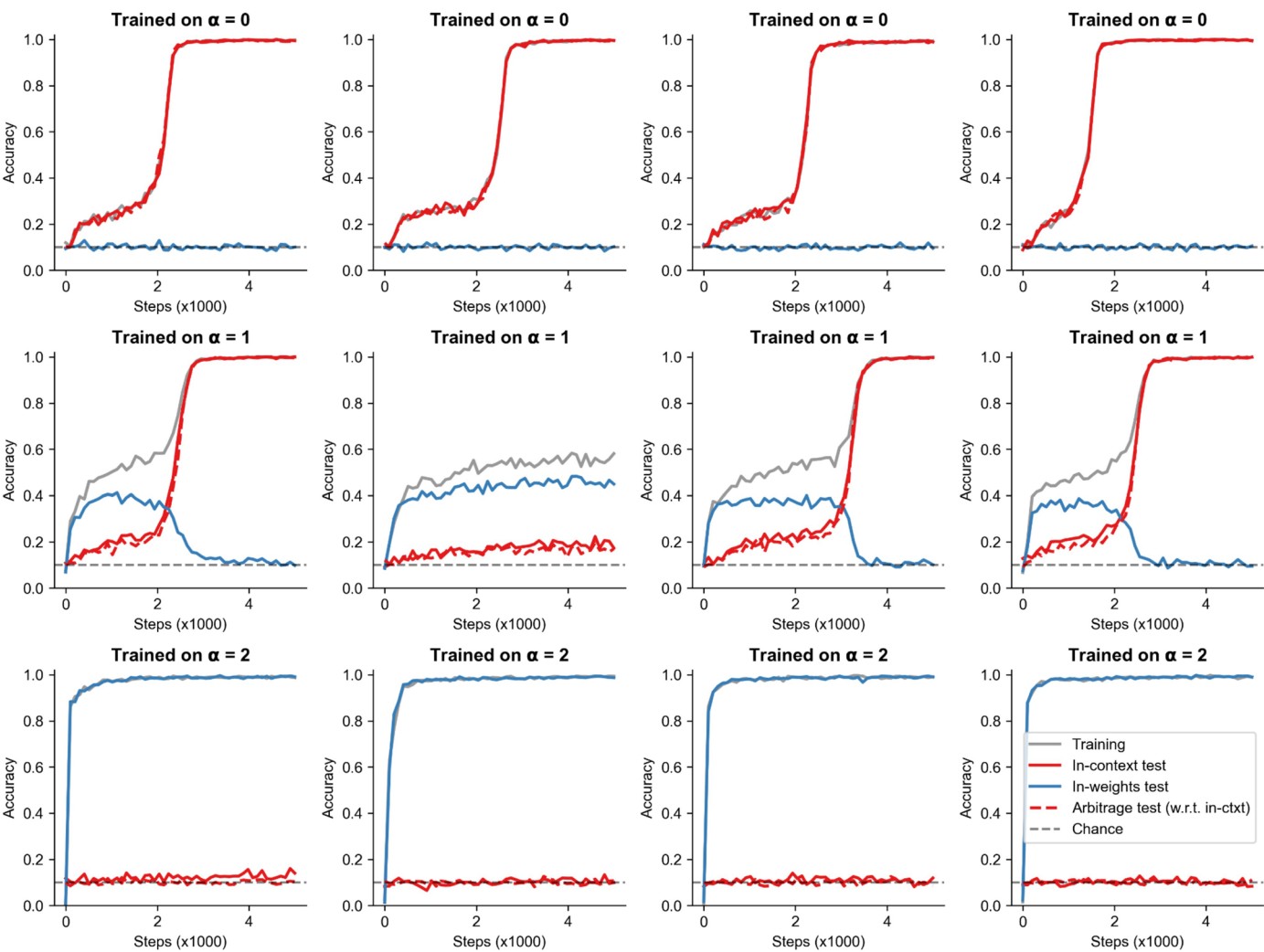

**Extended Data Fig. 1 | Example learning curves for multiple transformer networks.** Accuracy curves for multiple example transformer networks trained on different training distributions, uniform ($\alpha = 0$, top row), moderately skewed ($\alpha = 1$, middle row) and skewed ($\alpha = 2$, bottom row). In-context test performance and arbitrage test performance (with respect to in-context learning) strongly overlap. Over the course of training, in-context test performance trade-off with in-weights test performance.

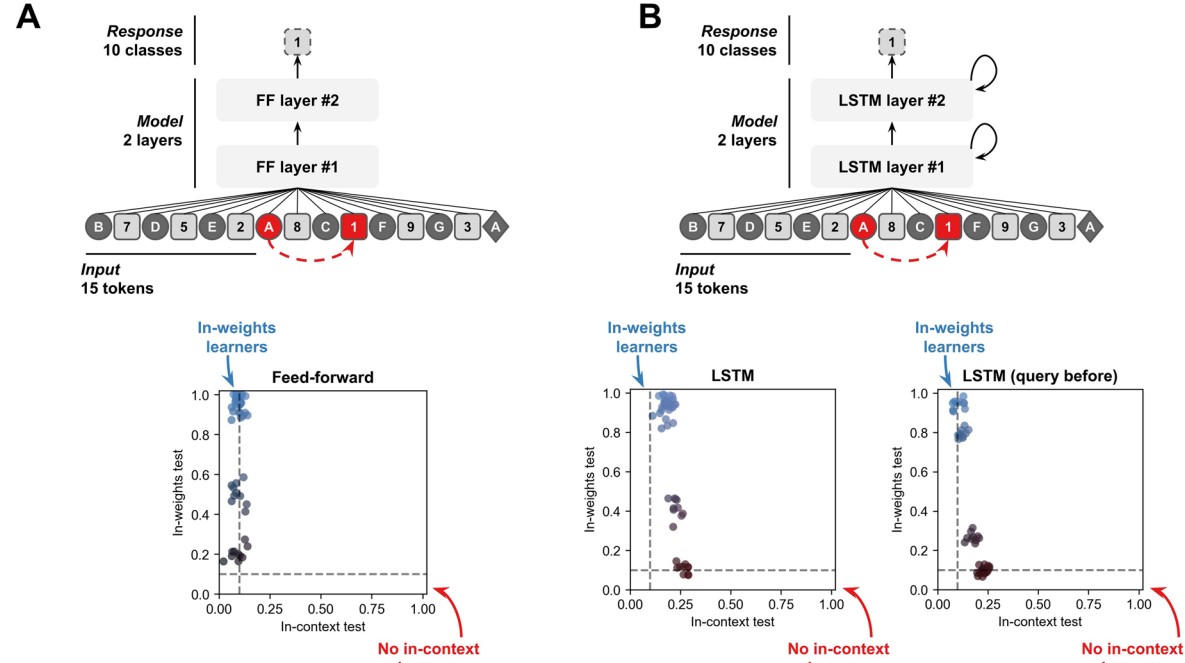

**Extended Data Fig. 2 | Feed-forward networks and LSTM networks do not become in-context learners in the same task. a.** (top) 2-layer feed-forward fully-connected network. (bottom) Scatter plots of the in-context vs in-weights test performances after training. **b.** (top) 2-layer LSTM network. (bottom) Scatter plots of the in-context vs in-weights test performances after training. Each dot is an individual network (N = 30 per training data distribution for each architecture).

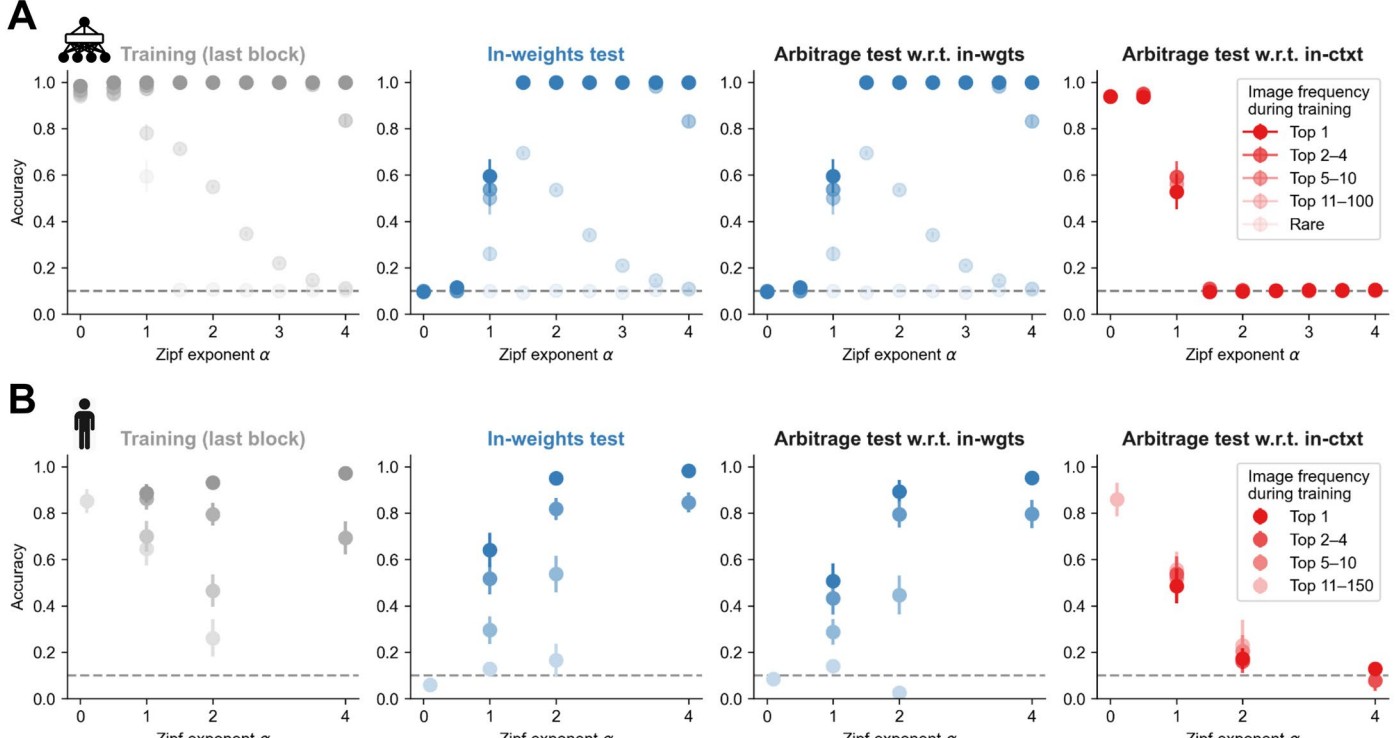

**Extended Data Fig. 3 | Performance as a function of the image frequency during training. a,b,** Training and test performances for transformers (top, N = 30 per training data distribution) and human participants (bottom, Exp. 1, N = 30 per training data distribution) as a function of the frequency of the image during training. For each value of $\alpha$, test items were grouped by how often they appeared during training. For example, in $\alpha$ = 2: 'top 1' corresponds to the image that was seen 92 times during training, 'top 2–4' to images that were seen ~13 times, and 'top 5–10' to images that were seen ~2 times. Large dots are group average. Errors are s.e.m.

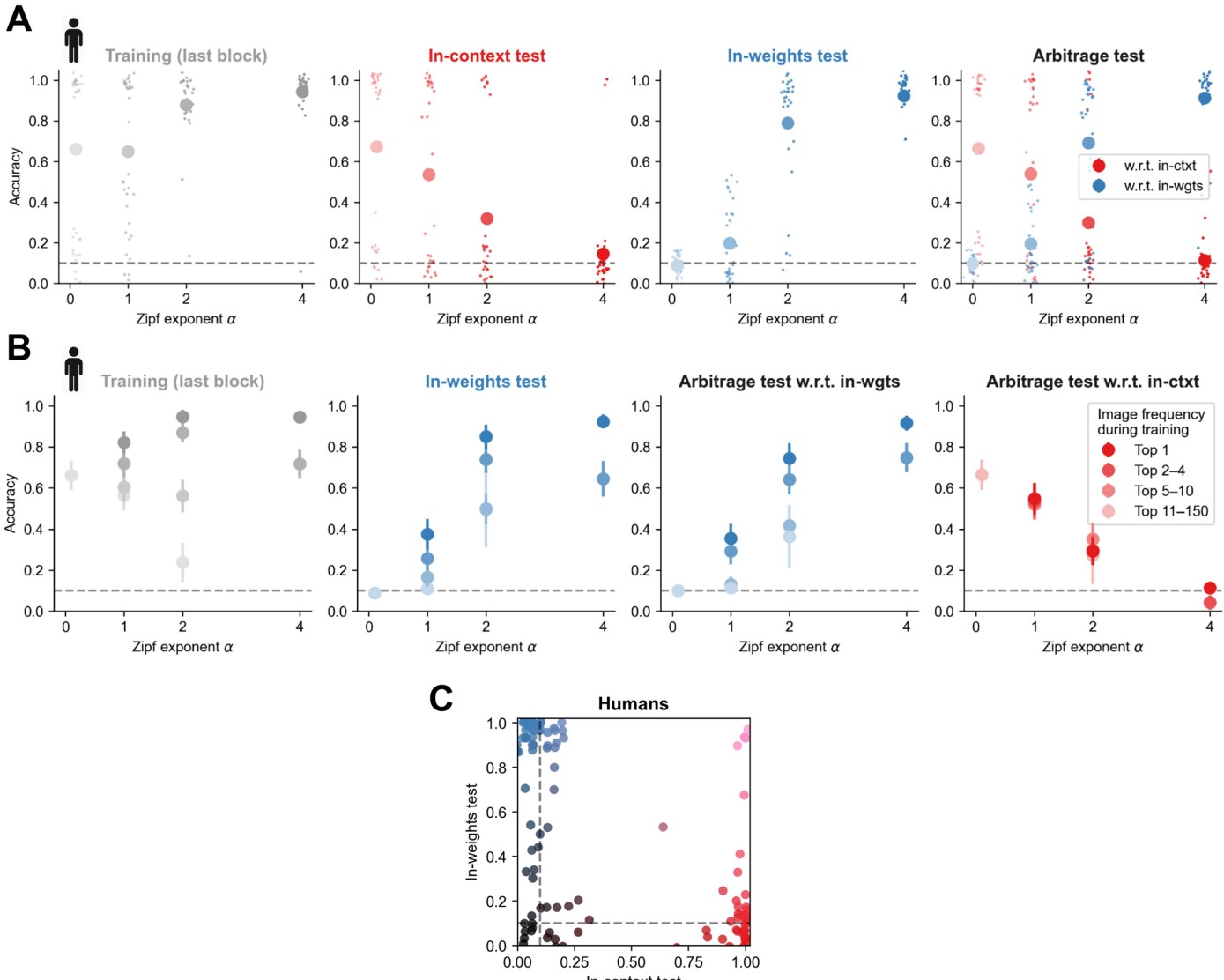

**Extended Data Fig. 4 | Replication of Experiment 1. a.** Training and test performances of human participants (bottom, replication of Exp. 1, N = 30 per training data distribution). Small dots are individuals, large dots are group average. Our pre-registered effects (AsPredicted #231356, https://aspredicted. org/rqgz-rdfk.pdf) were all verified. In particular, there was a negative effect of $\alpha$ on accuracy in in-context test block ($\beta = -1.145 \pm 0.208$, p = 0.0, BF > 100, 'decisive' evidence), a positive effect of $\alpha$ on accuracy in in-weights test block ($\beta = 1.786 \pm$ 0.138, p = 0.0, BF > 100, 'decisive' evidence), a negative effect of $\alpha$ on accuracy with respect to in-context learning in arbitrage blocks ($\beta = -1.097 \pm 0.168$, p = 0.0, BF > 100, 'decisive' evidence), and a positive effect of $\alpha$ on accuracy with respect to in-weights learning in arbitrage blocks ($\beta = 1.669 \pm 0.128$, p = 0.0, BF > 100, 'decisive' evidence). **b.** Training and test performances as a function of the frequency of the image during training. **c.** Scatter plots of the in-context vs in-weights test performances. Each dot is an individual model/human.

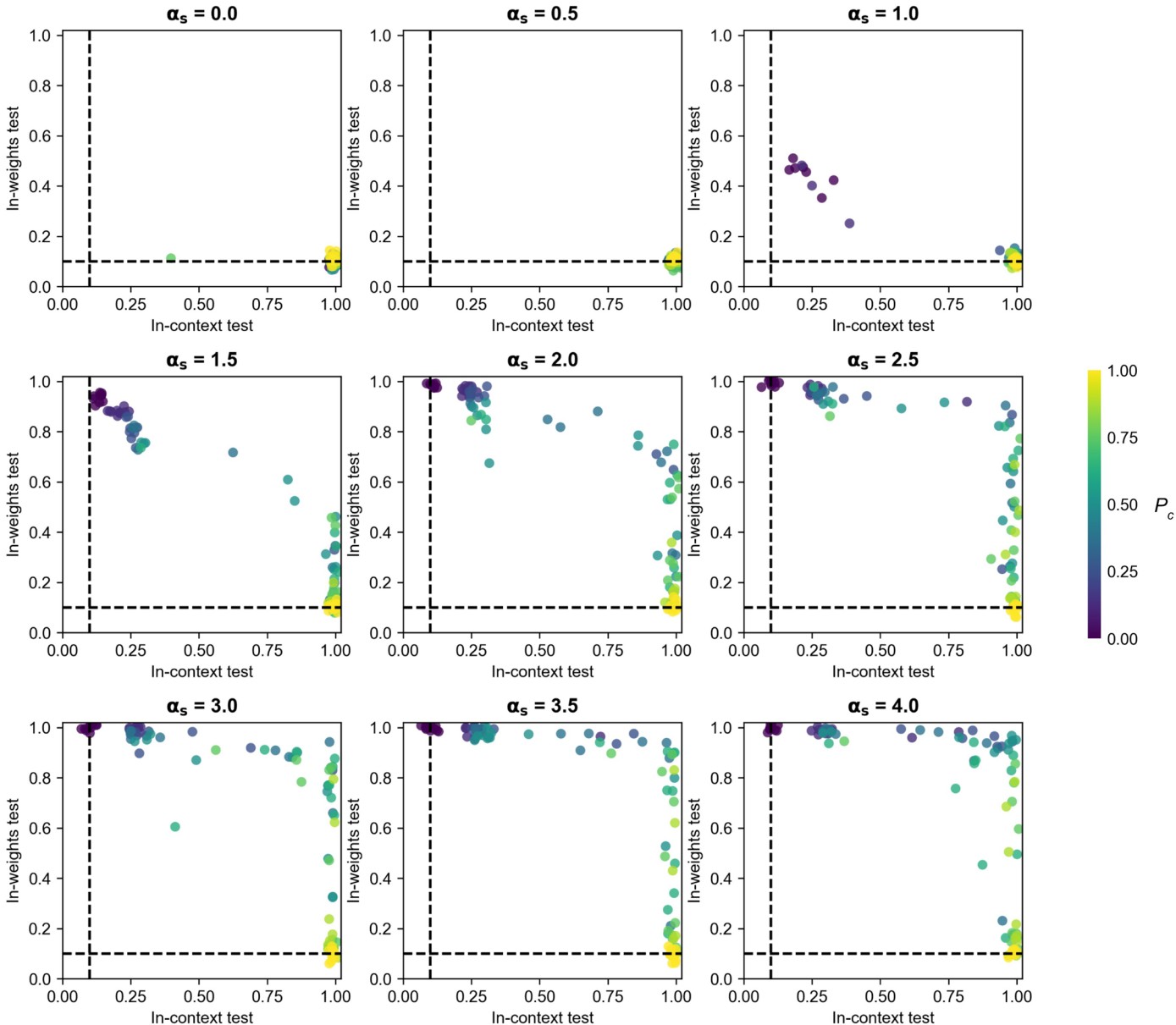

**Extended Data Fig. 5 | Performance of transformers trained on a wide range of composite distributions.** Scatter plots of the in-context vs in-weights test performances of transformers after training on different values of Pc (proportion of in-context trials during training) and αs (the rest of the trials). Dots are individual models.

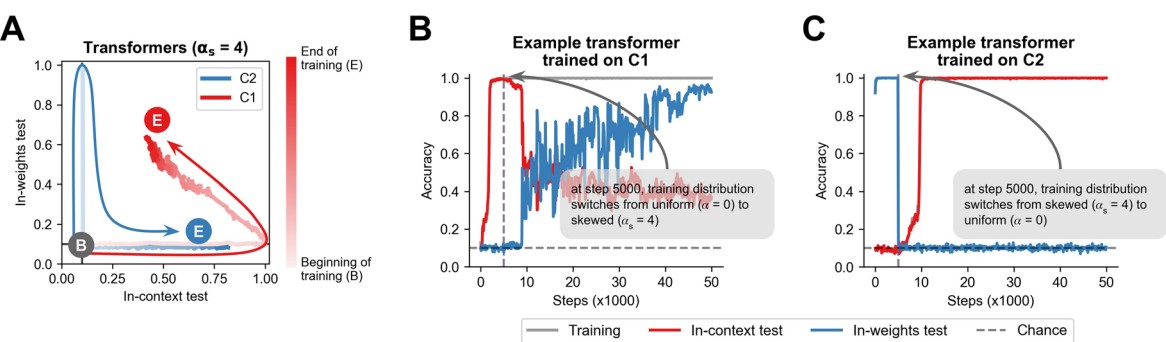

**Extended Data Fig. 6 | Transformers do not benefit from structured curricula.** **a**. Test performances over the course of training of transformers trained on C1 (red) and C2 (blue). Bold lines are group average (N = 20 transformers per curriculum). Arrows were manually added to emphasise the direction of the trajectories. **b**. Accuracy curves for one example transformer network trained on C1. **c**. Accuracy curves for one example transformer network trained on C2.

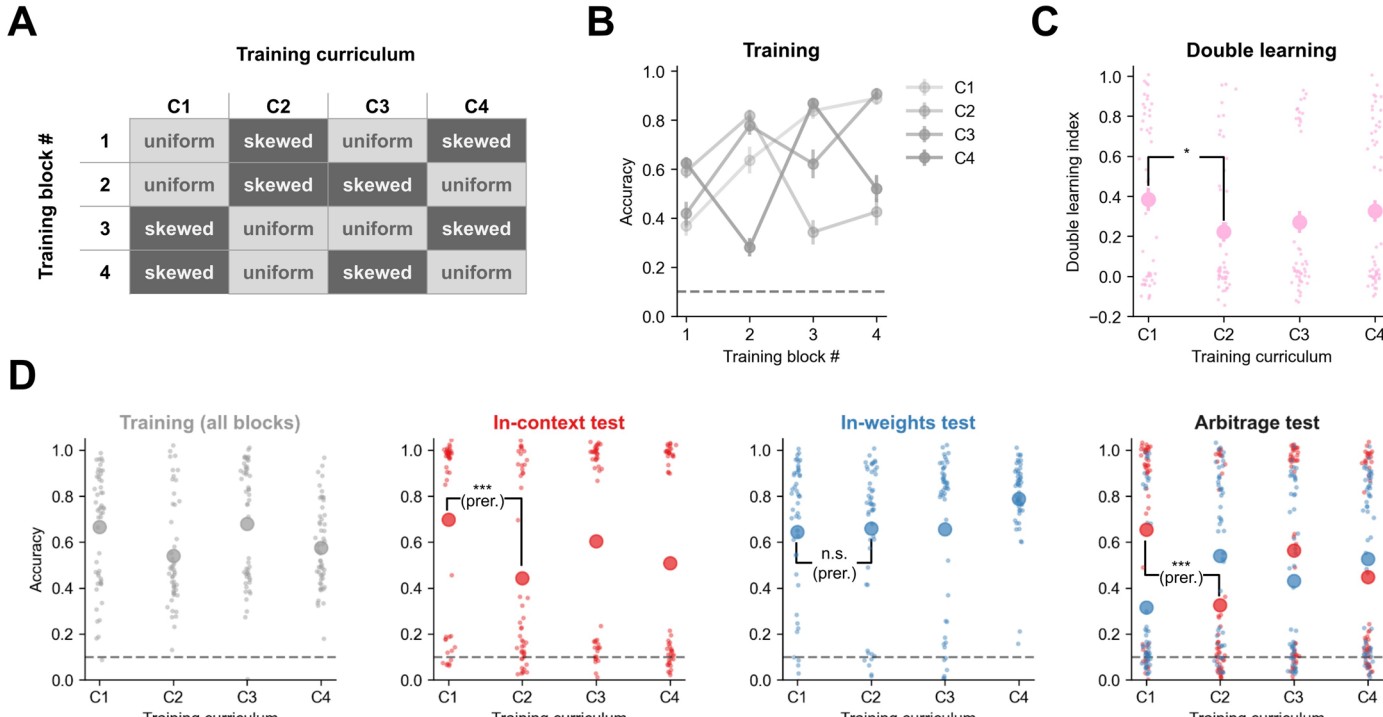

**Extended Data Fig. 7 | Performance of human participants in all curricula.**
**a**. Four groups of human participants (Exp. 3, N = 50 per group) were exposed
to a composite distribution (Pc = 0.5, $\alpha$s = 2) with different training curricula,
that is different block order, denoted C1 to C4 ('uniform', $\alpha$ = 0; 'skewed', $\alpha$s = 2).
**b**. Performance during training per curriculum. **c**. Double learning index per

curriculum. n.s. p > 0.05, * p < 0.05, ** p < 0.01, *** p< 0.001. **d**. Training and test
performances for humans per curriculum. A curriculum that promotes learning
first in-context and then in-weights improves the in-context performance
without impairing in-weights learning. Small dots are individuals, large dots are
group average. n.s. p > 0.05, * p < 0.05, ** p < 0.01, *** p< 0.001.

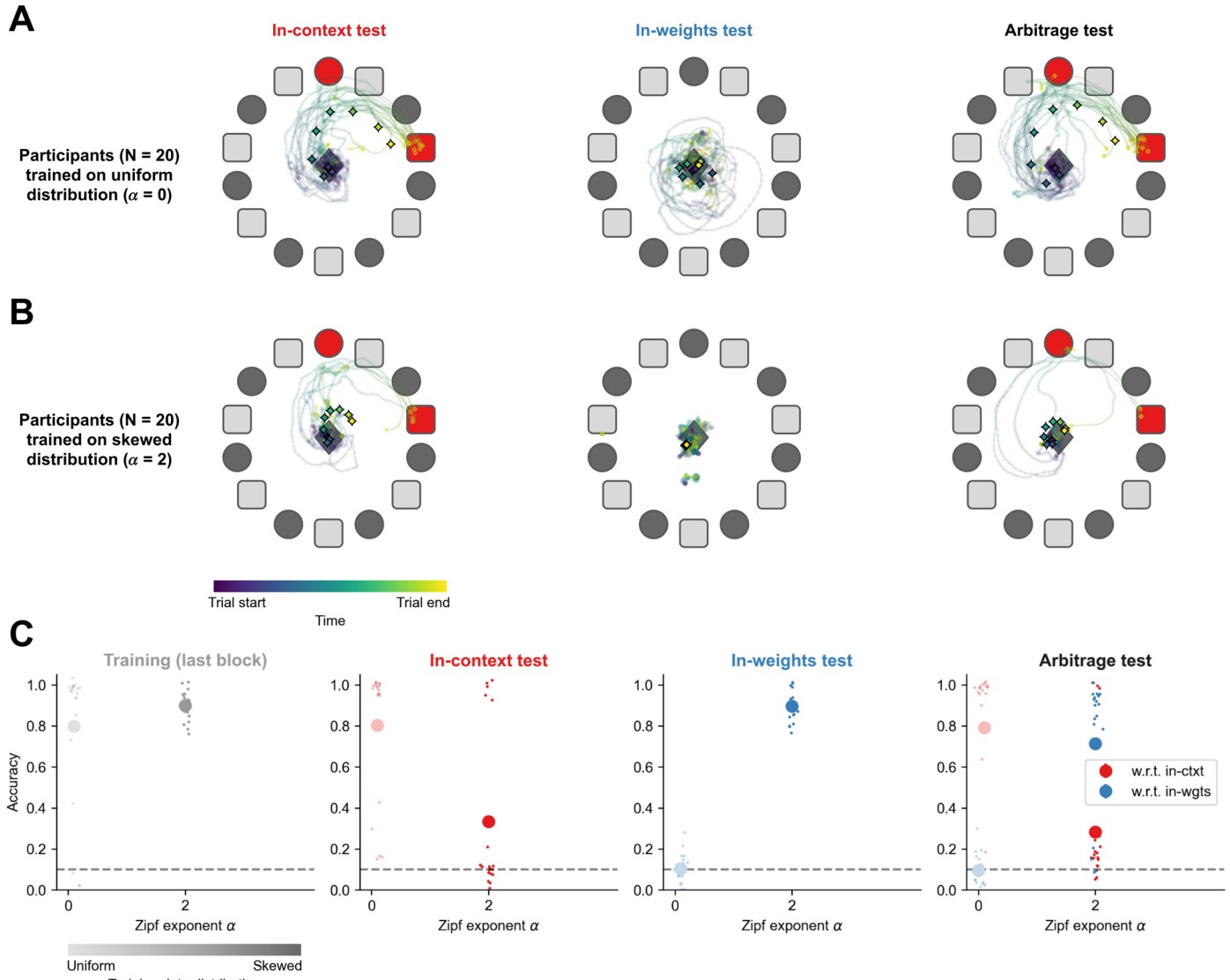

**Extended Data Fig. 8 | Cursor trajectories and performances of participants in all test blocks (Exp. 4, N = 20 per group). a**. Trajectories for participants trained on a uniform distribution ($\alpha = 0$) in the (left) in-context test block, (middle) in-weights test block and (right) arbitrage block. **b**. Same for participants trained on a skewed distribution ($\alpha = 2$). Trajectories were aligned trial-by-trial to a common frame where the target image is located on the top of the context circle. Small lines are individual average trajectories, diamonds are group average trajectories. **c**. Training and test performances. Small dots are individual, large dots are group average.

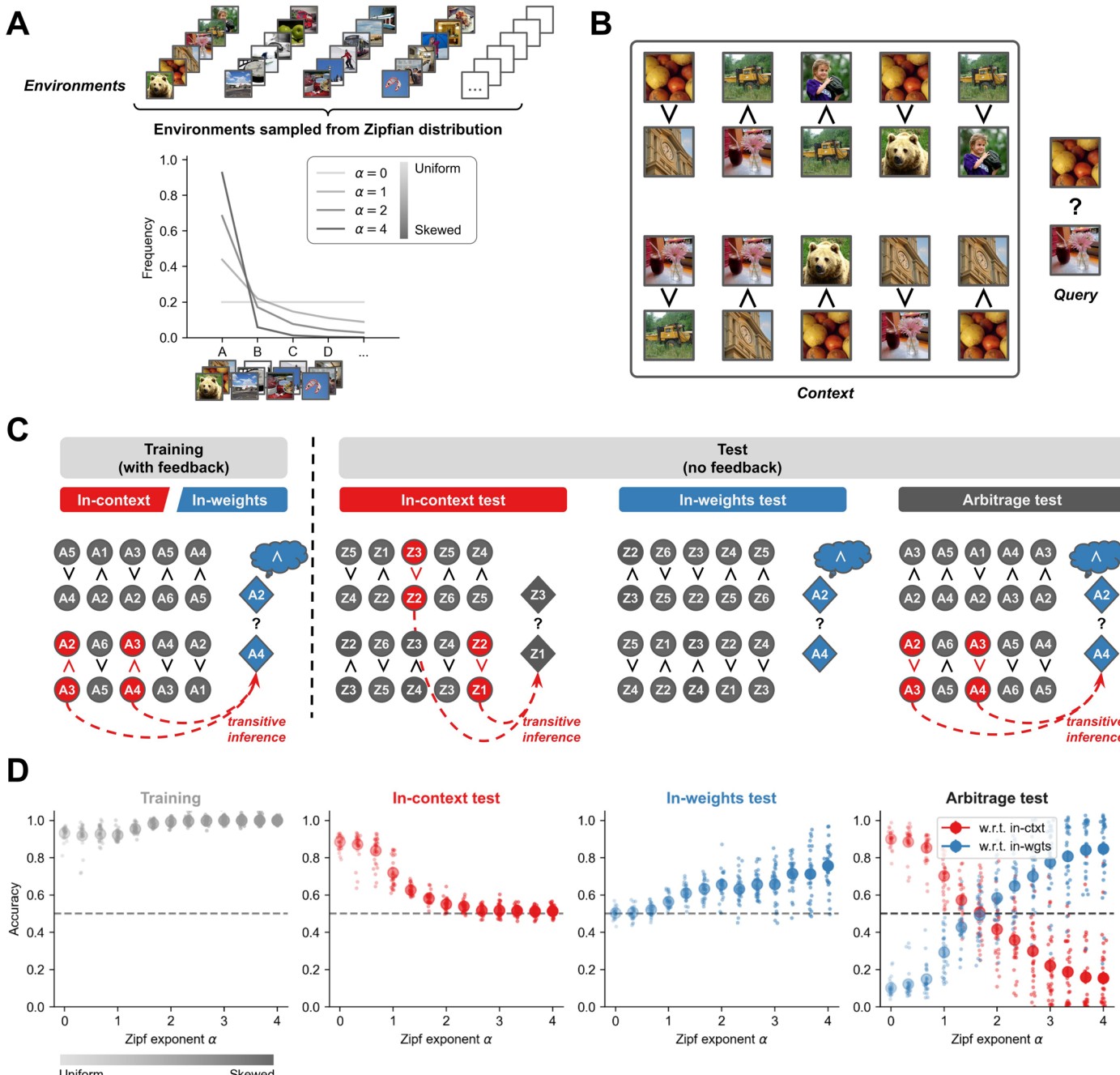

**Extended Data Fig. 9 | Modelling results in a transitive inference task. a**. We replicated our modelling results in a distinct task probing transitive inference. As in the image-label association task (Fig. 1), we manipulated the distribution of the training data: under a uniform distribution ($\alpha = 0$), all environments are equally likely; under a skewed distribution ($\alpha \gg 0$), some environments are more frequent. Each environment consisted of six images ordered along an underlying dimension. **b**. Example training trial. The context presented ten triplets, each comprising two images and a symbol, corresponding to all one-step comparisons within a given environment (for example, 'image 4 > image 3'). The query consisted of a two-step comparison (for example, 'image 4 ? image 2'), and the model had to select the correct relational symbol ('>' or '<'). **c**. Paradigm overview. During training, two learning strategies are available. The 'in-context' learning strategy consists in using local comparison given in the context to infer the correct relational symbol via transitive inference (for example, relying

on 'image 4 > image 3' and 'image 3 > image 2' to infer 'image 4 > image 2'). The 'in-weighs' learning strategy consists in learning the association between pairs of images and relational symbols in memory using the feedback. Test blocks were designed to probe which strategy(ies) the model is using. On in-context test blocks, images from novel environments (depicted in grey) were presented, such that the only way to be accurate is to use information from the context, a.k.a. the in-context strategy. On in-weights test blocks, a training pair (depicted in blue) was presented as the query pair but images from novel environments (depicted in grey) were presented in the context, such that the only to be accurate is to use information stored in memory, a.k.a. the in-weights strategy. On 'arbitrage' test blocks, a trained environment was presented but the order of the images was reversed (for example, 'image 4 < image 3'). **d**. Training and test performances for transformers (N = 30 per training data distribution). Small dots are individual transformers, large dots are group average.

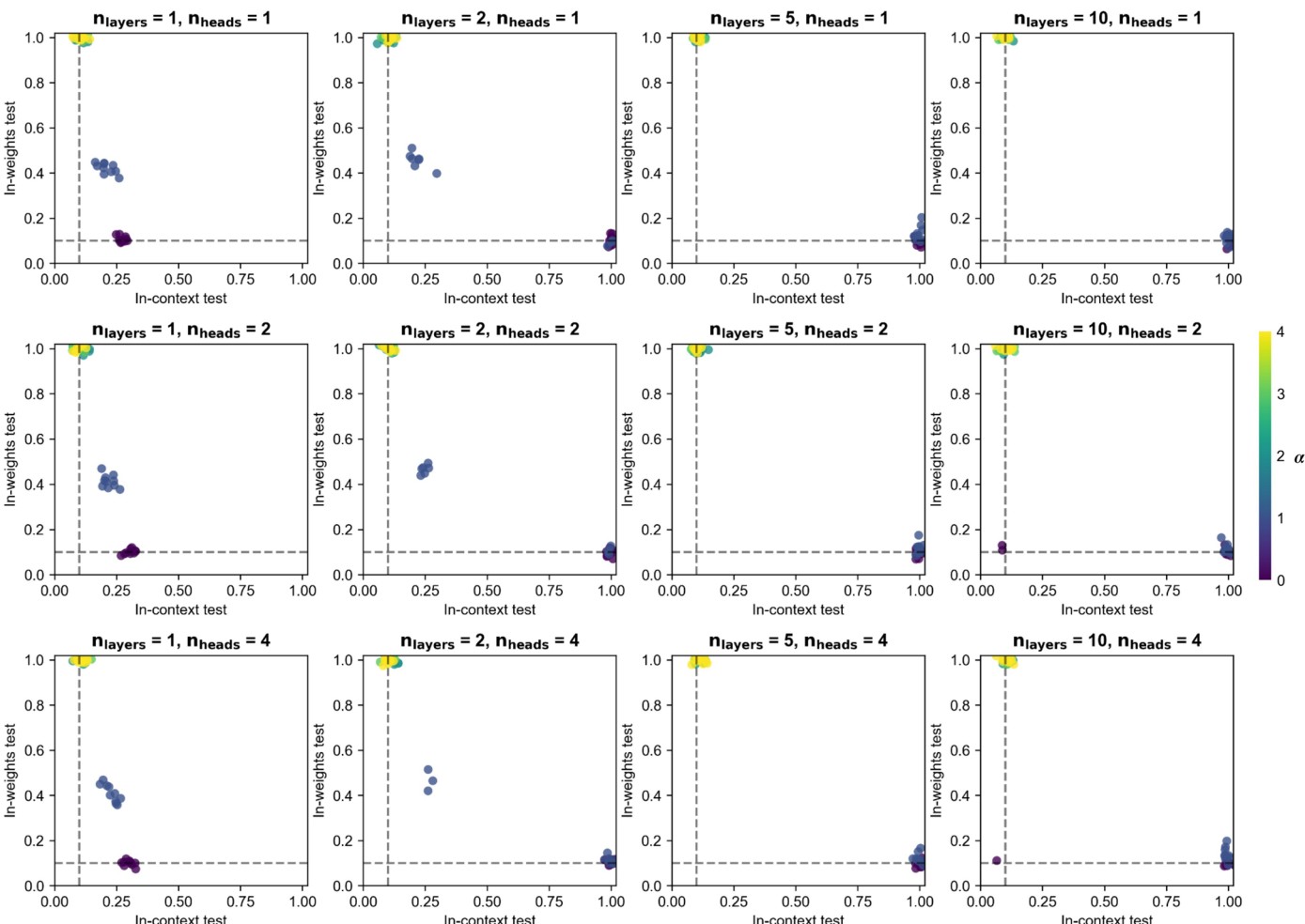

**Extended Data Fig. 10 | Performance of transformers with varying architecture sizes.** Scatter plots of the in-context vs in-weights test performances for transformers with varying numbers of layers, number of heads per layers, and varying training distributions. Each dot represents a model trained with a specific number of layers, attention heads, and training data distribution. Dot color indicates the $\alpha$ exponent of the training distribution. Dotted lines indicate chance-level performance.

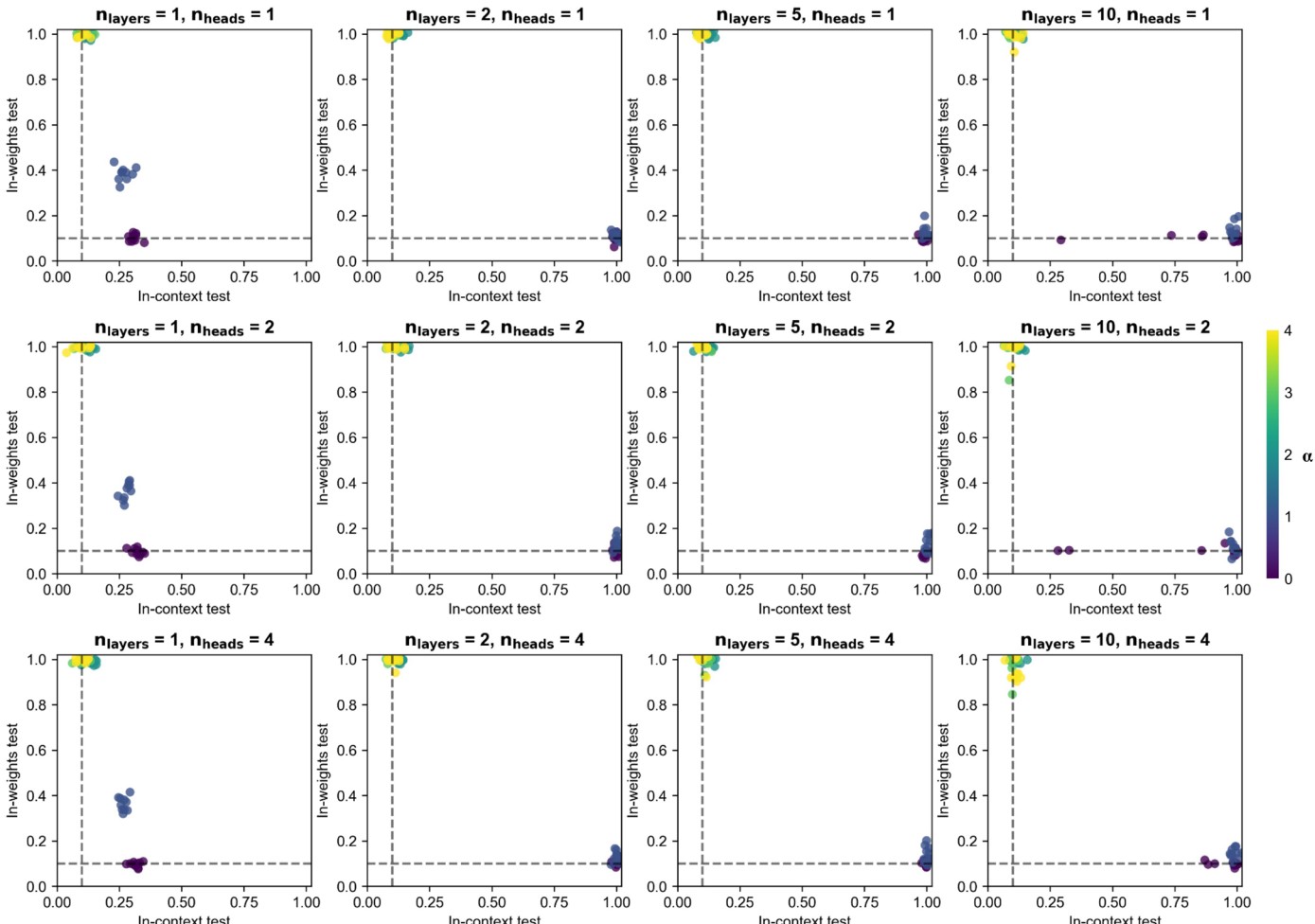

**Extended Data Fig. 11 | Performance of transformers with interleaved MLP with varying architecture sizes.** The MLP blocks consist of two dense layers with a ReLU activation, followed by a residual connection and layer normalization. Scatter plots of the in-context vs in-weights test performances for transformers with varying numbers of layers, number of heads per layers, and varying training distributions. Each dot represents a model trained with a specific number of layers, attention heads, and training data distribution. Dot color indicates the $\alpha$ exponent of the training distribution. Dotted lines indicate chance-level performance.

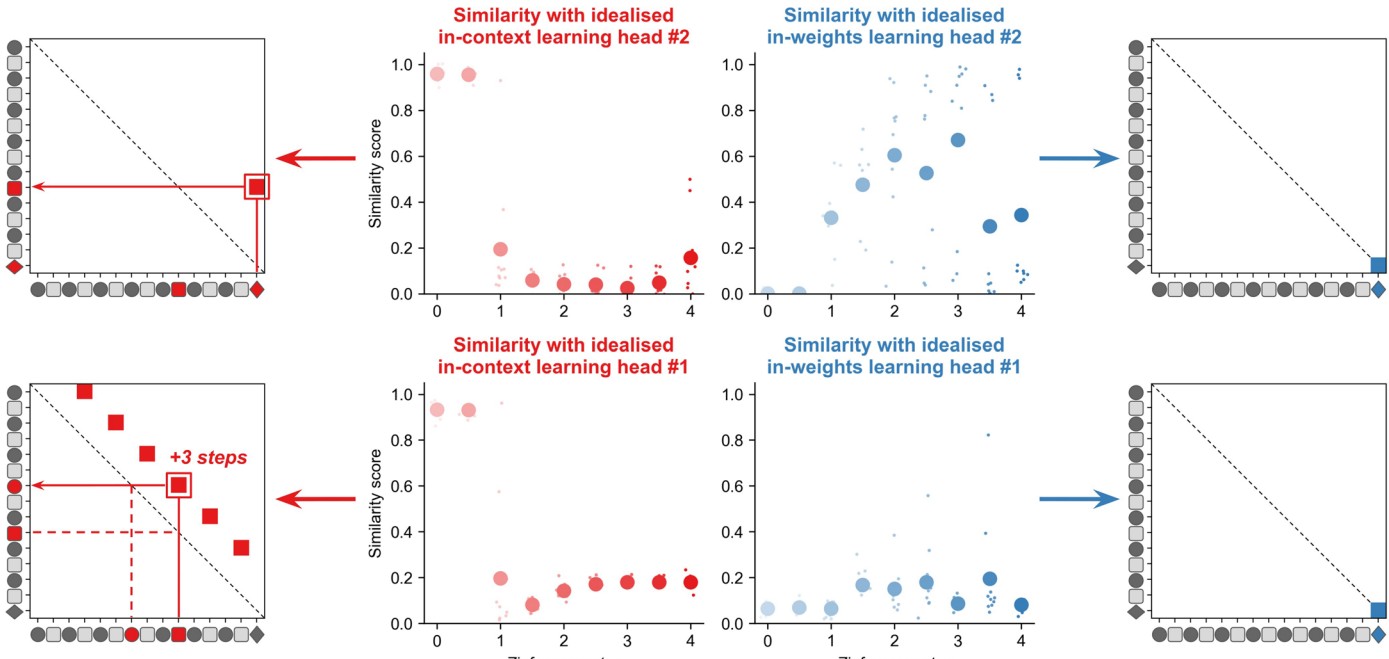

**Extended Data Fig. 12 | Similarity score with respect to idealised attention patterns.** (left) Similarity score between observed attention patterns (N = 10 transformers per training distribution) and idealised attention patterns performing in-context learning. (right) Same with idealised attention patterns performing in-weights learning. The similarity score was a dot product normalised by the ℓ1-norm of the idealised head. Models trained on $\alpha < 1$ were similar to in-context learning heads while models trained on $\alpha > 1$ were similar to in-weights learning. Results were less clear for in-weights learning head #1 because these heads tended to have more diverse patterns (attention spread to all tokens, or restricted to some tokens, and most of the time restricted to the last token).

**Extended Data Table 1 | Pairwise comparisons between curricula (Experiment 3)**

| | $\beta$ | SE | p | BF |
|---|---|---|---|---|
| **In-context test blocks** | | | | |
| C1 vs C2 | -2.635 | 0.784 | 0.001 | 5.348 |
| C1 vs C3 | -1.261 | 0.95 | 0.185 | 0.045 |
| C1 vs C4 | -1.953 | 0.917 | 0.033 | 0.176 |
| C2 vs C3 | 1.495 | 0.73 | 0.041 | 0.138 |
| C2 vs C4 | 0.796 | 0.713 | 0.264 | 0.034 |
| C3 vs C4 | -0.691 | 0.824 | 0.402 | 0.024 |
| | | | | |
| **In-weights test blocks** | | | | |
| C1 vs C2 | 0.096 | 0.428 | 0.822 | 0.019 |
| C1 vs C3 | 0.06 | 0.455 | 0.895 | 0.019 |
| C1 vs C4 | 0.793 | 0.342 | 0.02 | 0.244 |
| C2 vs C3 | -0.036 | 0.441 | 0.935 | 0.019 |
| C2 vs C4 | 0.702 | 0.329 | 0.033 | 0.164 |
| C3 vs C4 | 0.731 | 0.358 | 0.041 | 0.137 |
| | | | | |
| **Aribtrage test blocks (w.r.t. in-context)** | | | | |
| C1 vs C2 | -2.676 | 0.71 | 0 | 12.133 |
| C1 vs C3 | -0.559 | 0.77 | 0.468 | 0.023 |
| C1 vs C4 | -1.563 | 0.704 | 0.026 | 0.188 |
| C2 vs C3 | 2.122 | 0.712 | 0.003 | 1.221 |
| C2 vs C4 | 1.093 | 0.65 | 0.093 | 0.072 |
| C3 vs C4 | -1.007 | 0.705 | 0.153 | 0.049 |
| | | | | |
| **Aribtrage test blocks (w.r.t. in-weights)** | | | | |
| C1 vs C2 | 1.397 | 0.475 | 0.003 | 1.129 |
| C1 vs C3 | 0.675 | 0.441 | 0.126 | 0.058 |
| C1 vs C4 | 1.185 | 0.432 | 0.006 | 0.658 |
| C2 vs C3 | -0.715 | 0.461 | 0.121 | 0.06 |
| C2 vs C4 | -0.203 | 0.451 | 0.652 | 0.02 |
| C3 vs C4 | 0.51 | 0.42 | 0.224 | 0.038 |

Complete results of the pairwise comparisons between curricula (Experiment 3). Each row reports the effect size (β), standard error (SE), uncorrected p-value (p), and Bayes Factor (BF). Bonferroni-corrected p-values are used in the main text to ensure a consistent and conservative analysis.

# Reporting Summary

## Statistics

For all statistical analyses, confirm that the following items are present in the figure legend, table legend, main text, or Methods section.

| n/a | Confirmed | |
|---|---|---|
| ☐ | ☒ | The exact sample size (*n*) for each experimental group/condition, given as a discrete number and unit of measurement |
| ☐ | ☒ | A statement on whether measurements were taken from distinct samples or whether the same sample was measured repeatedly |
| ☐ | ☒ | The statistical test(s) used AND whether they are one- or two-sided<br>*Only common tests should be described solely by name; describe more complex techniques in the Methods section.* |
| ☐ | ☒ | A description of all covariates tested |
| ☐ | ☒ | A description of any assumptions or corrections, such as tests of normality and adjustment for multiple comparisons |
| ☐ | ☒ | A full description of the statistical parameters including central tendency (e.g. means) or other basic estimates (e.g. regression coefficient) AND variation (e.g. standard deviation) or associated estimates of uncertainty (e.g. confidence intervals) |
| ☐ | ☒ | For null hypothesis testing, the test statistic (e.g. *F*, *t*, *r*) with confidence intervals, effect sizes, degrees of freedom and *P* value noted<br>*Give P values as exact values whenever suitable.* |
| ☐ | ☒ | For Bayesian analysis, information on the choice of priors and Markov chain Monte Carlo settings |
| ☐ | ☒ | For hierarchical and complex designs, identification of the appropriate level for tests and full reporting of outcomes |
| ☐ | ☒ | Estimates of effect sizes (e.g. Cohen's *d*, Pearson's *r*), indicating how they were calculated |

*Our web collection on statistics for biologists contains articles on many of the points above.*

## Software and code

Policy information about availability of computer code

| Data collection | Participants were recruited on the crowdsourcing platform Prolific (https://app.prolific.co/ ). The experiments were written in JavaScript, using jsPsych (version 7.3.1, https://www.jspsych.org/7.3/ ), 44, and hosted on a web server. |
|---|---|
| Data analysis | Data analysis was done using custom scripts in Python available at https://osf.io/xb43k. |

For manuscripts utilizing custom algorithms or software that are central to the research but not yet described in published literature, software must be made available to editors and reviewers. We strongly encourage code deposition in a community repository (e.g. GitHub). See the Nature Portfolio guidelines for submitting code & software for further information.

## Data

Policy information about availability of data

All manuscripts must include a data availability statement. This statement should provide the following information, where applicable:
- Accession codes, unique identifiers, or web links for publicly available datasets
- A description of any restrictions on data availability
- For clinical datasets or third party data, please ensure that the statement adheres to our policy

Anonymized data, materials and pre-registration documents are available at https://osf.io/xb43k.

# Research involving human participants, their data, or biological material

Policy information about studies with human participants or human data. See also policy information about sex, gender (identity/presentation), and sexual orientation and race, ethnicity and racism.

| | |
|---|---|
| Reporting on sex and gender | We do not report the sex and gender of our participants. |
| Reporting on race, ethnicity, or other socially relevant groupings | No information about race, ethnicity, or other socially relevant groupings was recorded. |
| Population characteristics | No covariates included. |
| Recruitment | Participants were recruited on the crowdsourcing platform Prolific (https://app.prolific.com). |
| Ethics oversight | All experiments were approved by the Medical Sciences Research Ethics Committee of the University of Oxford (approval reference R50750/RE005). Before starting the experiment, informed consent was taken through an online form, and participants indicated that they understood the goals of the study, knew how to raise any questions, how their data would be handled, and that they were free to withdraw from the experiment at any time. |

Note that full information on the approval of the study protocol must also be provided in the manuscript.

# Field-specific reporting

Please select the one below that is the best fit for your research. If you are not sure, read the appropriate sections before making your selection.

☐ Life sciences ☒ Behavioural & social sciences ☐ Ecological, evolutionary & environmental sciences

For a reference copy of the document with all sections, see nature.com/documents/nr-reporting-summary-flat.pdf

# Life sciences study design

All studies must disclose on these points even when the disclosure is negative.

| | |
|---|---|
| Sample size | *Describe how sample size was determined, detailing any statistical methods used to predetermine sample size OR if no sample-size calculation was performed, describe how sample sizes were chosen and provide a rationale for why these sample sizes are sufficient.* |
| Data exclusions | *Describe any data exclusions. If no data were excluded from the analyses, state so OR if data were excluded, describe the exclusions and the rationale behind them, indicating whether exclusion criteria were pre-established.* |
| Replication | *Describe the measures taken to verify the reproducibility of the experimental findings. If all attempts at replication were successful, confirm this OR if there are any findings that were not replicated or cannot be reproduced, note this and describe why.* |
| Randomization | *Describe how samples/organisms/participants were allocated into experimental groups. If allocation was not random, describe how covariates were controlled OR if this is not relevant to your study, explain why.* |
| Blinding | *Describe whether the investigators were blinded to group allocation during data collection and/or analysis. If blinding was not possible, describe why OR explain why blinding was not relevant to your study.* |

# Behavioural & social sciences study design

All studies must disclose on these points even when the disclosure is negative.

| | |
|---|---|
| Study description | Quantitative experimental study. |
| Research sample | Inclusion criteria included being between 18 and 40 years old, reporting no neurological condition, being an English speaker, being located in the US or the UK, not having participated in another version of the task, having a minimal approval rate of 90% on Prolific, and having a minimum of 5 previous submissions on Prolific. These criteria were chosen to ensure participants could understand task instructions (native English), reduce variability due to neurological conditions, and include individuals with sufficient prior experience on Prolific to provide reliable data. As participants were recruited via Prolific, the sample may over-represent younger, English-speaking, and more internet-active individuals (potential selection bias). |
| Sampling strategy | The sampling was random. Sample size was determined based on a pilot study and chosen to provide adequate power to detect medium-sized effects. The sample size for the replication of Experiment 1 was determined via power simulations based on data from Experiment 1, assuming a 50% smaller effect size than observed in that study. The simulations suggested a minimum of 10–20 participants per group, depending on the block. To ensure robust power across all analyses, we conservatively set the sample size to 30 per group. |

| Data collection | All data were collected online on the crowdsourcing platform Prolific (https://app.prolific.co/). During data collection, participants were blind to the experimental condition. No members of the study team were present when participants were performing the task. |
|---|---|
| Timing | Data collection took place between the 31/01/2024 and the 31/05/2024. |
| Data exclusions | No data were excluded from the analysis. |
| Non-participation | All experiments consisted of one session only (no dropout). |
| Randomization | Allocation to experimental conditions was randomized. |

# Ecological, evolutionary & environmental sciences study design

All studies must disclose on these points even when the disclosure is negative.

| Study description | *Briefly describe the study. For quantitative data include treatment factors and interactions, design structure (e.g. factorial, nested, hierarchical), nature and number of experimental units and replicates.* |
|---|---|
| Research sample | *Describe the research sample (e.g. a group of tagged Passer domesticus, all Stenocereus thurberi within Organ Pipe Cactus National Monument), and provide a rationale for the sample choice. When relevant, describe the organism taxa, source, sex, age range and any manipulations. State what population the sample is meant to represent when applicable. For studies involving existing datasets, describe the data and its source.* |
| Sampling strategy | *Note the sampling procedure. Describe the statistical methods that were used to predetermine sample size OR if no sample-size calculation was performed, describe how sample sizes were chosen and provide a rationale for why these sample sizes are sufficient.* |
| Data collection | *Describe the data collection procedure, including who recorded the data and how.* |
| Timing and spatial scale | *Indicate the start and stop dates of data collection, noting the frequency and periodicity of sampling and providing a rationale for these choices. If there is a gap between collection periods, state the dates for each sample cohort. Specify the spatial scale from which the data are taken* |
| Data exclusions | *If no data were excluded from the analyses, state so OR if data were excluded, describe the exclusions and the rationale behind them, indicating whether exclusion criteria were pre-established.* |
| Reproducibility | *Describe the measures taken to verify the reproducibility of experimental findings. For each experiment, note whether any attempts to repeat the experiment failed OR state that all attempts to repeat the experiment were successful.* |
| Randomization | *Describe how samples/organisms/participants were allocated into groups. If allocation was not random, describe how covariates were controlled. If this is not relevant to your study, explain why.* |
| Blinding | *Describe the extent of blinding used during data acquisition and analysis. If blinding was not possible, describe why OR explain why blinding was not relevant to your study.* |

Did the study involve field work? ☐ Yes ☐ No

# Field work, collection and transport

| Field conditions | *Describe the study conditions for field work, providing relevant parameters (e.g. temperature, rainfall).* |
|---|---|
| Location | *State the location of the sampling or experiment, providing relevant parameters (e.g. latitude and longitude, elevation, water depth).* |
| Access & import/export | *Describe the efforts you have made to access habitats and to collect and import/export your samples in a responsible manner and in compliance with local, national and international laws, noting any permits that were obtained (give the name of the issuing authority, the date of issue, and any identifying information).* |
| Disturbance | *Describe any disturbance caused by the study and how it was minimized.* |

# Reporting for specific materials, systems and methods

We require information from authors about some types of materials, experimental systems and methods used in many studies. Here, indicate whether each material, system or method listed is relevant to your study. If you are not sure if a list item applies to your research, read the appropriate section before selecting a response.

## Materials & experimental systems

| n/a | Involved in the study |
|---|---|
| ☐ ☐ | Antibodies |
| ☐ ☐ | Eukaryotic cell lines |
| ☐ ☐ | Palaeontology and archaeology |
| ☐ ☐ | Animals and other organisms |
| ☐ ☐ | Clinical data |
| ☐ ☐ | Dual use research of concern |
| ☐ ☐ | Plants |

## Methods

| n/a | Involved in the study |
|---|---|
| ☐ ☐ | ChIP-seq |
| ☐ ☐ | Flow cytometry |
| ☐ ☐ | MRI-based neuroimaging |

# Antibodies

| Antibodies used | Describe all antibodies used in the study; as applicable, provide supplier name, catalog number, clone name, and lot number. |
|---|---|
| Validation | Describe the validation of each primary antibody for the species and application, noting any validation statements on the manufacturer's website, relevant citations, antibody profiles in online databases, or data provided in the manuscript. |

# Eukaryotic cell lines

Policy information about cell lines and Sex and Gender in Research

| Cell line source(s) | State the source of each cell line used and the sex of all primary cell lines and cells derived from human participants or vertebrate models. |
|---|---|
| Authentication | Describe the authentication procedures for each cell line used OR declare that none of the cell lines used were authenticated. |
| Mycoplasma contamination | Confirm that all cell lines tested negative for mycoplasma contamination OR describe the results of the testing for mycoplasma contamination OR declare that the cell lines were not tested for mycoplasma contamination. |
| Commonly misidentified lines (See ICLAC register) | Name any commonly misidentified cell lines used in the study and provide a rationale for their use. |

# Palaeontology and Archaeology

| Specimen provenance | Provide provenance information for specimens and describe permits that were obtained for the work (including the name of the issuing authority, the date of issue, and any identifying information). Permits should encompass collection and, where applicable, export. |
|---|---|
| Specimen deposition | Indicate where the specimens have been deposited to permit free access by other researchers. |
| Dating methods | If new dates are provided, describe how they were obtained (e.g. collection, storage, sample pretreatment and measurement), where they were obtained (i.e. lab name), the calibration program and the protocol for quality assurance OR state that no new dates are provided. |

☐ Tick this box to confirm that the raw and calibrated dates are available in the paper or in Supplementary Information.

| Ethics oversight | Identify the organization(s) that approved or provided guidance on the study protocol, OR state that no ethical approval or guidance was required and explain why not. |
|---|---|

Note that full information on the approval of the study protocol must also be provided in the manuscript.

# Animals and other research organisms

Policy information about studies involving animals; ARRIVE guidelines recommended for reporting animal research, and Sex and Gender in Research

| Laboratory animals | For laboratory animals, report species, strain and age OR state that the study did not involve laboratory animals. |
|---|---|
| Wild animals | Provide details on animals observed in or captured in the field; report species and age where possible. Describe how animals were caught and transported and what happened to captive animals after the study (if killed, explain why and describe method; if released, say where and when) OR state that the study did not involve wild animals. |
| Reporting on sex | Indicate if findings apply to only one sex; describe whether sex was considered in study design, methods used for assigning sex. Provide data disaggregated for sex where this information has been collected in the source data as appropriate; provide overall |

|                        |                                                                                                                                                                                                                          |
| ---------------------- | ------------------------------------------------------------------------------------------------------------------------------------------------------------------------------------------------------------------------ |
|                        | *numbers in this Reporting Summary. Please state if this information has not been collected.  Report sex-based analyses where performed, justify reasons for lack of sex-based analysis.*                                 |
| Field-collected samples | *For laboratory work with field-collected samples, describe all relevant parameters such as housing, maintenance, temperature, photoperiod and end-of-experiment protocol OR state that the study did not involve samples collected from the field.* |
| Ethics oversight       | *Identify the organization(s) that approved or provided guidance on the study protocol, OR state that no ethical approval or guidance was required and explain why not.*                                                  |

Note that full information on the approval of the study protocol must also be provided in the manuscript.

# Clinical data

Policy information about clinical studies

All manuscripts should comply with the ICMJE guidelines for publication of clinical research and a completed CONSORT checklist must be included with all submissions.

| Clinical trial registration | *Provide the trial registration number from ClinicalTrials.gov or an equivalent agency.* |
| --------------------------- | --------------------------------------------------------------------------------------- |
| Study protocol              | *Note where the full trial protocol can be accessed OR if not available, explain why.*   |
| Data collection             | *Describe the settings and locales of data collection, noting the time periods of recruitment and data collection.* |
| Outcomes                    | *Describe how you pre-defined primary and secondary outcome measures and how you assessed these measures.* |

# Dual use research of concern

Policy information about dual use research of concern

## Hazards

Could the accidental, deliberate or reckless misuse of agents or technologies generated in the work, or the application of information presented in the manuscript, pose a threat to:

No | Yes

- ☐ ☐ Public health
- ☐ ☐ National security
- ☐ ☐ Crops and/or livestock
- ☐ ☐ Ecosystems
- ☐ ☐ Any other significant area

## Experiments of concern

Does the work involve any of these experiments of concern:

No | Yes

- ☐ ☐ Demonstrate how to render a vaccine ineffective
- ☐ ☐ Confer resistance to therapeutically useful antibiotics or antiviral agents
- ☐ ☐ Enhance the virulence of a pathogen or render a nonpathogen virulent
- ☐ ☐ Increase transmissibility of a pathogen
- ☐ ☐ Alter the host range of a pathogen
- ☐ ☐ Enable evasion of diagnostic/detection modalities
- ☐ ☐ Enable the weaponization of a biological agent or toxin
- ☐ ☐ Any other potentially harmful combination of experiments and agents

# Plants

Seed stocks

*Report on the source of all seed stocks or other plant material used. If applicable, state the seed stock centre and catalogue number. If plant specimens were collected from the field, describe the collection location, date and sampling procedures.*

Novel plant genotypes

*Describe the methods by which all novel plant genotypes were produced. This includes those generated by transgenic approaches, gene editing, chemical/radiation-based mutagenesis and hybridization. For transgenic lines, describe the transformation method, the number of independent lines analyzed and the generation upon which experiments were performed. For gene-edited lines, describe the editor used, the endogenous sequence targeted for editing, the targeting guide RNA sequence (if applicable) and how the editor was applied.*

Authentication

*Describe any authentication procedures for each seed stock used or novel genotype generated. Describe any experiments used to assess the effect of a mutation and, where applicable, how potential secondary effects (e.g. second site T-DNA insertions, mosiacism, off-target gene editing) were examined.*

# ChIP-seq

## Data deposition

☐ Confirm that both raw and final processed data have been deposited in a public database such as GEO.

☐ Confirm that you have deposited or provided access to graph files (e.g. BED files) for the called peaks.

Data access links
*May remain private before publication.*

*For "Initial submission" or "Revised version" documents, provide reviewer access links.  For your "Final submission" document, provide a link to the deposited data.*

Files in database submission

*Provide a list of all files available in the database submission.*

Genome browser session
(e.g. UCSC)

*Provide a link to an anonymized genome browser session for "Initial submission" and "Revised version" documents only, to enable peer review.  Write "no longer applicable" for "Final submission" documents.*

## Methodology

Replicates

*Describe the experimental replicates, specifying number, type and replicate agreement.*

Sequencing depth

*Describe the sequencing depth for each experiment, providing the total number of reads, uniquely mapped reads, length of reads and whether they were paired- or single-end.*

Antibodies

*Describe the antibodies used for the ChIP-seq experiments; as applicable, provide supplier name, catalog number, clone name, and lot number.*

Peak calling parameters

*Specify the command line program and parameters used for read mapping and peak calling, including the ChIP, control and index files used.*

Data quality

*Describe the methods used to ensure data quality in full detail, including how many peaks are at FDR 5% and above 5-fold enrichment.*

Software

*Describe the software used to collect and analyze the ChIP-seq data. For custom code that has been deposited into a community repository, provide accession details.*

# Flow Cytometry

## Plots

Confirm that:

☐ The axis labels state the marker and fluorochrome used (e.g. CD4-FITC).

☐ The axis scales are clearly visible. Include numbers along axes only for bottom left plot of group (a 'group' is an analysis of identical markers).

☐ All plots are contour plots with outliers or pseudocolor plots.

☐ A numerical value for number of cells or percentage (with statistics) is provided.

## Methodology

Sample preparation

*Describe the sample preparation, detailing the biological source of the cells and any tissue processing steps used.*

Instrument

*Identify the instrument used for data collection, specifying make and model number.*

Software

*Describe the software used to collect and analyze the flow cytometry data. For custom code that has been deposited into a community repository, provide accession details.*

| Cell population abundance | *Describe the abundance of the relevant cell populations within post-sort fractions, providing details on the purity of the samples and how it was determined.* |
| Gating strategy | *Describe the gating strategy used for all relevant experiments, specifying the preliminary FSC/SSC gates of the starting cell population, indicating where boundaries between "positive" and "negative" staining cell populations are defined.* |

☐ Tick this box to confirm that a figure exemplifying the gating strategy is provided in the Supplementary Information.

# Magnetic resonance imaging

## Experimental design

| Design type | *Indicate task or resting state; event-related or block design.* |
| Design specifications | *Specify the number of blocks, trials or experimental units per session and/or subject, and specify the length of each trial or block (if trials are blocked) and interval between trials.* |
| Behavioral performance measures | *State number and/or type of variables recorded (e.g. correct button press, response time) and what statistics were used to establish that the subjects were performing the task as expected (e.g. mean, range, and/or standard deviation across subjects).* |

## Acquisition

| Imaging type(s) | *Specify: functional, structural, diffusion, perfusion.* |
| Field strength | *Specify in Tesla* |
| Sequence & imaging parameters | *Specify the pulse sequence type (gradient echo, spin echo, etc.), imaging type (EPI, spiral, etc.), field of view, matrix size, slice thickness, orientation and TE/TR/flip angle.* |
| Area of acquisition | *State whether a whole brain scan was used OR define the area of acquisition, describing how the region was determined.* |

Diffusion MRI      ☐ Used      ☐ Not used

## Preprocessing

| Preprocessing software | *Provide detail on software version and revision number and on specific parameters (model/functions, brain extraction, segmentation, smoothing kernel size, etc.).* |
| Normalization | *If data were normalized/standardized, describe the approach(es): specify linear or non-linear and define image types used for transformation OR indicate that data were not normalized and explain rationale for lack of normalization.* |
| Normalization template | *Describe the template used for normalization/transformation, specifying subject space or group standardized space (e.g. original Talairach, MNI305, ICBM152) OR indicate that the data were not normalized.* |
| Noise and artifact removal | *Describe your procedure(s) for artifact and structured noise removal, specifying motion parameters, tissue signals and physiological signals (heart rate, respiration).* |
| Volume censoring | *Define your software and/or method and criteria for volume censoring, and state the extent of such censoring.* |

## Statistical modeling & inference

| Model type and settings | *Specify type (mass univariate, multivariate, RSA, predictive, etc.) and describe essential details of the model at the first and second levels (e.g. fixed, random or mixed effects; drift or auto-correlation).* |
| Effect(s) tested | *Define precise effect in terms of the task or stimulus conditions instead of psychological concepts and indicate whether ANOVA or factorial designs were used.* |

Specify type of analysis:      ☐ Whole brain      ☐ ROI-based      ☐ Both

Statistic type for inference      *Specify voxel-wise or cluster-wise and report all relevant parameters for cluster-wise methods.*

(See Eklund et al. 2016)

| Correction | *Describe the type of correction and how it is obtained for multiple comparisons (e.g. FWE, FDR, permutation or Monte Carlo).* |

## Models & analysis

| n/a | Involved in the study |
|---|---|
| ☐ | ☐ Functional and/or effective connectivity |
| ☐ | ☐ Graph analysis |
| ☐ | ☐ Multivariate modeling or predictive analysis |

Functional and/or effective connectivity

*Report the measures of dependence used and the model details (e.g. Pearson correlation, partial correlation, mutual information).*

Graph analysis

*Report the dependent variable and connectivity measure, specifying weighted graph or binarized graph, subject- or group-level, and the global and/or node summaries used (e.g. clustering coefficient, efficiency, etc.).*

Multivariate modeling and predictive analysis

*Specify independent variables, features extraction and dimension reduction, model, training and evaluation metrics.*

nature portfolio | reporting summary

April 2023

