## [Peer Review File · Nature Human Behaviour]

Shared sensitivity to data distribution during learning in humans and transformer networks

Corresponding Author: Dr Jacques Pesnot Lerousseau

Version 0:

Decision Letter:

24th April 2025

Dear Dr Pesnot Lerousseau,

Thank you once again for your manuscript, entitled "Do humans learn like transformer networks?", and for your patience during the peer review process.

Your Article has now been evaluated by 3 referees. You will see from their comments copied below that, although they find your work of considerable potential interest, they have raised quite substantial concerns. In light of these comments, we cannot accept the manuscript for publication, but would be interested in considering a revised version if you are willing and able to fully address reviewer and editorial concerns.

We hope you will find the referees' comments useful as you decide how to proceed. If you wish to submit a substantially revised manuscript, please bear in mind that we will be reluctant to approach the referees again in the absence of major revisions. We are committed to providing a fair and constructive peer-review process. Do not hesitate to contact us if there are specific requests from the reviewers that you believe are technically impossible or unlikely to yield a meaningful outcome.

In particular, we feel that in the light of the comments of Reviewer #2 (on novel insight and ecological validity) and Reviewer #3 (on robustness), additional data collection will be necessary to address the concerns raised.

If you wish to submit a suitably revised manuscript, we would hope to receive it within 4 months. I would be grateful if you could contact us as soon as possible if you foresee difficulties with meeting this target resubmission date.

- Include a "Response to the editors and reviewers" document detailing, point-by-point, how you addressed each editor and referee comment. If no action was taken to address a point, you must provide a compelling argument. When formatting this document, please respond to each reviewer comment individually, including the full text of the reviewer comment verbatim followed by your response to the individual point. This response will be used by the editors to evaluate your revision and sent back to the reviewers along with the revised manuscript.
- Highlight all changes made to your manuscript or provide us with a version that tracks changes.

Link Redacted

Thank you for the opportunity to review your work. Please do not hesitate to contact me if you have any questions or would like to discuss the required revisions further.

Sincerely,

██████████
██████████
██████████
Nature Human Behaviour

REVIEWER COMMENTS:

Reviewer #1 (Remarks to the Author):

This fascinating work presents an analysis of the multiple mechanisms that humans and transformer models use to solve a classification task. The work finds some very interesting parallels in the behavior of the two systems across multiple levels. First, manipulating the skew of the data distribution causes both humans and transformers to cross over from a memory-based (in-weights) to an in-context strategy. Furthermore, the human behavioral implementation of the in-context strategy in a (restricted-information) mouse-tracking experiment mirrors the transformer induction mechanism (replicated from prior work). The authors use diagnostic tests to determine which strategy each system is learning. Most humans and models trained at the critical tipping point value learn either one strategy or the other, but a few learn to use both. A composite distribution that is targeted to encourage both strategies increases rates of dual learning for both humans and models. The authors further propose a curriculum that encourages humans to learn to perform both strategies, but the transformers do not benefit.

Overall, I think this work shows many surprising and interesting parallels in learning between humans and transformers on this task, and I recommend it for publication. I have some follow-up questions and suggestions about the results below:

On the ML experiments:

* One interesting finding of this work is about the extent to which transformers become dual learners at $\alpha = 1$. This is summarized succinctly in the discussion: "Previous studies using a similar methodology have argued that $\alpha = 1$ represents a "sweet spot" at which both in-weights and in-context learning are possible in transformers. We show here that what seems to be true at the level of the population is not true at the individual model level, as no single network learned both strategies in tandem using Zipf distributions." — these are quite interesting differences in interpretation from the prior results, e.g. in the Chan et al. work with which I'm most familiar (and where I believe this finding originated). However, qualitatively, the performance numbers reported by Chan et al. for both strategies in their work at $\alpha=1$ (yellow bars in Fig. 6c-d) seem too high to be explained by averaging a mixture of models performing only one or the other — I would expect such results to at best linearly interpolate between ceiling performance and chance on the two evaluations, and their results appear to exceed that simplex, and are much higher than the tradeoff performances you see at $\alpha=1$. This makes me wonder whether other differences between the setups (e.g. the much smaller number of classes you used) are important in determining the solutions the models arrive at.

- It seems that if you want to make a strong claim about the differences, it would be worth varying other dataset parameters like the number of classes to identify whether these are the source of the different results — it seems to me that there may simply be inadequate pressure for ICL once memorization occurs with few classes.

- I appreciated the architecture sweep (S5) suggesting that features like depth alone do not explain the differences. However, were these still attention-only models until the output? It seems possible that the lack of MLPs interleaved with the attention layers throughout contributes to the differences; for example, a model with interleaved MLPs would have much greater capacity to memorize while still dedicating some MLPs to ICL-relevant computations, thereby perhaps supporting both strategies.

* Several follow-ups to some of the ML work on ICL cited

(https://proceedings.neurips.cc/paper_files/paper/2023/hash/58692a1701314e09cbd7a5f5f3871cc9-Abstract-Conference.html and <https://arxiv.org/abs/2503.05631>) use sequences similar to your arbitrage ones for evaluating models, and might be worth reviewing/discussing. For example, the asymmetry in transitions between ICL and IWL that you observe could be related to the direction of transience observed in those papers, where ICL tends to give way to IWL asymptotically, rather than the other way around.

On the human experiments:

* If I understand correctly, the distractor labels and images in the context are sampled uniformly and independently. Thus, the humans will have substantial exposure to *incorrect* image->label pairings in the contexts if they bother to look. It seems to me that it would be more natural to have correct image->label pairs for the distractors that simply don't match the target image. Was there a motivation for this design choice? It might be worth discussing in the text.

* A question related to that: it wasn't entirely clear to me from the text whether the mapping from images -> labels was semantically meaningful (e.g. all bears get label 1) or arbitrary (i.e. each image is arbitrarily assigned a label independently). I assume it is likely the latter or people would readily generalize from in-weights knowledge, but might be worth stating more explicitly (unless I missed it somewhere).

* One question I had regarding the shared induction mechanism is whether the results could be otherwise. Of course the humans will need to look at both the image and its corresponding label if they use in-context information to solve the task.

Given that they use in-context information, is there a reasonable alternative strategy that they could take besides the one that they did?

* Was there a motivation for always using “3 steps clockwise” as the rule for the humans, rather than randomizing the distance from image to label across subjects?

* I would appreciate seeing the exact instructions the participants received in the materials & methods; for example, it seems to me that what, if anything, was said about the reason for numbers appearing on screen between the images might potentially contribute to the participants succeeding or failing to learn the in-context rule.

Once again, really interesting work.

Best,

Andrew Lampinen

Reviewer #1 (Remarks on code availability):

I did not thoroughly review the code, but glanced through it briefly to try to find answers to one of my questions; based on my past experiences with jspsych this looks to be reasonably clear and usable code.

Reviewer #2 (Remarks to the Author):

Summary

The paper presents results from experiments testing humans and neural networks on a task where images were associated with labels. In the task, two strategies were available: an “in-context” strategy where the label could be inferred from information given in context (by looking at the label that was +3 positions clockwise from the query image), and an “in-weight” strategy where the labels corresponding to each image were memorized. The experiments were designed to study how the training distribution and curriculum affected which of the two strategies was learned. In both humans and transformers, redundancy (i.e., a skewed distribution where some items occurred very frequently) drove in-weight learning: when most trials could be performed by memorizing only a few specific examples, participants/models were not incentivized to make the effort to extract the rule from the context. Conversely, diversity drove in-context learning: when there were too many things to memorize, participants were incentivized to find the simple rule and avoid memorization, and could thereby generalize to novel items. A separate set of experiments found that while humans benefit when the training curriculum emphasized diversity early in training, transformers did not. Finally, analysis of the internal mechanisms used by the transformers and the cursor trajectories from the human participants revealed similar mechanisms used by humans and transformers for the in-context and in-weight strategies: in both, the in-context strategy involved attending to the labels +3 positions clockwise from the query image, while the in-weight strategy involved attending only to the query image itself.

Review

Overall, we thought this was a strong paper that has the potential to make an important bridge between ongoing research in AI on LLMs and existing work in cognitive psychology. However, our main concerns are that 1) the types of in-context learning and in-weight learning studied here just barely correspond to those studied in LLMs – i.e., they are so simple that they may not actually be relevant to the advanced abilities of modern LLMs, and 2) it is unclear what theoretical contributions the findings make to cognitive psychology (i.e., without the transformer experiments one might have made the same predictions linked to existing dual-process theories). The conclusions of the paper would be better substantiated if more advanced forms of ICL/IWL were investigated, and/or if deeper connections to existing literature in cognitive psychology were discussed in much more detail.

Strengths

* The paper explores an important and interesting topic, bridging cognitive science / psychology and AI. In-context vs. in-weight learning has become an important area of research in NLP, and there are many interesting questions about how these processes can be related to human learning and reasoning. This paper has potential to make an important contribution in this area, especially to anyone who has been thinking about what, if anything, the advanced in-context learning abilities of AI systems (LLMs) have to do with analogous abilities in human cognition.

* The experiments were very well executed. The humans and transformers can be very cleanly compared. This is a great example of how to do research at the intersection of cognitive science and AI.

* The diversity of methods used in the experiments is also impressive. The behavioral experiments and analyses were very well complemented by the transformer simulations, and the mechanistic interpretability study was very well complemented by the analysis of the cursor trajectories (Figure 5).

Weaknesses

* Although the methodology seems sound and the experiments are well-grounded in ongoing research on LLMs (the experimental paradigm directly replicates an existing paradigm used in work on ICL in transformers), the results are not particularly surprising or diagnostic of ICL vs IWL specifically (see specific comments 2 and 3 below for more details).

-- The basic finding that the distribution of examples determines the strategy learned by humans is intuitive, and seems like it

has more to do with the affordances of the inputs than with architecture per se (see below).

-- The findings about curriculum effects were more interesting, but the finding that neural networks don't do very well when samples aren't presented i.i.d. (due to catastrophic interference) is very well established (see below).

* Given that most of the results were relatively intuitive, it is unclear what theoretical significance they have for cognitive psychology / neuroscience. For example, various studies suggest that when rules can be extracted over trials (e.g., in category learning or in reinforcement learning etc) people can discover them to generalize across related examples. Conversely, if the same response is almost always correct, there is no need or incentive to discover or apply any rule. There are many multiple-system computational theories across cognitive neuroscience that are relevant here, e.g., episodic vs. procedural memory, complementary learning systems, model based vs model free learning, state abstraction, structure learning vs statistical learning, rapid learning vs. later retention under varying memory demands, and the costs and benefits of engaging in cognitive effort... A broader discussion of relevant literature from cognitive science / neuroscience would help to further connect these recent phenomena observed in AI systems with existing theories about human cognition.

-- Relatedly, we are skeptical about how relevant the current findings would be to education research (as the authors mentioned in the discussion), as the task used in the experiments is very simple and seems far removed from any learning scenario in the real world.

-- The results could also be interesting to researchers outside of psychology/neuroscience (in AI), but as mentioned above, we also have some minor reservations about how representative these minimal forms of ICL/IWL are when compared to the broader capabilities exhibited by LLMs (see note 2 below).

Specific comments

1. Title: In our opinion the title is too vague – humans almost certainly “learn like transformer networks” in some ways, and certainly don't in other ways. A more informative title would mention the specific similarities / dissimilarities found in this particular study. In general some people in the field tend to overgeneralize on these kinds of generic questions, so specificity should be prioritized wherever possible.

2. The in-context vs. in-weight learning studied here is quite minimal. In-context learning is critical to current AI systems, as it allows them to be deployed for many different purposes without retraining/finetuning. This is why understanding how it is accomplished in LLMs has become such a big topic in AI/NLP. However, the minimal cases of ICL that are used in some of that work (including in the task presented here), are often so simple as to be unrepresentative. Is it really “in-context learning” to simply apply a known (+3 positions) rule – discovered over previous examples – to a specific set of stimuli? This is more akin to any of the other frameworks mentioned above studying rule-based systems in category learning, RL and generalization, etc. So while this task technically also “counts” as involving a minimal form of ICL, the more impressive form of ICL attributed to transformers and humans involves the ability to infer new rules based on a few examples (or instructions) given in context rather than applying a single known rule learned from many previous instances. Similarly, the IWL studied here is very minimal: in the case of the extremely skewed distribution, using a single response would achieve 92% accuracy (see 3b below).

a. Related to the point about the lack of ecological validity w.r.t. real-world AI systems (LLMs): the simple attention-only transformer was presumably used so that the interpretability methods would be cleaner/easier to do (?), but it seems like this could affect results in some cases, as the MLPs in LLMs are known to be involved in memorizing specific facts (and therefore might end up changing the tendency toward the IWL memorization strategy). It might be worthwhile to include some analogous experiments with larger transformers that had MLPs, so we can be more confident the results would generalize to LLMs.

3. Per-item results should be included for in-weights test

a. It is unclear from the way results were presented how well people/models were doing on the in-weights test on the rare examples. It would be informative to also include the results on the in-weights test where items are weighted uniformly rather than weighted according to the skewed distributions. Alternatively, the accuracies could be shown on a per-item basis in a supplemental figure. These would be particularly important for understanding the results related to double learning and in the arbitrage test.

b. If performance is poor for rare examples in the IWL test, then it means that people doing the IWL strategy have barely learned anything at all, and could even be simply responding with the most likely answer in the extremely skewed cases. This is important because if people aren't memorizing a ton of examples, this shows an asymmetry where they're not really choosing to do IWL or ICL in general, but just choose to try hard to find the ICL rule depending on how easily they can get away with memorizing a few examples. Seems like people are just trying to minimize effort overall, and it's easier to memorize 2-3 examples than it is to find the rule, but finding the rule is easier than memorizing if there are more than just a few examples.

4. Results about differences in architecture are unconvincing

a. The humans, like the transformer, had access to the entire context when considering the query. But for the recurrent neural network to learn the in-context strategy, it would have to learn to maintain all of the contextual information until the end of the sequence, where it could (upon seeing the query) retrieve the label presented 3 positions after the query in the context. I

wonder if the recurrent network would be more likely to learn the ICL strategy if it got the query before the in-context examples, so that it would know what to look for as the context examples were shown – or if all items were presented simultaneously at each time point (with some kind of positional encodings). Conversely, I wonder how often humans would learn the ICL strategy if they had to remember all of the item-label pairs in the context before receiving the query. It seems unlikely that they would ever learn such a strategy if many examples were included in the context.

b. As for the feedforward network, did it receive anything like positional encodings? If not, then it would have no way of generalizing a +3 rule, and would have to re-learn this rule separately for each position in the sequence.

c. This is all just to say that I think the inference that humans “learn like transformers” (compared to RNNs or MLPs) would be premature in this case, as I would guess that the basic results about ICL vs. IWL depend more on the way that the inputs are presented (to both the RNN/MLP and to the humans) than on the architecture itself.

5. Curriculum effects

a. The curriculum effects were interesting, but it is unclear whether any conclusions can be drawn other than the well-established fact that neural networks suffer from catastrophic interference when trials are blocked over time. For example, if items from previous blocks were mixed into later blocks (simulating a kind of episodic replay), do the networks become double learners more often with one of the two curricula?

b. We were a little confused about why transformers learn the in-weight strategy at all when trained on the C1 curriculum. If they’ve learned the in-context strategy after the uniform training, shouldn’t they be getting 0 loss on the skewed blocks from the start, and therefore shouldn’t learn anything during those latter blocks? It might be informative to include the training curves over the blocks – did the models converge to 0 loss before the next block was started?

c. Relatedly, it could be worthwhile to include the accuracy results over the course of training the humans in the experiments that used blocked curricula.

6. A few minor points about some of the language used in the introduction:

a. Is “relying on social norms” a good example of an inductive inference? This seems pretty far removed from the kind of inductive inference required in the current study.

b. Unclear what “inextricably intertwining computation and memory” means: for example, one way of disentangling “computation” and “memory” in neural networks would be in the distinction between in-context and in-weight learning...

c. “in-context learning is a kind of metalearning” – it may be more apt to think of in-context learning as a result of metalearning (learning how to learn in context).

d. The transformer isn’t an autoregressive algorithm per se -- transformers can also be trained on different non-autoregressive language modeling tasks (e.g., masked language modeling, translation). The transformer also isn’t technically a learning algorithm at all – it’s an architecture whose parameters can be trained by different learning algorithms (gradient descent / backprop).

e. This sentence is slightly incorrect and misleading: “transformer...uses a feedforward neural network to learn the relative importance (self-attention) among tokens and positions in a sequence”. Being a bit more descriptive might help readers who are not familiar with the transformer architecture.

7. What does “fully informative feedback” mean? Was the context still present when feedback was given? It would probably be important for people to see the feedback in the presence of the context for them to infer the +3 rule.

8. We didn’t see the accuracy results from the in-context and in-weights tests from Experiment 4 reported anywhere. Were the basic findings related to ICL and IWL replicated in the case where the display was blurred/obscured?

Formatting issues / typos

- The final supplementary figure says “2 heads” for the feedforward network and LSTM. I assume this is an error and those architectures didn’t have any attention heads?

- It might be nice to include Experiment X labels as section titles, or at least in the figures to help the reader navigate.

- Figure 1 caption: “the only to be accurate is to use information...”
“Corpuses” -> “corpora”? (or is this a britishism?)

- I think this one occurred multiple times: “trained on skewed distribution”, “trained on uniform distribution”

- “performance at training”?

- “the computational underlying”

- Fig 4C: the solid arrows are confusing to follow. The pattern of movement could be made more clear in the visualization as well as in the caption.

-- Discussion section – “Recent theoretical work has ... solutions”. The sentence seems ill-formed and less clear. Possibly elaborate to make it more clear.

-- “in weighs” typo Fig 1C caption

Reviewer #3 (Remarks to the Author):

This paper compares human and transformer learning. It is found that both agents are sensitive to the training environment; skewing the distribution makes them more in weight learners, a uniform distribution makes them more in context learners.

This paper presents an interesting comparison between the two agents. The authors have the tendency to overstate the novelty of the findings (e.g., that transformers can learn in context and in weight; or that they suffer from catastrophic interference whereas humans do not), but the systematic comparison and human experimentation is interesting. Some comments are the following.

One finding is that MLPs and LSTMs don't learn in context. But how sure are you of this null finding? How well did you check the parameter space to confirm this fact? I see that the number of parameters is matched between the agents, but this is not necessarily the best matching. Did you check the parameter space as thoroughly as for the transformer? It would be worthwhile to explore the parameter space much more thoroughly than done now; currently I have no idea how general the finding is. And in fact, the same comment holds for the transformer: How broad is the finding that it can learn in context and in weights (across parameter settings)?

N = 30 in a between-subjects design is small. Is there an a priori power analysis to justify this sample size?

For C3, C4, it's weird that for the comparisons where a null difference is “convenient”, the severe Bonferroni correction is applied. I would use the exact same procedure as for the other contrasts.

It's interesting to see how the transformer solves the problem. But it's an overstatement to claim that this is how humans do it. First, if the transformer is trained on +3, it likely cannot generalize at all (?) to +2 or +4. If it is then trained on +2 or +4, it likely does not generalize at all from its previous knowledge. Second, when going back to the +3 task, it likely then has to learn again from scratch. I'd be happy to see this intuition confirmed (or more interesting, refuted!). If confirmed, it leaves me thinking about what has been shown precisely in the paper.

In Exp 4, I find the mouse paradigm a clever idea, but the data in Fig 4B is misleading. The relevant comparison is the one in Suppl 4, comparing in context tests for the two distributions. The statistics support the authors' claim so ok to keep it, but the reader should be able to make the appropriate comparisons based on the figures in the main text.

At a conceptual level, it's not entirely clear to me what is the takehome message. Both agents can learn in context, but one is also resistant to catastrophic interference (and MLP is neither). What, then, is the message for human cognition? I realise this is a grand question that a single paper cannot answer, but the introduction also promises a lot, so it seems correct to ask it.

Relatedly, the comparisons between the two agents are a bit superficial; is it possible to do a more detailed fitting to human data so we can see which components precisely in a transformer are important for the match to human data. This question admittedly is itself a bit superficial, so ok to ignore it but I think a tighter match would be of interest, even if only as a promise for the future.

Reviewer #3 (Remarks on code availability):

N/A

Version 1:

Decision Letter:

Our ref: NATHUMBEHAV-25010469A

10th September 2025

Dear Dr. Pesnot Lerousseau,

Thank you for submitting your revised manuscript "Shared sensitivity to data distribution during learning in humans and transformer networks" (NATHUMBEHAV-25010469A). It has now been seen by the original referees and their comments

are below. As you can see, the reviewers find that the paper has improved in revision. We will therefore be happy in principle to publish it in Nature Human Behaviour, pending minor revisions to satisfy the referees' final requests and to comply with our editorial and formatting guidelines.

We are now performing detailed checks on your paper and will send you a checklist detailing our editorial and formatting requirements within two weeks. Please do not upload the final materials and make any revisions until you receive this additional information from us.

Sincerely,

[REDACTED]

[REDACTED]

[REDACTED]

Nature Human Behaviour

Reviewer #1 (Remarks to the Author):

Thanks to the authors for their deep engagement with my review; they've addressed my concerns, and I appreciate the clarifications, additional experiments and details. I really appreciated seeing the strategy learning curves. Also, the transitive inference results are quite nice, and enhance the generality of the findings. I maintain that this is a very interesting topic, and find this version to be a substantially improved paper, and I strongly recommend it for publication.

One minor note on the $\alpha = 1$ tradeoff: I would generally anticipate the possibility that $ICL + IWL_{common} > 1$ from a mixture of pure strategies; that's why I said the simplex interpolating between perfect and chance performance, not perfect and zero. Specifically, I would expect that the sum $ICL + IWL_{common}$ could be as large as $1 + chance_ICL + chance_IWL$ (if each of the models are doing perfectly on their preferred subset, and chance on the other) — which looks roughly compatible with the results shown in your response analysis. But Chan et al.'s performance exceeds that; the highest we'd expect from pure strategies is ~ 1.5 whereas their summed results are ~ 1.7 . It's a minor point though, and as the authors note a variety of differences in details like # of classes or data diversity might explain it.

Reviewer #2 (Remarks to the Author):

In the first round, we thought the paper had the the potential to make a strong contribution, but had reservations about 1) the narrowness of the kind of ICL required in the main task (applying a +3 rule), and 2) the lack of a broader discussion situating the findings in the context of other work in cognitive psychology and neuroscience. The additional experiments and revisions the authors have made in response were substantial, and meaningfully address our major concerns.

Narrowness of ICL. The new modeling experiments using the transitive inference task show that the effect of the distribution of the training data remains intact in the models even when the task requires a more sophisticated form of ICL. Rather than requiring a simple application of a known +3 rule learned throughout training, this task required the models to make transitive inferences about arbitrary stimuli given in context. The emergence of this more sophisticated form of ICL also depended on the skewness of the training distribution, where more diversity (i.e., less skew) again resulted in greater use of the in-context strategy. While the ICL abilities of LLMs are much more sophisticated and flexible than any that could be learned in this kind of controlled setting, we think this new experiment makes a major step in this direction. And we acknowledge that it would be difficult to study the development of the ICL abilities of LLMs in a controlled manner. It would be nice to know if humans also exhibit the same effects in this transitive inference paradigm, but the modeling experiments presented in the revised manuscript substantively address our previous concerns about generalization to larger-scale ICL abilities such as those of LLMs.

Connections to other cognitive theories. The revisions to the title, introduction, and discussion qualify the major claims that we found problematic in the first version, and go some way toward relating the work to existing theories in cognitive psychology / neuroscience. We still think the findings are largely consistent with alternative cognitive perspectives (e.g., it isn't very surprising that humans don't put in the effort to extract a general rule when they can get most answers correct by simply memorizing a handful of items — see [1] and [2] below for example papers studying the conditions under which humans engage effortful controlled processing). And while we agree that this specific dependence of the learned strategy on the precise statistical properties of the training distribution may fall outside the scope of classical dual-process models, even there some computational frameworks do suggest that the engagement of control vs habitual (in-weight, model-free like processing) is arbitrated based on Bayesian uncertainty [3], [4], which is in turn related to frequency of learned associations. Including further discussion of how the basic distinction between ICL and IWL might relate to existing ideas from cognitive psychology/neuroscience (e.g., about working memory, when humans choose to exert cognitive effort to extract rules, model-based vs model-free processes etc.) might broaden the interdisciplinary appeal of the paper, but we can also see the appeal of emphasizing links to recent research in machine learning / AI, and acknowledge that it might be difficult to accomplish both in such a short paper.

We also found the many additional figures / analyses included in the recent revision to be very helpful. The figures showing the main results binned by frequency were clear, and give a fuller picture of the results (although see minor point below about confusing colorbar). We also found the additional experiment testing the LSTM with query-first inputs to be pretty interesting.

Overall the additional experiments and substantive revisions to the manuscript represent a major effort to address our concerns. We think the revised version is much improved and represents a strong contribution with many interesting results.

Other miscellaneous points:

We appreciate the change to the title, and the additional qualifications in the discussion section.

We noticed that there is still a colorbar showing different shades for different Zipf exponents in Figures S3 and S4. This appears to be a mistake that was left over from the previous figures? The shading now only corresponds to the frequency bins right?

We appreciated the explanation about why models learn the IWL strategy even after uniform training in curriculum C1.

[1] Shenhav, A., Botvinick, M. M., & Cohen, J. D. (2013). The Expected Value of Control: An Integrative Theory of Anterior Cingulate Cortex Function. *Neuron*, 79(2), 217–240. <https://doi.org/10.1016/j.neuron.2013.07.007>

[2] Frömer, R., Lin, H., Dean Wolf, C. K., Inzlicht, M., & Shenhav, A. (2021). Expectations of reward and efficacy guide cognitive control allocation. *Nature Communications*, 12(1), 1030. <https://doi.org/10.1038/s41467-021-21315-z>

[3] Daw ND, Niv Y, Dayan P. Uncertainty-based competition between prefrontal and dorsolateral striatal systems for behavioral control. *Nat. Neurosci.* 2005;8:1704–1711. doi: 10.1038/nn1560.

[4] Lee SW, Shimojo S, O'Doherty JP. Neural computations underlying arbitration between model-based and model-free learning. *Neuron*. 2014 Feb 5;81(3):687-99. doi: 10.1016/j.neuron.2013.11.028.

Reviewer #3 (Remarks to the Author):

I think the authors have done a good job responding to the comments; they have collected additional data, evaluated robustness, and contextualized some of the claims. I have no remaining questions.

Two small comments (no followup required):

- I find it weird that for the replication the same (small) data sample size was used; but the predictions were replicated (and preregistered), so that is reassuring.
- In the intro, I don't think it's necessary to drag out each possible dual-system theory, some of which don't clearly link to the current one (e.g., associative versus symbolic). I see that this is in relation to another reviewer, so fine to keep it anyway.

Reviewer #1

This fascinating work presents an analysis of the multiple mechanisms that humans and transformer models use to solve a classification task. The work finds some very interesting parallels in the behavior of the two systems across multiple levels. First, manipulating the skew of the data distribution causes both humans and transformers to cross over from a memory-based (in-weights) to an in-context strategy. Furthermore, the human behavioral implementation of the in-context strategy in a (restricted-information) mouse-tracking experiment mirrors the transformer induction mechanism (replicated from prior work). The authors use diagnostic tests to determine which strategy each system is learning. Most humans and models trained at the critical tipping point value learn either one strategy or the other, but a few learn to use both. A composite distribution that is targeted to encourage both strategies increases rates of dual learning for both humans and models. The authors further propose a curriculum that encourages humans to learn to perform both strategies, but the transformers do not benefit.

Overall, I think this work shows many surprising and interesting parallels in learning between humans and transformers on this task, and I recommend it for publication. I have some follow-up questions and suggestions about the results below:

On the ML experiments

1. One interesting finding of this work is about the extent to which transformers become dual learners at $\alpha = 1$. This is summarized succinctly in the discussion: “Previous studies using a similar methodology have argued that $\alpha = 1$ represents a “sweet spot” at which both in-weights and in-context learning are possible in transformers. We show here that what seems to be true at the level of the population is not true at the individual model level, as no single network learned both strategies in tandem using Zipf distributions.” — these are quite interesting differences in interpretation from the prior results, e.g. in the Chan et al. work with which I’m most familiar (and where I believe this finding originated). However, qualitatively, the performance numbers reported by Chan et al. for both strategies in their work at $\alpha=1$ (yellow bars in Fig. 6c-d) seem too high to be explained by averaging a mixture of models performing only one or the other — I would expect such results to at best linearly interpolate between ceiling performance and chance on the two evaluations, and their results appear to exceed that simplex, and are much higher than the tradeoff performances you see at $\alpha=1$. This makes me wonder whether other differences between the setups (e.g. the much smaller number of classes you used) are important in determining the solutions the models arrive at.

- It seems that if you want to make a strong claim about the differences, it would be worth varying other dataset parameters like the number of classes to identify whether these are the source of the different results — it seems to me that there may simply be inadequate pressure for ICL once memorization occurs with few classes.

We thank the Reviewer for this comment. This has indeed been a challenging point to address, and we appreciate the opportunity to clarify it.

As the Reviewer notes, the performance numbers reported by Chan et al. (2022) at $\alpha = 1$ appear too high to result from averaging a mixture of models that each perform only one of the two strategies (ICL or IWL). However, it is important to emphasize that Chan et al. report in-weights learning (IWL) performance specifically on the 10 most common classes, not averaged across all classes as in our study:

“As we increase the Zipf exponent, i.e. increasing the skew on the class distribution, we see a decrease in in-context learning. In-weights learning of the 10 most common classes [among 1600] in contrast increases with more skew. Rare items from training are never memorized (performance is at chance for all Zipf exponents).” (Chan et al., 2022).

Under this setup, averaging over a mixture of models that specialize in either ICL or IWL can lead to $ICL + IWL_{\text{common}} > 1$.

We replicated Chan et al.’s analysis as closely as possible by partitioning performance into "common" (top 10) and "rare" (bottom 100) classes (see Fig. R#1–1). We indeed find that at $\alpha = 1$, $ICL + IWL_{\text{common}} > 1$, even though each of our models learns only one strategy — either in-context or in-weights learning. Anecdotally, we also replicate their observation that rare classes are never learned. A direct comparison of the accuracy is challenging because the chance level is not the same in our setup and in Chan et al.’s setup.

Figure for Reviewer #1–1. We replicated Chan et al. (2022) analysis by splitting the in-weights test performance for common (10 most common classes) and rare classes (100 least common classes).

We now also plot training curves for multiple examples networks trained on different values of α in a new Fig. S2. On this figure, we show four representative training curves from models trained at $\alpha = 1$ (middle row), highlighting how models tend to trade-off between the two strategies:

Figure Supp. 1. Accuracy curves for multiple example transformer networks trained on different training distributions, uniform ($\alpha = 0$, top row), moderately skewed ($\alpha = 1$, middle row) and skewed ($\alpha = 2$, bottom row). In-context test performance and arbitrage test performance (with respect to in-context learning) strongly overlap. Over the course of training, in-context test performance trade-off with in-weights test performance.

- I appreciated the architecture sweep (S5) suggesting that features like depth alone do not explain the differences. However, were these still attention-only models until the output? It seems possible that the lack of MLPs interleaved with the attention layers throughout contributes to the differences; for example, a model with interleaved MLPs would have much greater capacity to memorize while still dedicating some MLPs to ICL-relevant computations, thereby perhaps supporting both strategies.

We thank the Reviewer for this suggestion, shared with Reviewer #2. To investigate this further, we repeated the architecture sweep using transformer models with MLPs interleaved between attention layers. This modification did not result in any qualitative changes in the outcomes – the results closely replicated those obtained with attention-only models.

Figure Supp. 11. Performance of transformers with interleaved MLP with varying architecture sizes. The MLP blocks consist of two dense layers with a ReLU activation, followed by a residual connection and layer normalization. Scatter plots of the in-context vs in-weights test performances for transformers with varying numbers of layers, number of heads per layers, and varying training distributions. Each dot represents a model trained with a specific number of layers, attention heads, and training data distribution. Dot color indicates the α exponent of the training distribution. Dotted lines indicate chance-level performance.

“We also tested an interleaved MLP model (Fig. S11), where each attention layer was followed by a feedforward (MLP) block consisting of two dense layers with D_M units (with ReLU activation), a residual connection, and a layer normalization step.” (l. 831)

This aligns with preliminary tests we conducted that showed that MLPs do not appear to play a critical role in this learning paradigm, which initially motivated our choice to focus on attention-only architectures. We have added a supplementary figure reporting these results and included the following clarification in the main text:

“We tried with different model sizes, and confirmed that it was also the case with larger and deeper models, with and without interleaved feedforward layers between attention layers (up to 4 attention heads per layer, up to 10 layers, see Fig. S10-S11).” (l. 555)

2. Several follow-ups to some of the ML work on ICL cited (https://proceedings.neurips.cc/paper_files/paper/2023/hash/58692a1701314e09cbd7a5f5f3871cc9-Abstract-Conference.html and <https://arxiv.org/abs/2503.05631>) use sequences

similar to your arbitrage ones for evaluating models, and might be worth reviewing/discussing. For example, the asymmetry in transitions between ICL and IWL that you observe could be related to the direction of transience observed in those papers, where ICL tends to give way to IWL asymptotically, rather than the other way around.

We thank the Reviewer for these references, which help shed light on our results. Indeed, several recent papers have indeed explored transitions between ICL and IWL using paradigms very similar to our arbitrage trials. These works introduce related terms such as “ICL2” (Reddy et al., 2023), “Flip” (Saxe et al., 2024), and “Swap” (Tong & Pehlevan, 2024) to describe what we call the “arbitrage” condition. While terminology varies, the underlying design is nearly identical: the query can be answered using either a memorized label from training or a rule inferred from the local context, but these two strategies yield conflicting answers:

“Note that this condition is nearly identical to setups used in recent machine learning studies: the “ICL2” trials of Reddy (2023), the “Flip” condition in (Singh et al., 2025), and the “Swap” condition in (Tong & Pehlevan, 2024).” (l. 250)

To improve clarity, we now also plot arbitrage performance over the course of training (see **Fig. S1** above). Furthermore, we now cite and discuss recent papers investigating the asymmetry in transitions between ICL and IWL. In particular, Singh et al., (2025) and Saxe et al. (2024) show that once a model has developed strong in-weights associations (CIWL strategy), it tends to fail to recover in-context learning. In contrast, models that start with in-context learning can later switch to IWL. This dynamic directly aligns with our curriculum results: when humans or transformers are first trained on skewed data (favoring IWL), they fail to later adopt ICL; but when training begins with uniform data (favoring ICL), they remain able to incorporate redundancy and shift toward IWL:

(Discussion)

“This results aligns with recent findings on asymmetries between in-context and in-weights learning. Specifically, Singh et al. (2023) showed that in-context learning tends to give way to in-weights learning asymptotically, but not the reverse. Furthermore, Singh et al. (2025) show that once a model adopts an in-weights learning strategy, it struggles to recover in-context learning – while the reverse transition remains possible. We observe a similar pattern in humans: participants trained first on skewed data (favoring in-weights learning) fail to adopt in-context learning later, but those trained first on uniform data (favoring in-context learning) can shift strategies. These findings suggest that early learning conditions constrain later flexibility.” (l. 608)

On the human experiments

1. If I understand correctly, the distractor labels and images in the context are sampled uniformly and independently. Thus, the humans will have substantial exposure to *incorrect* image->label pairings in the contexts if they bother to look. It seems to me that it would be more natural to have correct image->label pairs for the distractors that simply don't match the target image. Was there a motivation for this design choice? It might be worth discussing in the text.

We apologize for the confusion caused by the wording in the original Methods section. To clarify: the “distractor image → label” pairs were indeed correct pairings; they simply did not correspond to the target image on that trial. We have revised the Methods section to make this explicit:

~~“The other six labels in the context were sampled uniformly between 0 and 9.”~~

“The other six labels in the context also followed the same rule: each was located three steps clockwise from its corresponding context image.” (l. 692)

2. A question related to that: it wasn’t entirely clear to me from the text whether the mapping from images → labels was semantically meaningful (e.g. all bears get label 1) or arbitrary (i.e. each image is arbitrarily assigned a label independently). I assume it is likely the latter or people would readily generalize from in-weights knowledge, but might be worth stating more explicitly (unless I missed it somewhere).

The mapping between images and labels was indeed arbitrary. We have added a sentence in the Methods section:

“The mapping between images and labels was arbitrary and not semantically meaningful.” (l. 695)

3. One question I had regarding the shared induction mechanism is whether the results could be otherwise. Of course the humans will need to look at both the image and its corresponding label if they use in-context information to solve the task. Given that they use in-context information, is there a reasonable alternative strategy that they could take besides the one that they did?

We agree that the observed human strategy is expected given the task design. Our main interest was not so much in the existence of this strategy per se, but in demonstrating that the model also adopts a two-step process – first binding image-label pairs, then applying the rule – and we were pleased to find converging evidence for this same mechanism in humans using the mouse trajectory paradigm. It is always challenging to get participants to explicitly articulate their ongoing strategies, so we believe this direct behavioral evidence remains valuable, even if it confirms a relatively trivial strategy.

4. Was there a motivation for always using “3 steps clockwise” as the rule for the humans, rather than randomizing the distance from image to label across subjects?

The choice of using a “+3 steps clockwise” rule, as well as the number of context images, was based on pilot experiments.

We found that a “+1” rule was too simple, leading to trivial learning. Similarly, a symmetric rule such as “+4” with 8 images resulted in very fast learning, likely due to prior expectations or symmetry biases in human participants. The “+3 steps” rule with 7 context images avoids these priors and yields better learning curves. We have clarified this rationale in the Methods section of the revised manuscript:

“The three steps clockwise rule and use of seven context images were chosen based on pilot data to avoid trivial or symmetry-based rules that led to rapid learning.” (l. 697)

5. I would appreciate seeing the exact instructions the participants received in the materials & methods; for example, it seems to me that what, if anything, was said about the reason for numbers appearing on screen between the images might potentially contribute to the participants succeeding or failing to learn the in-context rule.

We intentionally chose instructions that were as general and minimal as possible, in order to reduce bias and avoid cueing any specific strategy. The exact instructions provided to participants were:

“This task is a learning task. You may have poor performances at the beginning but you will improve over the course of the experiment. On each trial, you will see a sequence of images and numbers. Your task is to press on the correct number on your keyboard, from 0 to 9. The rule determining which number you have to choose is 100% deterministic. This means that once you have discovered the rule, you will have 100% of correct responses.” (l. 669)

We have now included these exact instructions in the Materials & Methods section of the revised manuscript.

Once again, really interesting work.

Best,

Andrew Lampinen

Reviewer #2

Summary

The paper presents results from experiments testing humans and neural networks on a task where images were associated with labels. In the task, two strategies were available: an “in-context” strategy where the label could be inferred from information given in context (by looking at the label that was +3 positions clockwise from the query image), and an “in-weight” strategy where the labels corresponding to each image were memorized. The experiments were designed to study how the training distribution and curriculum affected which of the two strategies was learned. In both humans and transformers, redundancy (i.e., a skewed distribution where some items occurred very frequently) drove in-weight learning: when most trials could be performed by memorizing only a few specific examples, participants/models were not incentivized to make the effort to extract the rule from the context. Conversely, diversity drove in-context learning: when there were too many things to memorize, participants were incentivized to find the simple rule and avoid memorization, and could thereby generalize to novel items. A separate set of experiments found that while humans benefit when the training curriculum emphasized diversity early in training, transformers did not. Finally, analysis of the internal mechanisms used by the transformers and the cursor trajectories from the human participants revealed similar mechanisms used by humans and transformers for the in-context and in-weight strategies: in both, the in-context strategy involved attending to the labels +3 positions clockwise from the query image, while the in-weight strategy involved attending only to the query image itself.

Review

Overall, we thought this was a strong paper that has the potential to make an important bridge between ongoing research in AI on LLMs and existing work in cognitive psychology. However, our main concerns are that 1) the types of in-context learning and in-weight learning studied here just barely correspond to those studied in LLMs – i.e., they are so simple that they may not actually be relevant to the advanced abilities of modern LLMs, and 2) it is unclear what theoretical contributions the findings make to cognitive psychology (i.e., without the transformer experiments one might have made the same predictions linked to existing dual-process theories). The conclusions of the paper would be better substantiated if more advanced forms of ICL/IWL were investigated, and/or if deeper connections to existing literature in cognitive psychology were discussed in much more detail.

Strengths

1. The paper explores an important and interesting topic, bridging cognitive science / psychology and AI. In-context vs. in-weight learning has become an important area of research in NLP, and there are many interesting questions about how these processes can be related to human learning and reasoning. This paper has potential to make an important contribution in this area, especially to anyone who has been thinking about what, if anything, the advanced in-context learning abilities of AI systems (LLMs) have to do with analogous abilities in human cognition.

2. The experiments were very well executed. The humans and transformers can be very cleanly compared. This is a great example of how to do research at the intersection of cognitive science and AI.

3. The diversity of methods used in the experiments is also impressive. The behavioral experiments and analyses were very well complemented by the transformer simulations, and the mechanistic interpretability study was very well complemented by the analysis of the cursor trajectories (Figure 5).

Weaknesses

1. Although the methodology seems sound and the experiments are well-grounded in ongoing research on LLMs (the experimental paradigm directly replicates an existing paradigm used in work on ICL in transformers), the results are not particularly surprising or diagnostic of ICL vs IWL specifically (see specific comments 2 and 3 below for more details).

- The basic finding that the distribution of examples determines the strategy learned by humans is intuitive, and seems like it has more to do with the affordances of the inputs than with architecture per se (see below).
- The findings about curriculum effects were more interesting, but the finding that neural networks don't do very well when samples aren't presented i.i.d. (due to catastrophic interference) is very well established (see below).

2. Given that most of the results were relatively intuitive, it is unclear what theoretical significance they have for cognitive psychology / neuroscience. For example, various studies suggest that when rules can be extracted over trials (e.g., in category learning or in reinforcement learning etc) people can discover them to generalize across related examples. Conversely, if the same response is almost always correct, there is no need or incentive to discover or apply any rule. There are many multiple-system computational theories across cognitive neuroscience that are relevant here, e.g., episodic vs. procedural memory, complementary learning systems, model based vs model free learning, state abstraction, structure learning vs statistical learning, rapid learning vs. later retention under varying memory demands, and the costs and benefits of engaging in cognitive effort... A broader discussion of relevant literature from cognitive science / neuroscience would help to further connect these recent phenomena observed in AI systems with existing theories about human cognition.

- Relatedly, we are skeptical about how relevant the current findings would be to education research (as the authors mentioned in the discussion), as the task used in the experiments is very simple and seems far removed from any learning scenario in the real world.
- The results could also be interesting to researchers outside of psychology/neuroscience (in AI), but as mentioned above, we also have some minor reservations about how representative these minimal forms of ICL/IWL are when compared to the broader capabilities exhibited by LLMs (see note 2 below).

We thank Reviewer #2 for these thoughtful and constructive comments on our manuscript. We have done our best to incorporate all their remarks and have revised the manuscript accordingly to address each of them in detail.

In their overall summary, Reviewer #2 makes two general points, in addition to the specific issues raised below. These are (1) that our results may not scale to LLMs, and (2) that our results might have been predicted from dual process theories. We agree that these are issues that we could have engaged with better in the initial submission, and we have endeavoured to do so in the revised manuscript:

(1) Scaling.

We agree that the relationship to large-scale models is important. While our study focuses on small, controlled models, we find that the effects we report are robust to variations in model size and task:

- Across a wide range of architectures: from 2 to 10 layers, with 1 to 4 attention heads per layer, with or without intermediate MLPs (see our response below and the new **Fig. Supp. 11**),
- Across tasks, including the new transitive inference task (see our response below and the new **Fig. Supp. 9**).

We did not observe any qualitative change in behavior with scale or task.

We agree that it would be interesting to scale this to much larger models, but this is beyond the scope of the current project. We do note, however, that prior work showed that core inductive behaviors observed in small transformer models often generalize to LLMs (e.g., Chan et al., 2022b; Olsson et al., 2022; Wortsman et al., 2023). Thus, while LLMs involve different objectives and training data, these results suggest that the effects we report are very unlikely to be fragile artifacts of a narrow setting.

Our goal here is not to capture the full behavioral complexity of LLMs, but rather to isolate how properties of the training distribution shape the learning behavior in a controlled setting. To acknowledge these limitations, we have added a sentence to the Discussion:

“While our results are based on relatively small transformer models trained from scratch, prior work suggests that many such behaviors generalize to larger-scale settings (Chan et al., 2022b; Olsson et al., 2022; Wortsman et al., 2023). We nonetheless caution that scaling and pretraining introduce additional factors that may alter the dynamics of learning strategy selection.” (l. 566)

(2) Dual systems.

We agree that better connections could be made to dual-process theories in psychology and neuroscience. We have now referenced these connections in the introductory paragraph.

(Introduction)

“This dichotomy prefigured seminal “dual-process” frameworks in psychology and neuroscience, which separated heuristics from rational computation (Stanovich and

West 2000), information integration from rule-based categorisation (Ashby and Maddox 2005), associative from symbolic processes (Sloman 1996), and model-free from model-based reinforcement learning (Dolan and Dayan 2013).” (l. 43)

However, we would push back on the idea that our main findings follow naturally from dual-system theories. Dual-process theories do not make predictions about how learning strategy should vary as a function of the statistical properties of the training data. In contrast, our work shows that both humans and transformer models adapt their learning strategy depending on the training data distribution. This kind of dynamic arbitration falls outside the scope of classical dual-process models. We have added this clarification in the Introduction:

(Introduction)

“While the distinction between in-context and in-weights learning is reminiscent of dual-process frameworks, it is important to note that classical dual-system models do not make specific predictions about how learning strategy should vary with the statistical structure of the training data. This is the central focus of our work.” (l. 71)

(3) Implications for education research

Finally, we also agree that the implications for education research were somewhat overstated. We have now caveated these claims more carefully in the discussion:

(Discussion)

“This speaks to a longstanding debate in education research, which has asked whether schools should emphasise rote learning or critical thinking ⁴². The answer implied by our data is that both are important. Presenting diverse examples that teach students how to tackle new problems is crucial, but being able to retrieve information about past experiences requires repetition. Of course, we cannot know whether insights from the simple, stylised setting employed here would translate to the classroom, but at least our work sets up a hypothesis that could be tested in more translational settings.” (l. 596)

Specific comments

1. Title: In our opinion the title is too vague – humans almost certainly “learn like transformer networks” in some ways, and certainly don’t in other ways. A more informative title would mention the specific similarities / dissimilarities found in this particular study. In general some people in the field tend to overgeneralize on these kinds of generic questions, so specificity should be prioritized wherever possible.

Thanks – good suggestion. At the Editor’s discretion, we propose changing the title to “*Shared sensitivity to data distribution during learning in humans and transformer networks*”, which we hope avoids any overgenerality.

2. The in-context vs. in-weight learning studied here is quite minimal. In-context learning is critical to current AI systems, as it allows them to be deployed for many different purposes without retraining/finetuning. This is why understanding how it is accomplished in LLMs has become such a big topic in AI/NLP. However, the minimal cases of ICL that are used in some

of that work (including in the task presented here), are often so simple as to be unrepresentative. Is it really “in-context learning” to simply apply a known (+3 positions) rule – discovered over previous examples – to a specific set of stimuli? This is more akin to any of the other frameworks mentioned above studying rule-based systems in category learning, RL and generalization, etc. So while this task technically also “counts” as involving a minimal form of ICL, the more impressive form of ICL attributed to transformers and humans involves the ability to infer new rules based on a few examples (or instructions) given in context rather than applying a single known rule learned from many previous instances. Similarly, the IWL studied here is very minimal: in the case of the extremely skewed distribution, using a single response would achieve 92% accuracy (see 3b below).

We thank the Reviewer for raising this point, which led us to design and test a new paradigm to check whether our findings generalise beyond the original task.

Our choice to use a minimal task was deliberate: it provided a tightly controlled framework in which we could cleanly isolate and causally manipulate the contributions of in-context learning and in-weights learning, and apply mechanistic interpretability tools to small transformer models.

That said, we agree with the Reviewer that our paradigm is relatively minimal compared to more complex forms studied in large language models, where new rules or mappings are inferred from limited demonstrations. To address this concern and test the generalizability of our findings, we conducted a new modeling experiment using a transitive inference task. In this task, the model must infer a relation such as $A > C$ from contextual demonstrations like $A > B$ and $B > C$, with stimuli (A, B, C) drawn from a large pool of exemplars. Transitive inference is a paradigmatic example of structure learning, and a challenge that requires observers to make inferences about entirely novel queries. Our new results confirm that:

- Transformer networks are able to solve this task via in-context learning.
- The same pattern of sensitivity to the training distribution holds: training with diverse examples supports in-context learning ($\alpha < 1$), while redundant training promotes in-weights learning ($\alpha > 1$).

These findings support the idea that the relationship between data diversity/redundancy and learning strategies is robust across tasks, even those requiring more abstract forms of generalization. We now include this extension and its implications in the revised manuscript:

Figure Supp. 9. Modelling results in a transitive inference task. **A.** We replicated our modelling results in a distinct task probing transitive inference. As in the image-label association task (Fig. 1), we manipulated the distribution of the training data: under a uniform distribution ($\alpha = 0$), all environments are equally likely; under skewed distributions ($\alpha \gg 0$), some environments are more frequent. Each environment consisted of six images ordered along an underlying dimension. **B.** Example training trial. The context presented ten triplets, each comprising two images and a symbol, corresponding to all one-step comparisons within a given environment (e.g., “image 4 > image 3”). The query consisted of a two-step comparison (e.g., “image 4 ? image 2”), and the model had to select the correct relational symbol (“>” or “<”). **C.** Paradigm overview. During training, two learning strategies are available. The “in-context” learning strategy consists in using local comparison given in the context to infer the correct relational symbol via transitive inference (e.g., relying on “image 4 > image 3” and “image 3 > image 2” to infer “image 4 > image 2”). The “in-weights” learning strategy consists in learning the association between pairs of images and relational symbols in memory using the feedback. Test blocks were designed to probe which strategy(ies) the model is using. On in-context test blocks, images from novel environments (depicted in grey) were presented, such that the only way to be accurate is to use information from the context, *a.k.a.* the in-context strategy. On in-weights test blocks, a training pair (depicted in blue) was presented as the query pair but images from novel environments (depicted in grey) were presented in the context, such that the only way to be accurate is to use information stored in memory, *a.k.a.* the in-weights strategy. On “arbitrage” test blocks, a trained environment was presented but the order of the images was reversed (e.g., “image 4 < image 3”). **D.** Training and test performances for transformers ($N = 30$ per training data distribution). Small dots are individual transformers, large dots are group average.

(Main text)

“Finally, to test whether our findings generalize to more abstract forms of reasoning, we trained transformers on a transitive inference task. In this task, the model had to

*infer $A > C$ from examples like $A > B$ and $B > C$ presented in the context. As in the main task, performance depended on the training distribution: models trained on a uniform distribution ($\alpha = 0$) solved the task using in-context learning, while models trained on a skewed distribution ($\alpha > 1$) relied on in-weights learning. These results confirm that the link between training distribution and learning strategy holds even in tasks requiring more abstract generalization (see **Fig. S9**).” (l. 524)*

(Methods section)

“We designed a second modeling task to test whether the effects of training distribution on learning strategy generalise beyond the image-label association setup. In this task, each training environment consisted of six unique images, each with an implicit rank. The model received ten training triplets per trial, each expressing a one-step comparison between images (e.g., “image 4 > image 3”), followed by a query that required a two-step transitive inference (e.g., “image 4 ? image 2”).

On each trial, the model received:

- *A context of one-step comparisons between image pairs from a single environment (e.g., “image 4 > image 3”, “image 3 > image 2”).*
- *A query requiring a two-step inference (e.g., “image 4 ? image 2”), where the model had to choose the correct relational symbol (“>” or “<”).*

We manipulated the training distribution by varying the skewness (Zipf exponent α) of how often each environment appeared during training, following the same logic as in our main task. At test, three types of blocks were used:

- *In-context test: the context came from a novel environment, so the only way to respond correctly was to use in-context learning.*
- *In-weights test: the query pair had been seen during training, but the context came from a novel environment; accuracy relied on memorized pair-label associations.*
- *Arbitrage test: environments were reused from training but with reversed item orders (e.g., “image 4 < image 3”), to probe which strategy dominated when in-context and in-weights learning gave conflicting answers.*

*We used the same architecture, training setup, and evaluation metrics as in the image-label association task. The full results are presented in **Fig. S9**.” (l. 863)*

a. Related to the point about the lack of ecological validity w.r.t. real-world AI systems (LLMs): the simple attention-only transformer was presumably used so that the interpretability methods would be cleaner/easier to do (?), but it seems like this could affect results in some cases, as the MLPs in LLMs are known to be involved in memorizing specific facts (and therefore might end up changing the tendency toward the IWL memorization strategy). It might be worthwhile to include some analogous experiments with larger transformers that had MLPs, so we can be more confident the results would generalize to LLMs.

We thank the Reviewer for this suggestion, shared with Reviewer #1. To investigate this further, we repeated the architecture sweep using transformer models with MLPs interleaved between attention layers. This modification did not result in any qualitative

changes in the outcomes – the results closely replicated those obtained with attention-only models.

Figure Supp. 11. Performance of transformers with interleaved MLP with varying architecture sizes. The MLP blocks consist of two dense layers with a ReLU activation, followed by a residual connection and layer normalization. Scatter plots of the in-context vs in-weights test performances for transformers with varying numbers of layers, number of heads per layers, and varying training distributions. Each dot represents a model trained with a specific number of layers, attention heads, and training data distribution. Dot color indicates the α exponent of the training distribution. Dotted lines indicate chance-level performance.

“We also tested an interleaved MLP model (Fig. S11), where each attention layer was followed by a feedforward (MLP) block consisting of two dense layers with D_M units (with ReLU activation), a residual connection, and a layer normalization step.” (l. 831)

This aligns with preliminary tests we conducted that showed that MLPs do not appear to play a critical role in this learning paradigm, which initially motivated our choice to focus on attention-only architectures. We have added a supplementary figure reporting these results and included the following clarification in the main text:

“We tried with different model sizes, and confirmed that it was also the case with larger and deeper models, with and without interleaved feedforward layers between attention layers (up to 4 attention heads per layer, up to 10 layers, see Fig. S10-S11).” (l. 555)

3. Per-item results should be included for in-weights test

a. It is unclear from the way results were presented how well people/models were doing on the in-weights test on the rare examples. It would be informative to also include the results on the in-weights test where items are weighted uniformly rather than weighted according to the skewed distributions. Alternatively, the accuracies could be shown on a per-item basis in a supplemental figure. These would be particularly important for understanding the results related to double learning and in the arbitrage test.

b. If performance is poor for rare examples in the IWL test, then it means that people doing the IWL strategy have barely learned anything at all, and could even be simply responding with the most likely answer in the extremely skewed cases. This is important because if people aren't memorizing a ton of examples, this shows an asymmetry where they're not really choosing to do IWL or ICL in general, but just choose to try hard to find the ICL rule depending on how easily they can get away with memorizing a few examples. Seems like people are just trying to minimize effort overall, and it's easier to memorize 2-3 examples than it is to find the rule, but finding the rule is easier than memorizing if there are more than just a few examples.

We agree this is important to clarify. In response, we now report per-item performance in the in-weights test as a function of item frequency during training, for both transformer networks and human participants (see **Fig. S3**). These results confirm that both systems are sensitive to training frequency: frequent items are learned well, while rare items are not.

To make these patterns interpretable, we grouped test items into frequency bins (e.g., "top 1", "top 2–4", "top 5–10") based on how often each item appeared during training. We chose this approach because averaging over individual items was too noisy to be meaningful – especially for rare items, which were often seen only once or twice per participant or model. Grouping by frequency allowed us to reveal the general trend without overinterpreting unreliable estimates at the item level.

However, even among relatively rare items (e.g., seen fewer than 4 times), human performance remains above chance. In the $\alpha = 2$ condition, the "top 5–10" images were seen 4, 3, 2, 2, 2, and 1 times, yet participants still reached ~50% accuracy. This suggests that participants did learn item-label associations, even with minimal exposure, and were not simply guessing or defaulting to the most common label.

Nevertheless, human performance was at chance for very rare items (e.g., seen 1 time) – which is expected. These findings also match prior results in neural networks: performance on rare items in in-weights tests is also poor in transformers (e.g., see Chan et al., 2022; Reddy, 2023), highlighting a shared bias toward learning what is frequent and falling back to guessing when not.

Figure Supp. 3. Performance as a function of the image frequency during training. Training and test performances for **A.** transformers ($N = 30$ per training data distribution) and **B.** human participants (Exp. 1, $N = 30$ per training data distribution) as a function of the frequency of the image during training. For each value of α , test items were grouped by how often they appeared during training. For example, in $\alpha = 2$: “top 1” corresponds to the image that was seen 92 times during training, “top 2–4” to images that were seen ~ 13 times, and “top 5–10” to images that were seen ~ 2 times. Large dots are group average. Errors are s.e.m.

We also replicated this analysis on the newly acquired dataset in response to a comment by Reviewer #3 and found the same results (see Fig. S4).

Figure Supp. 4. Replication of Experiment 1. [...] B. Training and test performances as a function of the frequency of the image during training. [...]

We have added a sentence in the main text:

“To better understand what drives performance in the in-weights test, we analyzed accuracy as a function of item frequency during training (see Fig. S3-S4). Both transformer networks and human participants performed better on frequent items, confirming that they learned from repeated exposure.” (l. 232)

4. Results about differences in architecture are unconvincing

a. The humans, like the transformer, had access to the entire context when considering the query. But for the recurrent neural network to learn the in-context strategy, it would have to learn to maintain all of the contextual information until the end of the sequence, where it could (upon seeing the query) retrieve the label presented 3 positions

after the query in the context. I wonder if the recurrent network would be more likely to learn the ICL strategy if it got the query before the in-context examples, so that it would know what to look for as the context examples were shown – or if all items were presented simultaneously at each time point (with some kind of positional encodings). Conversely, I wonder how often humans would learn the ICL strategy if they had to remember all of the item-label pairs in the context before receiving the query. It seems unlikely that they would ever learn such a strategy if many examples were included in the context.

We agree that maintaining all contextual information until the end is likely a challenge for LSTMs in this task. To test this idea directly, we trained a query-first LSTM variant, where the query was presented before the context examples. This setup allows the model to process the context with the query already in memory, which might help guide attention to relevant information. However, as shown now in Fig. S2, even in this setup the LSTM did not learn the in-context strategy. Its performance remained similar to the standard LSTM that saw the query last. This suggests that, under our training regime and architecture size, LSTMs struggle to adopt the ICL strategy regardless of sequence order. We have added a sentence in the Methods:

“The standard LSTM received inputs one item at a time, with the query presented last, matching the setup used for transformers and human participants. We also tested a query-first LSTM variant, where the query appeared at the start of the sequence, followed by the context items. This was designed to test whether knowing the target query early would help the model focus on relevant context and learn an in-context strategy. Despite these variations, none of the models showed reliable in-context learning (see Fig. S2).” (l. 854)

Figure Supp. 2. Feed-forward networks and LSTM networks do not become in-context learners in the same task. A. (top) 2-layer feed-forward fully-connected network. (bottom) Scatter plots of the in-context vs in-weights test performances after

training. **B.** (top) 2-layer LSTM network. (bottom) Scatter plots of the in-context vs in-weights test performances after training. Each dot is an individual network.

b. As for the feedforward network, did it receive anything like positional encodings? If not, then it would have no way of generalizing a +3 rule, and would have to re-learn this rule separately for each position in the sequence.

We confirm that all models (transformers, feedforward, and LSTM networks) received positional encodings in their inputs. We now state this clearly in the Methods section:

“All models were trained on the same data and evaluated using the same procedure as the transformer, including positional encodings in their input representations.” (l. 850)

c. This is all just to say that I think the inference that humans “learn like transformers” (compared to RNNs or MLPs) would be premature in this case, as I would guess that the basic results about ICL vs. IWL depend more on the way that the inputs are presented (to both the RNN/MLP and to the humans) than on the architecture itself.

We agree with the Reviewer that our original wording may have overstated the similarity between humans and transformers, and risked confusion. Our intent was not to claim that humans “are like transformers and transformers only” but rather that transformers are a particularly effective model system for studying in-context learning – and in this case, show striking behavioral similarities to human learners.

We acknowledge that architecture is not strictly necessary for ICL. Indeed, recent studies support this idea. For example:

- Lee et al. (2024) showed that LSTMs can learn ICL-like behavior in very specific regimes – particularly when burstiness is high ($p_{\text{bursty}} = 1$) and the task complexity is low (2 classes). But performance drops significantly with more classes: at 8 classes, LSTMs no longer show reliable ICL. Our task involves 10 labels, putting it beyond that threshold.
- Similarly, Tong & Pehlevan (2025) showed that MLPs can display ICL under certain training setups, but again, transformers outperform them – especially when the burstiness and the number of clusters decreases (“[...] transformers transition to ICL for lower burstiness and lower number of clusters k ”). Their results highlight a general trend: transformers shift to ICL more easily, and require less extreme burstiness or fewer training tricks to do so.

We don’t claim that only transformers can learn in-context, but rather that transformers learn ICL strategies more robustly, reliably, and flexibly than LSTMs or MLPs under comparable conditions. We’ve now clarified this point in the manuscript:

(Results)

“It should be noted that in-context learning is not exclusive to transformer networks – under specific conditions, both feed-forward and recurrent architectures like LSTM networks can learn in-context^{30,31}. However, transformers adopt this strategy more

robustly and flexibly across a wider range of settings, including those used in our study.” (l. 171)

(Discussion)

“Our work compares humans and transformer networks. We find that in one interesting respect – the emergence of in-weights learning and in-context learning in response to the training data distribution – they show some striking similarities. However, this should not be taken to imply overlap between humans and transformers at the algorithmic level. Indeed, other classes of neural network, including simple multi-layer perceptrons, may in principle be capable of in-context learning^{34,35}. Transformers are feedforward networks with a highly structured architecture based on self-attention, diverging sharply from the recurrent, feedback-driven, and biologically grounded computations of the human brain. Nevertheless, the way that they trade off memory-based strategies and inference-based strategies exhibits surprising commonalities with how this happens in human cognition.” (l. 621)

5. Curriculum effects

a. The curriculum effects were interesting, but it is unclear whether any conclusions can be drawn other than the well-established fact that neural networks suffer from catastrophic interference when trials are blocked over time. For example, if items from previous blocks were mixed into later blocks (simulating a kind of episodic replay), do the networks become double learners more often with one of the two curricula?

We agree. Catastrophic interference is a well-known issue in neural networks, and we don't present this as a new finding. Our goal here was to temper the comparison between humans and transformers, and make clear to the reader that they are not the same. The curriculum results show one area where humans are more robust: they benefit from structured training, while transformers do not. We now make that point more clearly in the Discussion:

“Despite these striking similarities, transformers did not benefit from curricula that prioritised either diversity or redundancy in examples, whereas humans clearly did. This difference likely reflects a well-known limitation of neural networks: catastrophic interference. Once transformers settle on a strategy, they often forget earlier information – especially when training is blocked. In humans, early diversity boosts generalisation, even when redundancy comes later. In transformer networks, later training tends to overwrite earlier strategies, making them less flexible to curriculum structure.” (l. 571)

We haven't tested episodic replay or mixing items across blocks, but agree that would be a useful follow-up – especially to see if it helps transformers become better double learners under curriculum constraints.

b. We were a little confused about why transformers learn the in-weight strategy at all when trained on the C1 curriculum. If they've learned the in-context strategy after the uniform training, shouldn't they be getting 0 loss on the skewed blocks from the start, and

therefore shouldn't learn anything during those latter blocks? It might be informative to include the training curves over the blocks – did the models converge to 0 loss before the next block was started?

We have now included training curves in **Fig. S6** to clarify. These curves show that when training begins with the uniform distribution ($\alpha = 0$, as in C1), the model's loss indeed drops to near zero by the end of that first phase – indicating that the in-context strategy has been successfully learned. However, when training continues on the skewed distribution, the model eventually shifts toward the in-weights strategy. This switch does not happen immediately, and we believe it is triggered by small fluctuations or rare errors during the skewed phase. These local updates, though small, gradually steer the model toward in-weights learning.

Figure Supp. 6. Transformers do not benefit from structured curricula. **A.** Test performances over the course of training of transformers trained on C1 (red) and C2 (blue). Bold lines are group average ($N = 20$ transformers per curriculum). Arrows were manually added to emphasise the direction of the trajectories. **B.** Accuracy curves for one example transformer network trained on C1. **C.** Accuracy curves for one example transformer network trained on C2.

c. Relatedly, it could be worthwhile to include the accuracy results over the course of training the humans in the experiments that used blocked curricula.

We have now added these results in **Fig. S7B**. Interestingly, accuracy increases steadily in curriculum C1. In C2, performance drops sharply when the distribution switches from skewed to uniform. This supports our interpretation that early exposure to diversity helps generalization, while starting with redundancy may lead to overfitting.

Figure Supp. 7. Performance of human participants in all curricula. **A.** Four groups of human participants (Exp. 3, $N = 50$ per group) were exposed to a composite distribution ($P_c = 0.5$, $\alpha_s = 2$) with different training curricula, *i.e.* different block order, denoted C1 to C4 (“uniform”, $\alpha = 0$; “skewed”, $\alpha_s = 2$). **B.** Performance during training per curriculum. **C.** Double learning index per curriculum. **C.** Training and test performances for humans per curriculum. A curriculum that promotes learning first in-context and then in-weights improves the in-context performance without impairing in-weights learning. Small dots are individuals, large dots are group average.

6. A few minor points about some of the language used in the introduction:

a. Is “relying on social norms” a good example of an inductive inference? This seems pretty far removed from the kind of inductive inference required in the current study.

We have removed the sentence and modify the example to:

“[...] for example, using the laws of calculus to compute integrals on a maths exam, or applying grammar rules to understand a sentence never heard before.” (l. 37)

b. Unclear what “inextricably intertwining computation and memory” means: for example, one way of disentangling “computation” and “memory” in neural networks would be in the distinction between in-context and in-weight learning...

We agree that the phrase was unclear. We have removed it from the text.

c. “in-context learning is a kind of metalearning” – it may be more apt to think of in-context learning as a result of metalearning (learning how to learn in context).

We have modified to:

“Rather than relying on weight updates, in-context learning arises from the networks’ internal processing: it is best understood as an emergent result of meta-learning, where training leads the network to “learn how to learn” from the structure of its input, enabling it to perform few-shot learning without updating its weights.” (l. 59)

d. The transformer isn't an autoregressive algorithm per se -- transformers can also be trained on different non-autoregressive language modeling tasks (e.g., masked language modeling, translation).

The transformer also isn't technically a learning algorithm at all – it's an architecture whose parameters can be trained by different learning algorithms (gradient descent / backprop).

e. This sentence is slightly incorrect and misleading: "transformer...uses a feedforward neural network to learn the relative importance (self-attention) among tokens and positions in a sequence". Being a bit more descriptive might help readers who are not familiar with the transformer architecture.

We have modified to:

"[...] neural network architecture known as the transformer. Transformer networks use self-attention to compute how much each token in a sequence should influence the representation of every other token. This allows the model to integrate information across positions and build context-aware representations at each layer" (l. 64)

7. What does "fully informative feedback" mean? Was the context still present when feedback was given? It would probably be important for people to see the feedback in the presence of the context for them to infer the +3 rule.

"Fully informative feedback" means that participants were shown both whether their response was correct or incorrect and what the correct response was (e.g., "incorrect/the correct answer was 3"). We also confirm that the context remained visible on screen during the feedback period. We have clarified this in the Methods section:

"Stimuli remained visible on screen during feedback." (l. 684)

"Fully informative feedback was provided during training: after each trial, participants were shown whether their response was correct or incorrect, as well as the correct response." (l. 699)

8. We didn't see the accuracy results from the in-context and in-weights tests from Experiment 4 reported anywhere. Were the basic findings related to ICL and IWL replicated in the case where the display was blurred/obscured?

We have added the accuracy results in **Fig. S8C** (keeping the x-axis similar to the one in **Fig. 1** to ease the comparison). The basic findings do replicate, as indicated in the main text.

"First, we confirmed that the participants trained on $\alpha = 0$ became in-context learners, whereas the participants trained on $\alpha = 2$ did not, replicating once again the results of Experiment 1. Indeed, the training data significantly influenced performance on in-context test trials (effect of α on accuracy, $\beta = -2.437 \pm 0.636$, $p = 0.0$, $BF = 44.3$, "strong" evidence), in-weights test trials (effect of α on accuracy, $\beta = 2.262 \pm 0.129$, $p = 0.0$, $BF > 100$, "decisive" evidence), and arbitrage test trials (effect of α on accuracy with respect to in-context learning, $\beta = 2.002 \pm 0.264$, $p = 0.0$, $BF > 100$, "decisive" evidence). As in Experiment 1, we confirmed that the training distribution did not

directly influence the performance at the end of training ($\beta = -0.09 \pm 0.408$, $p = 0.824$, $BF = 0.027$, “strong” evidence) but only the strategy used by the participants.” (l. 493)

Figure Supp. 8. Cursor trajectories and performances of participants in all test blocks (Exp. 4, N = 20 per group). A. Trajectories for participants trained on a uniform distribution ($\alpha = 0$) in the (left) in-context test block, (middle) in-weights test block and (right) arbitrage block. **B.** Same for participants trained on a skewed distribution ($\alpha = 2$). Trajectories were aligned trial-by-trial to a common frame where the target image is located on the top of the context circle. Small lines are individual average trajectories, diamonds are group average trajectories. **C.** Training and test performances. Small dots are individual, large dots are group average.

Please also note that, in response to a comment by Reviewer #3, we now provide a full pre-registered replication (AsPredicted #231356, <https://aspredicted.org/rqgz-rdfk.pdf>) of Experiment 1 that also replicates our basic findings relative to ICL and IWL (see Fig. S4).

Figure Supp. 4. Replication of Experiment 1. **A.** Training and test performances of human participants (bottom, replication of Exp. 1, $N = 30$ per training data distribution). Small dots are individuals, large dots are group average. Our pre-registered effects (AsPredicted #231356, <https://aspredicted.org/rqgz-rdfk.pdf>) were all verified. In particular, there was a negative effect of α on accuracy in in-context test block ($\beta = -1.145 \pm 0.208$, $p = 0.0$, $BF > 100$, “decisive” evidence), a positive effect of α on accuracy in in-weights test block ($\beta = 1.786 \pm 0.138$, $p = 0.0$, $BF > 100$, “decisive” evidence), a negative effect of α on accuracy with respect to in-context learning in arbitrage blocks ($\beta = -1.097 \pm 0.168$, $p = 0.0$, $BF > 100$, “decisive” evidence), and a positive effect of α on accuracy with respect to in-weights learning in arbitrage blocks ($\beta = 1.669 \pm 0.128$, $p = 0.0$, $BF > 100$, “decisive” evidence). **B.** Training and test performances as a function of the frequency of the image during training. **C.** Scatter plots of the in-context vs in-weights test performances. Each dot is an individual model/human.

Formatting issues / typos

- The final supplementary figure says “2 heads” for the feedforward network and LSTM. I assume this is an error and those architectures didn’t have any attention heads?
- It might be nice to include Experiment X labels as section titles, or at least in the figures to help the reader navigate.
- Figure 1 caption: 'the only to be accurate is to use information...' "Corpus" -> "corpora"? (or is this a britishism?)
- I think this one occurred multiple times: "trained on skewed distribution", "trained on uniform distribution"

- "performance at training"?

- "the computational underlying"

- Fig 4C: the solid arrows are confusing to follow. The pattern of movement could be made more clear in the visualization as well as in the caption.

-- Discussion section – “Recent theoretical work has ... solutions”. The sentence seems ill-formed and less clear. Possibly elaborate to make it more clear.

-- “in weighs” typo Fig 1C caption

We have corrected the typos.

Reviewer #3

This paper compares human and transformer learning. It is found that both agents are sensitive to the training environment; skewing the distribution makes them more in weight learners, a uniform distribution makes them more in context learners.

This paper presents an interesting comparison between the two agents. The authors have the tendency to overstate the novelty of the findings (e.g., that transformers can learn in context and in weight; or that they suffer from catastrophic interference whereas humans do not), but the systematic comparison and human experimentation is interesting. Some comments are the following.

1. One finding is that MLPs and LSTMs don't learn in context. But how sure are you of this null finding? How well did you check the parameter space to confirm this fact? I see that the number of parameters is matched between the agents, but this is not necessarily the best matching. Did you check the parameter space as thoroughly as for the transformer? It would be worthwhile to explore the parameter space much more thoroughly than done now; currently I have no idea how general the finding is. And in fact, the same comment holds for the transformer: How broad is the finding that it can learn in context and in weights (across parameter settings)?

We agree with the Reviewer that our original wording may have overstated the similarity between humans and transformers, and risked confusion. Our intent was not to claim that humans "are like transformers and transformers only" but rather that transformers are a particularly effective model system for studying in-context learning – and in this case, show striking behavioral similarities to human learners.

We acknowledge that architecture is not strictly necessary for ICL. Indeed, recent studies support this idea. For example:

- Lee et al. (2024) showed that LSTMs can learn ICL-like behavior in very specific regimes – particularly when burstiness is high ($p_{\text{bursty}} = 1$) and the task complexity is low (2 classes). But performance drops significantly with more classes: at 8 classes, LSTMs no longer show reliable ICL. Our task involves 10 labels, putting it beyond that threshold.
- Similarly, Tong & Pehlevan (2025) showed that MLPs can display ICL under certain training setups, but again, transformers outperform them – especially when the burstiness and the number of clusters decreases ("[...] transformers transition to ICL for lower burstiness and lower number of clusters k "). Their results highlight a general trend: transformers shift to ICL more easily, and require less extreme burstiness or fewer training tricks to do so.

We don't claim that only transformers can learn in-context, but rather that transformers learn ICL strategies more robustly, reliably, and flexibly than LSTMs or MLPs under comparable conditions. We've now clarified this point in the manuscript:

"It should be noted that in-context learning is not exclusive to transformer networks – under specific conditions, both feed-forward and recurrent architectures like LSTM

networks can learn in-context^{30,31}. However, transformers adopt this strategy more robustly and flexibly across a wider range of settings, including those used in our study.” (l. 171)

Please also note that concerning the generality of the findings on MLP and LSTM, in response to a comment made by Reviewer #2, we have also tried LSTM with the query placed before the context sequence (see **Fig. S2**):

“The standard LSTM received inputs one item at a time, with the query presented last, matching the setup used for transformers and human participants. We also tested a query-first LSTM variant, where the query appeared at the start of the sequence, followed by the context items. This was designed to test whether knowing the target query early would help the model focus on relevant context and learn an in-context strategy. Despite these variations, none of the models showed reliable in-context learning (see **Fig. S2**).” (l. 854)

Figure Supp. 2. Feed-forward networks and LSTM networks do not become in-context learners in the same task. A. (top) 2-layer feed-forward fully-connected network. (bottom) Scatter plots of the in-context vs in-weights test performances after training. **B.** (top) 2-layer LSTM network. (bottom) Scatter plots of the in-context vs in-weights test performances after training. Each dot is an individual network.

Concerning the generality of our findings in transformer networks, we provide a full architecture sweep in **Fig. S10**. In response to comments from Reviewers #1 and #2, we also tested transformer models with interleaved MLP blocks in **Fig. S11** (i.e., standard transformer encoder layers). These models showed the same qualitative behavior, further confirming that our results are robust across architectural variants:

Figure Supp. 11. Performance of transformers with interleaved MLP with varying architecture sizes. The MLP blocks consist of two dense layers with a ReLU activation, followed by a residual connection and layer normalization. Scatter plots of the in-context vs in-weights test performances for transformers with varying numbers of layers, number of heads per layers, and varying training distributions. Each dot represents a model trained with a specific number of layers, attention heads, and training data distribution. Dot color indicates the α exponent of the training distribution. Dotted lines indicate chance-level performance.

2. $N = 30$ in a between-subjects design is small. Is there an a priori power analysis to justify this sample size?

This sample size was originally chosen based on pilot data. To fully address this concern, we conducted a complete pre-registered replication of Experiment 1, with sample size determined from a power analysis based on the original dataset. The pre-registration is available here: AsPredicted #231356, <https://aspredicted.org/rqgz-rdfk.pdf>. The replication fully confirmed our original findings and supported all preregistered hypotheses. Results are reported in **Fig. S4**:

Figure Supp. 4. Replication of Experiment 1. **A.** Training and test performances of human participants (bottom, replication of Exp. 1, $N = 30$ per training data distribution). Small dots are individuals, large dots are group average. Our pre-registered effects (AsPredicted #231356, <https://aspredicted.org/rqgz-rdfk.pdf>) were all verified. In particular, there was a negative effect of α on accuracy in in-context test block ($\beta = -1.145 \pm 0.208$, $p = 0.0$, $\text{BF} > 100$, “decisive” evidence), a positive effect of α on accuracy in in-weights test block ($\beta = 1.786 \pm 0.138$, $p = 0.0$, $\text{BF} > 100$, “decisive” evidence), a negative effect of α on accuracy with respect to in-context learning in arbitrage blocks ($\beta = -1.097 \pm 0.168$, $p = 0.0$, $\text{BF} > 100$, “decisive” evidence), and a positive effect of α on accuracy with respect to in-weights learning in arbitrage blocks ($\beta = 1.669 \pm 0.128$, $p = 0.0$, $\text{BF} > 100$, “decisive” evidence). **B.** Training and test performances as a function of the frequency of the image during training. **C.** Scatter plots of the in-context vs in-weights test performances. Each dot is an individual model/human.

(Methods)

“The sample size for the replication of Experiment 1 was determined via power simulations based on data from Experiment 1, assuming a 50% smaller effect size than observed in that study. The simulations suggested a minimum of 10–20 participants per group, depending on the block. To ensure robust power across all analyses, we conservatively set the sample size to 30 per group.” (l. 920)

(Results)

“To confirm the robustness of our findings, we conducted a pre-registered replication of Experiment 1 with a new sample of human participants ($N = 30$ per training distribution, preregistration available at AsPredicted #231356, <https://aspredicted.org/rqgz-rdfk.pdf>). All key effects were replicated (see Fig. S4), including the tradeoff between in-context learning and in-weights learning as a function of the training distribution.” (l. 267)

Finally, please also note that Experiment 4 is also replicating these findings. We have now made that clearer by adding a panel in **Fig. S8C** and adding a paragraph in the Results section:

“First, we confirmed that the participants trained on $\alpha = 0$ became in-context learners, whereas the participants trained on $\alpha = 2$ did not, replicating once again the results of Experiment 1. Indeed, the training data significantly influenced performance on in-context test trials (effect of α on accuracy, $\beta = -2.437 \pm 0.636$, $p = 0.0$, $BF = 44.3$, “strong” evidence), in-weights test trials (effect of α on accuracy, $\beta = 2.262 \pm 0.129$, $p = 0.0$, $BF > 100$, “decisive” evidence), and arbitrage test trials (effect of α on accuracy with respect to in-context learning, $\beta = 2.002 \pm 0.264$, $p = 0.0$, $BF > 100$, “decisive” evidence). As in Experiment 1, we confirmed that the training distribution did not directly influence the performance at the end of training ($\beta = -0.09 \pm 0.408$, $p = 0.824$, $BF = 0.027$, “strong” evidence) but only the strategy used by the participants.” (l. 493)

Figure Supp. 8. Cursor trajectories and performances of participants in all test blocks (Exp. 4, N = 20 per group). [...]. **C.** Training and test performances. Small dots are individual, large dots are group average.

3. For C3, C4, it's weird that for the comparisons where a null difference is “convenient”, the severe Bonferroni correction is applied. I would use the exact same procedure as for the other contrasts.

Our initial reasoning was that we should not use the same procedure for pre-registered contrasts and non-pre-registered contrasts, since pre-registration protects against biases linked to p-hacking. For the (20) non-pre-registered contrasts, we found that not applying Bonferroni correction led to some significant p-values with Bayes Factors (BF) below 1, which strongly suggests false positives.

Following the Reviewer’s comment and to ensure a more systematic and conservative approach, we now apply Bonferroni correction to all pairwise contrasts, including those that were pre-registered:

(Methods)

“We applied a Bonferroni correction to correct for multiple comparisons.” (l. 916)

To maintain transparency, we now also include a table (**Table S1**) reporting all uncorrected p-values and Bayes Factors. This allows readers to inspect the raw statistical evidence.

“Table Supp. 1. Complete results of the pairwise comparisons between curricula (Experiment 3). Each row reports the effect size (β), standard error (SE), uncorrected p-value (p), and Bayes Factor (BF). Bonferroni-corrected p-values are used in the main text.” (l. 1081)

4. It’s interesting to see how the transformer solves the problem. But it’s an overstatement to claim that this is how humans do it. First, if the transformer is trained on +3, it likely cannot generalize at all (?) to +2 or +4. If it is then trained on +2 or +4, it likely does not generalize at all from its previous knowledge. Second, when going back to the +3 task, it likely then has to learn again from scratch. I’d be happy to see this intuition confirmed (or more interesting, refuted!). If confirmed, it leaves me thinking about what has been shown precisely in the paper.

Based on the comments of Reviewer, we conducted a new set of simulations in which transformer networks were trained on alternating rules (e.g., +2 → +3 → +2 steps). The results, shown in **Fig. R#3–1**, confirm the Reviewer’s intuition: transformers do not generalize between rules, nor do they recover previously learned strategies once overwritten. After switching from +2 to +3, performance drops and the model must relearn the new mapping from scratch. When returning to the +2 rule, the model shows no memory of prior training and must relearn it again. This confirms that while transformers can do in-context learning, their ability to flexibly switch between strategies is very limited – a key difference from human learners, who often show robust transfer and reinstatement in similar settings.

Figure for Reviewer #3–1. Training dynamics of transformers on alternating tasks. Each curve shows average training performance of transformer networks ($N = 20$ per condition). Each line shows performance across these alternating training phases, with task switches occurring every 30,000 steps. The graph reads as follows: on the left, transformer networks were trained for 30,000 steps on the +2 steps task, then for 30,000 steps on the +3 steps task, and finally for another 30,000 steps on the +2 steps task. Bold curves are group averages; shading shows \pm s.e.m.

We agree this highlights a meaningful limitation of transformer models compared to humans – in particular, their lack of flexibility once a strategy has been learned. However, our goal in this work is not to make claims about flexible task-switching. Instead, we focus on how the structure of the training data influences the emergence of in-context versus in-weights learning strategies. In that respect, we believe our findings provide a useful step toward understanding how different learning regimes shape behavior in both biological and artificial systems.

5. In Exp 4, I find the mouse paradigm a clever idea, but the data in Fig 4B is misleading. The relevant comparison is the one in Suppl 4, comparing in context tests for the two distributions. The statistics support the authors' claim so ok to keep it, but the reader should be able to make the appropriate comparisons based on the figures in the main text.

We agree that the original version of this figure could be misleading. Our initial goal was to illustrate mouse trajectories from participants who performed well on the task – those trained on $\alpha = 0$ in in-context test blocks, and those trained on $\alpha = 2$ in in-weights test blocks. In both cases, participants were “succeeding,” but likely relying on different strategies. This allowed us to highlight qualitative differences in inference dynamics. However, we understand that readers expect to see a direct comparison of training distributions within the same test block. To address this, we have updated Fig. 5B to show the relevant contrast – participants from different training conditions performing on the same in-context test.

Figure 5. Experiment 4. Transformers and humans use an induction mechanism for in-context learning. **A.** (right) Schematic representation of the computations realised by a two-layer transformer performing in-context learning. (left) Attention matrices of both layers for the example sequence. The transformer binds the representations of the images and the labels in attention head #1, and search for the target image in the context in attention head #2 (the “induction” head). **B.** Cursor trajectories of participants revealing their attention patterns. (top) Trajectories in the in-context test block for human participants trained on uniform ($\alpha = 0$) distribution. Participants search for the target image in the context and then associate it with the target label. (bottom) Trajectories in the in-context test block for participants trained on skewed ($\alpha = 2$) distribution (Exp. 4, $N = 20$ per group). Trajectories were aligned trial-by-trial to a common frame where the target image is located on the top of the context circle. Small lines are individual average trajectories, diamonds are group average trajectories.

6. At a conceptual level, it's not entirely clear to me what is the takehome message. Both agents can learn in context, but one is also resistant to catastrophic interference (and MLP is neither). What, then, is the message for human cognition? I realise this is a grand question that a single paper cannot answer, but the introduction also promises a lot, so it seems correct to ask it.

We show that both humans and transformer networks rely on two learning strategies – ICL and IWL – and that the balance between these strategies depends on the distribution of training data. In both systems, increasing the skewness of the distribution pushes learning from ICL to IWL, with a critical point around $\alpha \approx 1$. We also find that curriculum structure affects humans and models differently, with humans benefiting more from early diversity.

We fully agree that interpreting these parallels across such different systems is challenging, and we don't claim to offer a unified theory of human cognition. Our aim is more modest: to explore whether simple changes in training data lead to similar learning outcomes across humans and models. We believe the effects we reveal, including the shift around $\alpha \approx 1$, the dependence on the training data, and the divergent impact of curriculum, are non-trivial, and offer useful starting points for thinking about how different systems adapt their learning strategies as a function of the environment. We hope this work helps frame future studies that look more closely at when, how, and why ICL and IWL emerge in different learners. We have adapted the final paragraph of our Discussion accordingly:

(Discussion)

“Our work compares humans and transformer networks. We find that in one interesting respect – the emergence of in-weights learning and in-context learning in response to the training data distribution – they show some striking similarities. However, this should not be taken to imply overlap between humans and transformers at the algorithmic level. Indeed, other classes of neural network, including simple multi-layer perceptrons, may in principle be capable of in-context learning^{34,35}. Transformers are feedforward networks with a highly structured architecture based on self-attention, diverging sharply from the recurrent, feedback-driven, and biologically grounded computations of the human brain. Nevertheless, the way that they trade off memory-based strategies and inference-based strategies exhibits surprising commonalities with how this happens in human cognition.” (l. 621)

7. Relatedly, the comparisons between the two agents are a bit superficial; is it possible to do a more detailed fitting to human data so we can see which components precisely in a transformer are important for the match to human data. This question admittedly is itself a bit superficial, so ok to ignore it but I think a tighter match would be of interest, even if only as a promise for the future.

We agree that a tighter match between humans and models would be valuable. For now, we focused on qualitative behavioral similarities. As a small step, we have added an analysis comparing accuracies split by item frequency in response to a comment by Reviewer #2 (**Fig. S3**). A deeper look at which parts of the model drive alignment with human behavior is definitely something we would like to explore next.

Reviewer #1

Thanks to the authors for their deep engagement with my review; they've addressed my concerns, and I appreciate the clarifications, additional experiments and details. I really appreciated seeing the strategy learning curves. Also, the transitive inference results are quite nice, and enhance the generality of the findings. I maintain that this is a very interesting topic, and find this version to be a substantially improved paper, and I strongly recommend it for publication.

One minor note on the $\alpha = 1$ tradeoff: I would generally anticipate the possibility that $ICL + IWL_{common} > 1$ from a mixture of pure strategies; that's why I said the simplex interpolating between perfect and chance performance, not perfect and zero. Specifically, I would expect that the sum $ICL + IWL_{common}$ could be as large as $1 + chance_ICL + chance_IWL$ (if each of the models are doing perfectly on their preferred subset, and chance on the other) — which looks roughly compatible with the results shown in your response analysis. But Chan et al.'s performance exceeds that; the highest we'd expect from pure strategies is ~ 1.5 whereas their summed results are ~ 1.7 . It's a minor point though, and as the authors note a variety of differences in details like # of classes or data diversity might explain it.

We thank Reviewer #1 for his review of our manuscript.

Reviewer #2

In the first round, we thought the paper had the the potential to make a strong contribution, but had reservations about 1) the narrowness of the kind of ICL required in the main task (applying a +3 rule), and 2) the lack of a broader discussion situating the findings in the context of other work in cognitive psychology and neuroscience. The additional experiments and revisions the authors have made in response were substantial, and meaningfully address our major concerns.

Narrowness of ICL. The new modeling experiments using the transitive inference task show that the effect of the distribution of the training data remains intact in the models even when the task requires a more sophisticated form of ICL. Rather than requiring a simple application of a known +3 rule learned throughout training, this task required the models to make transitive inferences about arbitrary stimuli given in context. The emergence of this more sophisticated form of ICL also depended on the skewness of the training distribution, where more diversity (i.e., less skew) again resulted in greater use of the in-context strategy. While the ICL abilities of LLMs are much more sophisticated and flexible than any that could be learned in this kind of controlled setting, we think this new experiment makes a major step in this direction. And we acknowledge that it would be difficult to study the development of the ICL abilities of LLMs in a controlled manner. It would be nice to know if humans also exhibit the same effects in this transitive inference paradigm, but the modeling experiments presented in the revised manuscript substantively address our previous concerns about generalization to larger-scale ICL abilities such as those of LLMs.

Connections to other cognitive theories. The revisions to the title, introduction, and discussion qualify the major claims that we found problematic in the first version, and go some way toward relating the work to existing theories in cognitive psychology / neuroscience. We still think the findings are largely consistent with alternative cognitive perspectives (e.g., it isn't very surprising that humans don't put in the effort to extract a general rule when they can get most answers correct by simply memorizing a handful of items – see [1] and [2] below for example papers studying the conditions under which humans engage effortful controlled processing). And while we agree that this specific dependence of the learned strategy on the precise statistical properties of the training distribution may fall outside the scope of classical dual-process models, even there some computational frameworks do suggest that the engagement of control vs habitual (in-weight, model-free like processing) is arbitrated based on Bayesian uncertainty [3], [4], which is in turn related to frequency of learned associations. Including further discussion of how the basic distinction between ICL and IWL might relate to existing ideas from cognitive psychology/neuroscience (e.g., about working memory, when humans choose to exert cognitive effort to extract rules, model-based vs model-free processes etc.) might broaden the interdisciplinary appeal of the paper, but we can also see the appeal of emphasizing links to recent research in machine learning / AI, and acknowledge that it might be difficult to accomplish both in such a short paper.

We also found the many additional figures / analyses included in the recent revision to be very helpful. The figures showing the main results binned by frequency were clear, and give a fuller picture of the results (although see minor point below about confusing colorbar). We also found the additional experiment testing the LSTM with query-first inputs to be pretty interesting.

Overall the additional experiments and substantive revisions to the manuscript represent a major effort to address our concerns. We think the revised version is much improved and represents a strong contribution with many interesting results.

We thank Reviewer #2 for their review of our manuscript.

Other miscellaneous points:

We appreciate the change to the title, and the additional qualifications in the discussion section.

We noticed that there is still a colorbar showing different shades for different Zipf exponents in Figures S3 and S4. This appears to be a mistake that was left over from the previous figures? The shading now only corresponds to the frequency bins right?

This was indeed a mistake. We have now removed this colorbar.

We appreciated the explanation about why models learn the IWL strategy even after uniform training in curriculum C1.

[1] Shenhav, A., Botvinick, M. M., & Cohen, J. D. (2013). The Expected Value of Control: An Integrative Theory of Anterior Cingulate Cortex Function. *Neuron*, 79(2), 217–240. <https://doi.org/10.1016/j.neuron.2013.07.007>

[2] Frömer, R., Lin, H., Dean Wolf, C. K., Inzlicht, M., & Shenhav, A. (2021). Expectations of reward and efficacy guide cognitive control allocation. *Nature Communications*, 12(1), 1030. <https://doi.org/10.1038/s41467-021-21315-z>

[3] Daw ND, Niv Y, Dayan P. Uncertainty-based competition between prefrontal and dorsolateral striatal systems for behavioral control. *Nat. Neurosci.* 2005;8:1704–1711. doi: 10.1038/nn1560.

[4] Lee SW, Shimojo S, O'Doherty JP. Neural computations underlying arbitration between model-based and model-free learning. *Neuron.* 2014 Feb 5;81(3):687-99. doi: 10.1016/j.neuron.2013.11.028.

Reviewer #3

I think the authors have done a good job responding to the comments; they have collected additional data, evaluated robustness, and contextualized some of the claims. I have no remaining questions.

Two small comments (no followup required):

- I find it weird that for the replication the same (small) data sample size was used; but the predictions were replicated (and preregistered), so that is reassuring.
- In the intro, I don't think it's necessary to drag out each possible dual-system theory, some of which don't clearly link to the current one (e.g., associative versus symbolic). I see that this is in relation to another reviewer, so fine to keep it anyway.

We thank Reviewer #3 for their review of our manuscript.